# Optimality and NP-Hardness of Transformers in Learning Markovian Dynamical Functions

**Yanna Ding**[1]*, **Songtao Lu**[2] , **Yingdong Lu**[3], **Tomasz Nowicki**[3], **Jianxi Gao**[1]

[1] Department of Computer Science, Rensselaer Polytechnic Institute
[2] Department of Computer Science and Engineering, The Chinese University of Hong Kong
[3] IBM Research

{dingy6,gaoj8}@rpi.edu, stlu@cse.cuhk.edu.hk,
{yingdong,tnowicki}@us.ibm.com

## Abstract

Transformer architectures can solve unseen tasks based on input-output pairs in a given prompt due to in-context learning (ICL). Existing theoretical studies on ICL have mainly focused on linear regression tasks, often with i.i.d. inputs. To understand how transformers express ICL when modeling dynamics-driven functions, we investigate Markovian function learning through a structured ICL setup, where we characterize the loss landscape to reveal underlying optimization behaviors. Specifically, we (1) provide the closed-form expression of the global minimizer (in an enlarged parameter space) for a single-layer linear self-attention (LSA) model; (2) prove that recovering transformer parameters that realize the optimal solution is NP-hard in general, revealing a fundamental limitation of one-layer LSA in representing structured dynamical functions; and (3) supply a novel interpretation of a multilayer LSA as performing preconditioned gradient descent to optimize multiple objectives beyond the square loss. These theoretical results are numerically validated using simplified transformers.

## 1 Introduction

Transformer-based language models have demonstrated remarkable in-context learning (ICL) capabilities, predicting outputs for unseen inputs using only examples provided in the prompt, without parameter updates [1, 2, 3, 4, 5, 6, 7]. This phenomenon has motivated a growing body of theoretical work aiming to understand the mechanisms underlying ICL. Much of this work focuses on regression tasks with i.i.d. Gaussian inputs [8, 9, 10, 11, 12, 13, 14, 15, 16, 17, 11, 18, 19, 20, 21], showing that transformers can emulate classical algorithms like gradient descent. Others have begun to explore the limitations of transformer expressivity, especially under structured inputs [22, 23, 24, 25, 25, 26, 27, 28, 29], revealing that even moderate-depth transformers struggle with certain algorithmic tasks.

A complementary learning scenario involves predicting dynamical functions from temporally structured data, such as those governed by Markovian processes. Recent studies have made progress in understanding how transformers perform next-token prediction in structured single-sequence settings, including Markov chains [24, 22, 23], autoregressive models [30], and causal sequences [28, 29]. In contrast, we focus on a different task formulation: learning latent transition dynamics from multiple in-context sequences, where the model must integrate structural information across examples to generalize to a new sequence.

---

*Yanna Ding, Songtao Lu, Yingdong Lu, and Tomasz Nowicki contributed equally to this work.

39th Conference on Neural Information Processing Systems (NeurIPS 2025).

In this work, we study ICL for Markovian dynamical functions, where each in-context example is a trajectory sampled from a discrete-time Markov chain, and the task is to predict the next token of a query sequence by leveraging shared transition structure across the examples. This setting introduces new challenges that go beyond classical ICL tasks with i.i.d. inputs. In the standard regression case, each input-label pair corresponds to a single token with a simple statistical structure, typically zero-mean and independent. In contrast, each in-context example in our setting is a full sequence sampled from a Markov chain. While a convex reparameterization enables us to characterize the global optimum in an enlarged parameter space, mapping it back to transformer parameters is highly nontrivial due to dense parameter interactions induced by the underlying dynamics. These challenges call for a deeper understanding of how transformers generalize from dynamics-driven functions in context.

**Our contributions.** To this end, we study how transformers learn Markovian dynamical functions in-context through the lens of optimization. Given the challenges posed by non-convexity and stochasticity, we focus on binary Markov chains with first-order memory as our first step, which are a classical model for statistical language modeling [31, 32, 22]. The major contributions of this work are highlighted as follows.

▶ We establish an analytical framework for understanding ICL of Markovian dynamical functions, and characterize the global minimum of the loss landscape for 1-layer LSA under a tractable case of length-2 chains with both independent and correlated initial conditions. This result reveals how the optimal solution adapts to the Markovian dynamics, exhibiting denser structure compared to the i.i.d. linear regression case (Theorems 3.3, 3.4).

▶ Going beyond the length-2 case, we identify a fundamental gap between the global optimum of a structured prediction objective and what a 1-layer LSA can realize for arbitrary-length Markov chains. Even when the optimal solution is well-defined and analytically characterized in an extended space, recovering the corresponding transformer parameters is NP-hard. This realizability gap reveals a structural limitation of 1-layer LSA when learning Markovian dynamical functions in-context (Theorem 3.5).

▶ Finally, we advance the understanding of multilayer transformer expressivity by exploring a parameter subspace that mirrors the structure of the derived global minimum for Markovian dynamics. Our results show that the forward pass of the multilayer linear transformers is equivalent to solving a multi-objective optimization problem. This problem minimizes a squared loss while simultaneously maximizing multiple linear objectives (Theorem 4.1).

**Related work.** The capability of transformers to perform ICL [1, 33, 2, 4, 3] has inspired extensive work aiming to uncover the underlying mechanisms [34, 10, 35, 36, 37, 38, 39, 40, 41, 21, 30, 42]. These studies can be broadly categorized into two groups based on the nature of the in-context task.

(1) **Regression tasks.** This line of work typically focuses on linear regression, where each in-context token represents a pair consisting of an input i.i.d. sampled from a Gaussian distribution and its corresponding output generated by a ground-truth function. From the expressiveness perspective, [9, 8, 14] demonstrate that transformers trained on such prompts can implicitly perform classical algorithms such as gradient descent, ridge regression, and algorithm selection. From the optimization theory angle, [20, 19] analyze the loss landscape of trained LSA networks and show that they emulate preconditioned gradient descent. [43] extends this by proving that the global minimizer corresponds to multi-step preconditioned gradient descent in looped transformer architectures. From the viewpoint of training dynamics, [15, 16] prove convergence to the global optimum under mild distribution shift by leveraging the Polyak-Łojasiewicz (PL) condition.

Departing from the standard i.i.d. setup, our work considers input-output samples as realizations from a Markov chain governed by a shared kernel. We aim to characterize how transformers behave when the input distribution exhibits structured temporal dependencies and the prediction task is governed by the same latent dynamics.

(2) **Sequence generation.** Another line of work investigates transformer behavior in sequence generation tasks, particularly under structured or Markovian data. [22] analyzes transformers trained on first-order Markov data and shows that shallow models often converge to oversimplified solutions that ignore sequential dependencies, while deeper architectures can capture bigram transitions. [23] further demonstrates that the convergence behavior in this setting is highly sensitive to initialization and its alignment with the data structure. [24, 25] study transformers trained on higher-order

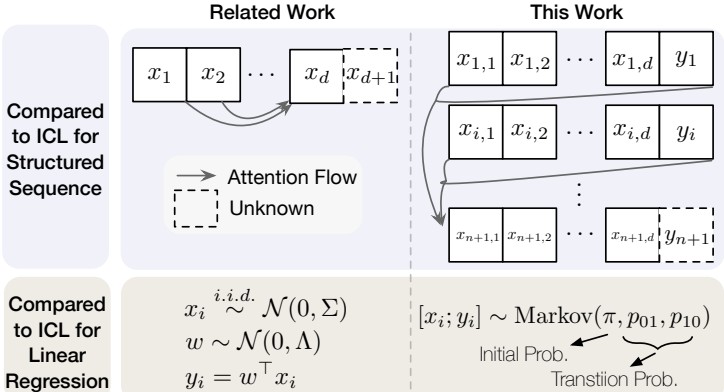

Figure 1: Comparison between the sequence-level in-context Markovian data based attention structures and the existing works. Top: Compared to ICL for structured sequences, the key difference is that the exiting studies of the attention mechanism [22, 30, 24, 28] is adopted on a token-level, whereas our study studies sequence-level attention. Bottom: While prior work for linear regression samples in-context input and task vectors independently from some given Gaussian distribution [19, 15], we consider input vectors generated through a Markovian transition kernel with parameters $p_{01}, p_{10}$ from given initial distributions.

Markov sequences, with a focus on how model depth and tokenization affect the ability to model such processes, respectively. While our setup also involves fixed-length substrings as input units, similar to [25], we do not study the role of tokenization. Instead, we focus on how attention organizes information across correlated examples to enable prediction for unseen inputs. On the other hand, several works [26, 27] have shown that transformers can learn to implement the induction head mechanism [44, 45] from an empirical perspective. Additionally, [28] proves that a two-layer transformer trained with a sequential layer-wise scheme can recover the empirical transition probabilities of single-parent Markov chains. [29] shows that gradient descent on $n$-gram data yields layered specialization: the first attention layer copies parent tokens, the feedforward network selects relevant features, and the final attention layer predicts the next token. Although our study also involves structured sequences, our formulation aligns more closely with regression-style ICL tasks: each token represents a full training sample, and the model learns to interpolate from these structured examples to predict a new query.

## 2 Preliminaries

### 2.1 In-Context Learning

ICL refers to the operation on a prompt consisting of $n$ input-output pairs and a query input $\boldsymbol{p} = (\{(x_i, y_i)\}_{i=1}^n, x_{n+1})$ where $y_i = h(x_i)$, $\forall i \in [n+1]$ for some unknown function $h \in \mathcal{H}$, and $x_i, y_i$ belong to some input space $\mathcal{X}$ and output space $\mathcal{Y}$, respectively. ICL aims to form an output $\hat{y}_{n+1}$ for the query input $x_{n+1}$ that approximates its true label $\hat{y}_{n+1} \approx h(x_{n+1})$. The function $h : \mathcal{X} \to \mathcal{Y}$ remains the same within a single prompt yet varies across prompts.

Prior works have focused on linear function space $\mathcal{H}$: $h(x) = y = w^\top x$ for some $w \in \mathcal{X}$. Under such a construction, $y$ is deterministic once $x$ is provided. Despite being commonly encountered in many real-world applications, the case where $h$ is stochastic remains largely unexplored. For example, $h$ can represent a text generation mechanism that provides descriptions revolving a given topic. Then the token generated in the next step is associated with a probability based on the previously generated words [46]. To approach the ICL for such scenarios, we consider a simplified setting of next token prediction for Markov chains. The state space resembles vocabulary and the transition probability is akin to the conditional probability of the next word given the previous text.

## 2.2 Markov Chains

The evolution of a Markov chain $s$ of order $k$ on a state space $\mathcal{S}$ depends solely on the $k$ most recent states. For time step $\tau \in \mathbb{Z}_{\geq 1}$, we let $s_\tau$ denote the $\tau$th state in the sequence $s$, the probability of observing state $j \in \mathcal{S}$ at time step $\tau + 1$ is: $\mathbb{P}(s_{\tau+1} = j \mid s_{1:\tau}) = \mathbb{P}(s_{\tau+1} = j \mid s_{\tau-k+1:\tau})$. where $s_{\tau_1:\tau_2}$ denotes the subsequence from time step $\tau_1$ to $\tau_2$. For first-order Markov chains, the dynamics are determined by the transition probabilities $p_{ij} := \mathbb{P}(s_{\tau+1} = j \mid s_\tau = i)$, which indicate the probability of transitioning from state $i \in \mathcal{S}$ to state $j \in \mathcal{S}$. These probabilities constitute the Markov kernel $\mathsf{P} = (p_{ij}) \in [0,1]^{|\mathcal{S}| \times |\mathcal{S}|}$. For a binary state space $\mathcal{S} = \{0, 1\}$, the transition matrix is represented as $\mathsf{P}(p_{01}, p_{10}) := [1 - p_{01}, p_{01} ; p_{10}, 1 - p_{10}]$. Let $\pi_\tau \in [0,1]^{|\mathcal{S}|}$ denote the marginal probability at the $\tau$th time step. A binary Markov chain $s \sim (\pi_1, \mathsf{P}(p_{01}, p_{10}))$ can be generated by starting with an initial distribution $\pi_1$ and iteratively applying $\mathsf{P}(p_{01}, p_{10})$ to update the state probabilities at each time step.

## 2.3 Data Formalism

We introduce the input embedding matrix formulation used for our theoretical results. For a Markov chain $s$ with length $d + 1$, we take its first $d$ states to be the input $x = s_{1:d}$ and the final state to be the output $y = s_t$. The input and output space are $\mathcal{X} = \mathcal{S}^d$ and $\mathcal{Y} = \mathcal{S}$. We use subscripts to denote the indices of in-context samples, such that $x_i$ represents the first $d$ time steps of the $i$th in-context Markov chain, while $y_i$ denotes its final state. To form an input embedding matrix $Z_0 \in \mathcal{S}^{(d+1) \times (n+1)}$, we stack $(x_i, y_i) \in \mathcal{S}^{d+1}$ as the first $n$ columns and let the last column be $(x_{n+1}, 0)$, inspired by [15].

$$Z_0 = \begin{bmatrix} x_1 & x_2 & \cdots & x_n & x_{n+1} \\ y_1 & y_2 & \cdots & y_n & 0 \end{bmatrix} \tag{1}$$

where $z_i := [x_i; y_i] \sim (\pi_1, \mathsf{P}(p_{01}, p_{10}))$ for initial probability mass function $\pi_1 = [1 - p, p]$ with $p \in (0, 1)$ and transition probabilities $p_{01}, p_{10} \sim U(0, 1)$. We use double subscripts $(x_{i,j})$ to indicate the $j$th entry of the $i$th in-context input sequence. The Markov kernel varies for each prompt, while the initial probability $p$ remains constant across all prompts. Let TF denote a transformer-based autoregressive model. The goal of ICL is to learn a model TF that can accurately predict the label of the query input: $\hat{y}_{n+1} := \text{TF}(Z_0) \approx y_{n+1}$.

## 2.4 Model and Training Objective

We consider transformers with LSA layers [8, 19]. We recall a single-head self-attention layer [47] parameterized by key, queue, value weight matrices are defined as follows:

$$\text{Attn}_{W_{k,q,v}}(Z) = W_v Z M \cdot \text{softmax}\left(Z^\top W_k^\top W_q Z\right) \tag{2}$$

$$M := \begin{bmatrix} I_{n \times n} & 0 \\ 0 & 0 \end{bmatrix} \in \mathbb{R}^{(n+1) \times (n+1)} \tag{3}$$

where $W_k, W_q, W_v \in \mathbb{R}^{(d+1) \times (d+1)}$ are the (key, queue, value) weight matrices and $I_{n \times n}$ denotes the identity matrix. The attention scores are normalized by the $\text{softmax}$ operator. The mask matrix $M$ reflects the asymmetric prompt due to the absence of the label for $x^{(n+1)}$. Motivated by [19, 15], we simplify the architecture by (i) removing the $\text{softmax}$ nonlinearity and (ii) reorganizing the weights as $P := W_v$ and $Q := W_k^\top W_q$, merging the query and key matrices into a single matrix:

$$\text{Attn}_{P,Q}^{(\text{lin})}(Z) = PZM(Z^\top Q Z). \tag{4}$$

Despite its simplicity, LSA demonstrates ICL capability for linear functions [15] and has been shown to implement gradient descent [8] and preconditioned gradient descent [19] to solve linear regression in-context. We will prove in Sec. 4 that certain parameter configuration implements preconditioned gradient descent for a multi-objective optimization problem that includes linear regression. Finally, we consider architecture consists of $L$-layer LSA modules. Let $Z_l$ denote the output of the $l$th layer attention, we have

$$Z_{l+1} = Z_l + \frac{1}{n} \text{Attn}_{P_l, Q_l}^{(\text{lin})}(Z_l) \tag{5}$$

$$= Z_l + \frac{1}{n} P_l Z_l M(Z_l^\top Q_l Z_l) \tag{LSA}$$

for $l = 0, \ldots, L - 1$. The normalizing factor $n$ averages the attention weights gathered from the in-context examples. We consider the output of the transformer to be the bottom-right entry of the $L$th layer, i.e., $\text{TF}_L(Z_0; \{P_l, Q_l\}_{l=0,\ldots,L-1}) = [Z_L]_{(d+1),(n+1)}$. To train the in-context learner, we optimize the following population loss in the limit of an infinite number of training prompts such that each prompt corresponds to a distinct Markov kernel $\{p_{ij}\}_{i,j \in \mathcal{S}}$:

$$f(\{P_l, Q_l\}) = \mathbb{E}[(\text{TF}_L(Z_0; \{P_l, Q_l\}) - y_{n+1})^2] \tag{6}$$

where $l \in \{0, \ldots, L - 1\}$ and the expectation is taken over $Z_0, \{p_{ij}\}_{i,j \in \mathcal{S}}$. This objective function formulates the in-context task as last-token prediction for in-context sequences. Once trained, the model can autoregressively predict the next token of the query sequence. Unlike ICL for linear regression tasks [19, 15], where the in-context input distribution assumes a zero-mean Gaussian and the input-output relationship is linear, our setting involves a Markovian input and a stochastic input-output relationship. Furthermore, compared to ICL for a single sequence [28, 24], attention is applied across sequences rather than being restricted to local tokens within a single sequence.

# 3 Global Minimizers and Expressive Limits of 1-Layer LSA

In this section, we analyze the loss landscape of the in-context objective function $f$. To address its nonconvexity, we introduce a reparameterized objective in an expanded parameter space, resulting in a strictly convex formulation with a closed-form global minimum. This analysis shows that achieving optimality in $f$ requires more parameters than in linear tasks.

**Parameter space.** For single-layer LSA, only the last row of $P_0$ and the first $d$ columns of $Q_0$ affect the output. Therefore, we consider optimization over the following subset of $P_0$ and $Q_0$:

$$P_0 = \begin{bmatrix} 0_{d \times (d+1)} \\ b^\top \end{bmatrix}, \quad Q_0 = [A \quad 0_{d+1}] \tag{7}$$

where $b \in \mathbb{R}^{d+1}, A \in \mathbb{R}^{(d+1) \times d}$. Throughout this section, we assume that $P_0$ and $Q_0$ follow the above format and refer to them as $P$ and $Q$ for simplicity.

**Reparameterized objective.** We define a reparameterization $\phi$ which maps from LSA parameter space to $\mathbb{R}^{dm}$, where $m = \frac{(d+2)(d+1)}{2}$:

$$X_r = \phi(b, A) = \begin{cases} b_i A_{k,j}, & \text{if } i = k; \\ b_i A_{k,j} + b_k A_{i,j}, & \text{otherwise} \end{cases} \tag{8}$$

where $r = (j-1)m + i(d+1) + k - \sum_{i' \le i}(i' - 1)$. Here $\phi(\cdot)_r$ is the $r$th entry of the resulting vector in $\mathbb{R}^{dm}$ and $A_{k,j}$ denotes the $(k, j)$-th entry of $A$ and $b_i$ denotes the $i$th element of $b$. For clarity, we use $X$ to represent $\phi(b, A)$. Let $\tilde{f} : \mathbb{R}^{dm} \to \mathbb{R}$ denote the reparameterized objective s.t. $\tilde{f}(\phi(b, A)) = f(P, Q)$.

We collect the unique elements in the symmetric data matrix $\frac{1}{n} \sum_{i=1}^{n} z_i z_i^T$ into a vector g. Then $f$ can be expressed as a square loss of a linear model parameterized by $X$:

$$\tilde{f}(X) = \mathbb{E}\left[\left((x_{n+1} \otimes \text{g})^\top X - y_{n+1}\right)^2\right]. \tag{9}$$

The equivalence of the objective function before and after reparameterization is verified in Appendix C.1,C.3. The reparameterized objective, $\tilde{f}(X)$, exhibits the following desired property:

**Lemma 3.1 (*Strict convexity*).** *Suppose the initial probability of the Markov chains is $\pi_1 = [1 - p, p]$ with $p \in (0, 1)$ and the transition probabilities are sampled from $U(0, 1)$. Then $\tilde{f}$ (Eq. 9) is strictly convex w.r.t. $X \in \mathbb{R}^{dm}$.*

The proof, provided in Appendix C.2, leverages the nonzero transition probabilities and the properties of Markov chains to establish the positive definiteness of the Hessian of $\tilde{f}$. Consequently, $\tilde{f}$ is strictly convex, ensuring the existence and uniqueness of its global minimizer, denoted as $X^*$. We derive its expression by solving for the zero of the gradient of $\tilde{f}$ below.

**Lemma 3.2** (*Global minimum for reparameterized objective*). *Consider the in-context learning of length-$d+1$ ($d \geq 1$) Markov chains $\{(x_i, y_i)\}_{i=1}^n$ ($x_i, y_i \in \{0,1\}$) with transition kernel*
$\mathsf{P} = \begin{bmatrix} p_{00} & p_{01} \\ p_{10} & p_{11} \end{bmatrix} \in (0,1)^2$. *Suppose the initial states $x_i$ are i.i.d. sampled from $Bernoulli(p)$ for some constant $p \in (0,1)$. Consider indices $i, j \in [d]$, $i', j', k', l' \in [d+1]$ with $i' \leq j', k' \leq l'$. We denote $t_1 \leq t_2 \leq t_3 \leq t_4$ as the sorted version of $(i', j', k', l')$. Define $H \in \mathbb{R}^{dm \times dm}$ as*

$$H_{r,c} = \frac{1}{n}\mathbb{E}\left[ \left( p(\mathsf{P}^{t_1-1})_{11} + (1-p)(\mathsf{P}^{t_1-1})_{01} \right) \ (\mathsf{P}^{t_2-t_1})_{11}(\mathsf{P}^{t_3-t_2})_{11}(\mathsf{P}^{t_4-t_3})_{11} \right] +$$
$$\frac{n-1}{n}\mathbb{E}\left[ \left( p(\mathsf{P}^{i'-1})_{11} + (1-p)(\mathsf{P}^{i'-1})_{01} \right) (\mathsf{P}^{j'-i'})_{11} \ \left( p(\mathsf{P}^{k'-1})_{11} + (1-p)(\mathsf{P}^{k'-1})_{01} \right) (\mathsf{P}^{l'-k'})_{11} \right]$$

*where $r = (i-1)m + j' + \sum_{\tau=0}^{i'-2} d + 1 - \tau$, $c = (j-1)m + l' + \sum_{\tau=0}^{k'-2} d + 1 - \tau$. Define $b \in \mathbb{R}^{dm}$ as*

$$b_r = \mathbb{E}\left[ \left( p(\mathsf{P}^{j-1})_{11} + (1-p)(\mathsf{P}^{j-1})_{01} \right) (\mathsf{P}^{d+1-j})_{11} \ \left( p(\mathsf{P}^{i'-1})_{11} + (1-p)(\mathsf{P}^{i'-1})_{01} \right) (\mathsf{P}^{j'-i'})_{11} \right]$$

*for $r = (j-1)m + j' + \sum_{\tau=0}^{i'-2} d + 1 - \tau$. The global minimum $X^* \in \mathbb{R}^{dm}$ of the objective function described in Eq. 9 equals $X^* = H^{-1}b$.*

The full derivation, given in Appendix C.3, utilizes the first-order dependence in the in-context sequences to evaluate the expected value of each token. Specifically, the expectation of each token is expressed as the probability of the first token multiplied by successive powers of the transition kernel.

**Global minimum of the ICL objective.** We now characterize the global minimum of the original ICL loss. It suffices to find $(b, A)$ such that $\phi(b, A) = X^*$. However, since $\phi$ maps into an expanded parameter space, an inverse mapping is not always guaranteed. We derive an analytic solution where possible and provide an approximation for the more general case.

The following result presents the analytic solution in length-2 Markov chains.

**Theorem 3.3** (*Global minima for i.i.d. in-context initial states*). *Consider the in-context learning of length-2 Markov chains $\{(x_i, y_i)\}_{i=1}^n$ ($x_i, y_i \in \{0,1\}$) with transition probabilities $p_{01}, p_{11} \sim U(0,1)$. Suppose the initial states $x_i$ are i.i.d. sampled from $Bernoulli(p)$ for some $p \in (0,1)$.*

*Let $X^* := H^{-1} \begin{bmatrix} p^2/2 & p^2/3 & p^2/12 + p/4 \end{bmatrix}^\top$, where $H$ is a symmetric matrix defined as follows (repeating entries in the lower half triangle are omitted)*

$$p \begin{bmatrix} \frac{p}{n} + \frac{(n-1)p^2}{n} & \frac{p}{2n} + \frac{(n-1)p^2}{2n} & \frac{p}{2} \\ & \frac{p}{2n} + \frac{(n-1)p^2}{3n} & \frac{p}{2n} + \frac{(n-1)\left(\frac{p}{4}+\frac{p^2}{12}\right)}{n} \\ & & \frac{1}{2n} + \frac{(n-1)\left(\frac{1}{3}-\frac{p}{6}+\frac{p^2}{6}\right)}{n} \end{bmatrix}.$$

*Then the following choice of parameters*

$$P = \begin{bmatrix} 0 & 0 \\ 1 & \frac{X_2^* \pm \sqrt{X_2^{*2} - 4X_1^* X_3^*}}{2} \end{bmatrix} \quad Q = \begin{bmatrix} X_1^* & 0 \\ X_2^* - \frac{X_1^* X_2^* \pm X_1^* \sqrt{X_2^{*2} - 4X_1^* X_3^*}}{2} & 0 \end{bmatrix} \tag{10}$$

*is a global minimizer of $f(P, Q)$, where $X_i^*$ is the $i$th element of $X^*$.*

The proof is given in Appendix C.1. We observe that the key LSA parameters are nontrivial in the optimizer, unlike in-context linear tasks with zero-mean Gaussian feature and task vectors, which result in a sparser structure where the first $d$ entries of $b$ and the last row of $A$ is zero [19, 16, 15].

The independence assumption on the initial states in Theorem 3.3 can be relaxed, and the global minima of $f(P, Q)$ still maintain the same structure as in the independent case.

**Theorem 3.4** (*Global minima for generalized in-context initial states distribution*). *Consider the in-context learning of length-2 Markov chains $\{(x_i, y_i)\}_{i=1}^n$ ($x_i, y_i \in \{0,1\}$) with transition probabilities $p_{01}, p_{11} \sim U(0,1)$. Suppose the initial states $x_i$ are sampled from $Bernoulli(p)$ for some constant $p \in (0,1)$. Let $c_1 = \sum_{i=1}^n \mathbb{E}[x_i x_{n+1}]$, $c_2 = \sum_{i=1}^n \sum_{j=1, j\neq i}^n \mathbb{E}[x_i x_j x_{n+1}]$.*

*Define $X^*$ as $H^{-1} [c_1/2n \quad c_1/3n \quad p/4 + c_1/12n]$, and $H$ is a symmetric matrix defined as follows*

$$\begin{bmatrix} \frac{c_1}{n^2} + \frac{c_2}{n^2} & \frac{c_1}{2n^2} + \frac{c_2}{2n^2} & \frac{c_1}{2n} \\ & \frac{c_1}{2n^2} + \frac{c_2}{3n^2} & \frac{(n+1)c_1}{4n^2} + \frac{c_2}{12n^2} \\ & & \frac{(2n+1)p}{6n} - \frac{(n-1)c_1}{6n^2} + \frac{c_2}{6n^2} \end{bmatrix}.$$

*(repeating entries in the lower half triangle are omitted)*

*Then, substituting $X^*$ into Eq. 10 gives a global minimizer of $f(P, Q)$.*

The proof is deferred to Appendix C.2. The global minimizer is determined by the joint expectation of the query and in-context samples.

While the global minimizer $X^*$ of our reparameterized loss function $\tilde{f}(X)$ can be characterized analytically by convexity, it remains unclear whether such a solution can be realized by the actual transformer parameters $(b, A)$. In particular, the correspondence between $X^*$ and feasible transformer configurations is highly nontrivial as the input space exhibits structured dependencies. This motivates a fundamental question: given an optimal representation $X^*$ in the extended parameter space, is it computationally feasible to recover any compatible transformer parameters that achieve it? The following result addresses this question and reveals a structural limitation of 1-layer LSA transformers.

**Theorem 3.5** (***NP-hardness of transformer parameter reconstruction from reparameterization***). *Let $d \in \mathbb{Z}_{\geq 1}$ denote the dimension of the in-context data (the length of the Markov chain minus one), and let $m = \frac{(d+1)(d+2)}{2}$. Given $X \in \mathbb{R}^{dm}$ representing the global minimizer of the reparameterized loss, solving for 1-layer LSA parameters $(b, A)$ that satisfy the reparameterization equation Eq. 8 is NP-hard with respect to $d$.*

*Proof.* (sketch) We show that solving the reparameterization equation for transformer parameters $(b, A)$ belongs to a general class of bilinear feasibility problems. Specifically, we recast the system as a collection of bilinear constraints $b^\top D^{(r)} A_{:,j} = X_r$ $(r \in [dm])$, where each matrix $D^{(r)}$ encodes the structure of the corresponding term in $\phi(b, A)$. This allows us to further express the problem as a bilinear program with an objective and constraints over $(b, A)$.

To establish hardness, we reduce from the bilinear separability problem, as formulated in system (13) of Theorem 3.1 in [48], which asks whether two point sets in $\mathbb{R}^n$ can be strictly separated by a pair of hyperplanes such that one set occupies exactly three of the four induced regions. This problem is known to be NP-complete.

We construct a variant of our bilinear program through two standard transformations: variable splitting (to impose nonnegativity) and variable fixing (to introduce linear structure). A subset of transformer parameters is assigned to represent the decision variables in the separability problem. Other variables are fixed to constants to encode the geometric structure and auxiliary terms of the separability constraints. Through this construction, each constraint in the separability problem is converted to a constraint in our system.

Our bilinear program subsumes the bilinear separability formulation from [48]. First, the objective functions are equivalent: we replicate their bilinear objective by selecting a corresponding set of $(b, A)$ variables and constructing a matrix $D^{(r)}$ that enforces the same multiplicative interaction. Second, for every constraint in the separability problem, there exists a corresponding constraint in our system constructed via a combination of variable assignment and fixing. Finally, our program includes additional constraints beyond those in the separability setting, arising from the full reparameterization of the transformer system. This establishes that our recovery problem is at least as hard as bilinear separability, and therefore NP-hard. $\qquad \square$

The complete proof is provided in Appendix C.4, where we show that the dimension $d$ scales polynomially with the parameters of the separability program. This indicates that the problem remains NP-hard with respect to its dimension, as the reduction preserves the scaling of problem size.

In prior work [19], the reparameterized loss under an i.i.d. linear regression setting admits a structural simplification. Under Gaussian inputs, the optimal solution $X^*$ becomes sparse: each group of reparameterized variables associated with a column of $A$ contains only a single nonzero entry. This sparsity allows the full bilinear system to reduce to $d$ independent constraints, each involving a single

bilinear term between $b$ and $A_{:,j}$, which can be analytically solved. In our setting with Markovian input sequences, the reparameterized solution $X^*$ does not exhibit such sparsity. The temporal structure of the data introduces statistical dependencies and nonzero mean patterns that persist across in-context samples, leading to dense coupling between variables in the bilinear system. This increased structural complexity motivates the hardness result above.

Therefore, even if the global optimum can be attained in an extended parameter space, it may not correspond to any realizable configuration in the original transformer parameter space. Moreover, since the reparameterization defines an onto mapping from the transformer's native parameter space $(P, Q)$ to the enlarged space $X$, the optimal value $\tilde{f}^* := \min_X \tilde{f}(X)$ provides a lower bound for the best performance achievable by the transformer, i.e., $\tilde{f}^* \leq \min f(P, Q)$. This reveals an inherent architectural limitation: the 1-layer transformer's representational capacity may be insufficient to express dynamical functions induced by structured inputs, such as those governed by shared Markovian dynamics. Example C.16 in Appendix C.3 illustrates the absence of a corresponding LSA parameter for $X^*$ in longer Markov chains. Nevertheless, the optimal configuration of $(P, Q)$ maintains the same structure as in the length-2 case.

## 4 Multilayer LSA Implements Preconditioned Multi-objective Optimization

We now demonstrate that the forward pass of an $L$-layer LSA can be interpreted as preconditioned multi-objective optimization when trained within the parameter space with the following structure:

$$P = \begin{bmatrix} 0_{d \times (d+1)} \\ b_l \end{bmatrix}, \quad Q = - \begin{bmatrix} \bar{A}_l & 0_{d+1} \\ a_l & 0 \end{bmatrix}. \tag{11}$$

We identify two groups of objective functions, $R_1, R_2 : \mathbb{R}^d \to \mathbb{R}^{d+1}$, for the linear model $w^\top x_i$ ($w \in \mathbb{R}^d$), such that $\text{TF}_L$ performs gradient descent on these objectives, preconditioned by $b_l, \bar{A}_l, a_l$. Notably, this result does not rely on taking the expectation of the objective and holds for any given prompt instance.

**Theorem 4.1** (*Forward pass as minimizing multiple objectives*). *Consider the L-layer transformer parameterzed by $b_l, A_l = - \begin{bmatrix} \bar{A}_l \\ a_l^\top \end{bmatrix}$ where $b_l \in \mathbb{R}^{d+1}, \bar{A}_l \in \mathbb{R}^{d \times d}, a_l \in \mathbb{R}^d$ for $l \in [L]$. Let $y_{n+1}^{(l)}$ be the bottom-right entry of the lth layer output. Then $y_{n+1}^{(l)} = \langle w_l, x_{n+1} \rangle$ where $w_l$ is optimizing two multi-objective problems $R_1, R_2 : \mathbb{R}^d \to \mathbb{R}^{d+1}$, iteratively defined as follows: $w_0 = 0$ and*

$$w_{l+1}^\top = w_l^\top - b_l^\top \left( \nabla R_1(w_l) \bar{A}_l + \nabla R_2(w_l) \begin{bmatrix} 0_{(d-1) \times d} \\ a_l^\top \end{bmatrix} \right), \text{ where } R_1(w) = \frac{1}{n} \sum_{j=1}^n \begin{bmatrix} -x_i \otimes \langle w, x_j \rangle \\ (\langle w, x_j \rangle - y_j)^2 \end{bmatrix},$$

$$R_2(w) = \frac{1}{n} \sum_{j=1}^n \begin{cases} \begin{bmatrix} -(w_d(y_j - \langle w_{:d-1}, x_{j,:d-1} \rangle) + \frac{w_d^2}{2} x_{j,d}) x_j \\ \frac{1}{3x_{j,d}} (y_j - w^\top x_j)^3 \end{bmatrix} & \text{if } x_{j,d} \neq 0 \\ \begin{bmatrix} -(w_d(y_j - \langle w_{:d-1}, x_{j,:d-1} \rangle) + \frac{w_d^2}{2} x_{j,d}) x_j \\ -(y_j - \langle w_{:d-1}, x_{j,:d-1} \rangle)^2 w_d \end{bmatrix} & \text{if } x_{j,d} = 0 \end{cases}$$

The full derivation is provided in Appendix D.

In the above expression, $w_{:d-1}, x_{j,:d-1}$ denotes the first $d-1$ entries of $w$ and $x_j$, respectively. The first $d$ terms in $R_1$ ($x_{j,k} w^\top x_j$ for $k \in [d]$) capture the interaction between the $k$-th state of $x_j$ and the linear combination of states determined by $w$. These terms assign a weight to each state's contribution to the overall objective, emphasizing their individual roles within the sequence. The final term,

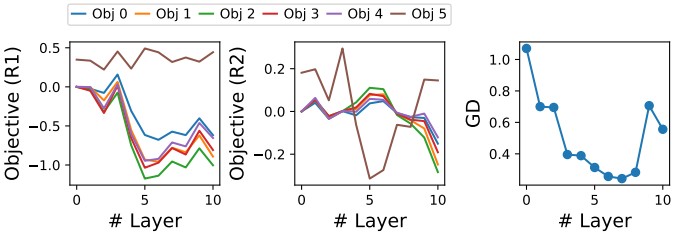

Figure 2: Preconditioned multi-objective optimization performance of a trained LSA across layers. (a) Values for the six objectives in the first group ($R1$), (b) Values for the six objectives in the second group ($R1$), and (c) Generational Distance (GD) measuring deviation from the Pareto front. The model is a 10-layer LSA trained on 100 in-context Markov chains of length 5.

$(w^\top x_j - y_j)^2$ ensures alignment between the linear model's prediction and the target state $y_j$.

$R_2$ specifically emphasizes the role of $w_d$, aligning with the structure of the last preconditioning matrix, which focuses updates on the last parameter. The first $d$ terms in $R_2$ scale the target state $y_j$ by $x_j$ and $w_d$, with additional quadratic terms like $\frac{w_d^2}{2} x_{j,d}$. These terms capture the influence of $w_d$, the alignment between $y_j$ and partial prediction $\langle w_{:d-1}, x_{j,:d-1} \rangle$, and the value of $x_{j,d}$, emphasizing key components of the input sequence. The final objective is in cubic penalty form $\frac{1}{3x_{j,d}}(y_j - w^\top x_j)^3$ when $x_{j,d} \neq 0$, emphasizing sequences with smaller $x_{j,d}$. When $x_{j,d} = 0$, the penalty changes to a quadratic term $-(y_j - \langle w_{:d-1}, x_{j,:d-1} \rangle)^2 w_d$, focusing solely on aligning $y_j$ with the partial prediction based on $w_{:d-1}$. This adaptation ensures that the optimization prioritizes the appropriate components of the input sequence depending on the presence or absence of $x_{j,d}$. Furthermore, when $x_{j,d} = 0$, the final objective becomes convex if $w_d < 0$ and concave otherwise.

**Empirical validation.** We evaluate the role of transformer weights in preconditioned multi-objective optimization, as proved in Theorem 4.1. We train a 10-layer LSA model within the parameter space specified by Eq. 11 on 100 in-context Markov chains of length 5, sampled with an initial probability of 0.5. As the forward pass progresses through deeper layers, we track the values of the multi-objectives and measure Generational Distance (GD) to quantify deviation from the Pareto front.

Fig. 2 demonstrates that initially, LSA weights move closer to the Pareto front, indicating effective multi-objective optimization. However, beyond a certain depth, all objective values begin to increase simultaneously, suggesting that the optimization process deteriorates rather than balancing competing objectives. This behavior implies that while TF initially optimizes multiple objectives, deeper layers may prevent sustained improvement, potentially due to the nature of the restricted parameter space.

To investigate how model depth and data complexity affect performance, we conduct synthetic experiments varying the number of layers and the size of the state space. Prompts are constructed from Markov chains with $|\mathcal{S}| \in \{2, 3, 4\}$, each consisting of 10 in-context sequences and one query of length 6. We train LSA models with $L \in \{1, 2, 3\}$ layers using gradient descent, and report mean accuracy over 5 seeds (see Fig. 3). The results show 1-layer LSA performs poorly, close to random guessing, regardless of the state space size. As the number of layers increases, accuracy improves significantly. For instance, in the $|\mathcal{S}| = 3$ case, the mean accuracy increases from $0.55$ (1-layer) to $0.98$ (3-layer). This aligns with our theoretical findings: while a single-layer transformer struggles to capture structured dynamics, slightly deeper architectures allow the forward computation to approximately reduce squared loss, as revealed by our multi-objective interpretation. Additional experiments using self-attention models and GPT-2 architectures under varied data and architectural configurations are provided in Appendix A,B.

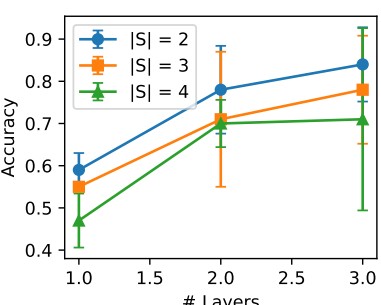

Figure 3: Accuracy of LSA models trained on Markovian prompts w.r.t. number of layers. Each curve corresponds to a different state space size.

## 5  Conclusion

In this work, we study ICL of Markovian dynamical functions using a LSA model. Focusing on one-layer transformers, we analyze the loss landscape induced by prompts constructed from first-order binary Markov chains. Our results show that the global minimum adapts to the underlying dynamics and deviates significantly from the sparsity structures typically observed in i.i.d. regression tasks.

Despite the existence of an analytically characterized optimum in an extended parameter space, we prove that recovering corresponding transformer parameters is NP-hard. This establishes a fundamental limitation: one-layer LSA transformers may be unable to realize optimal solutions,

even when those solutions are simple and fully specified. The result reveals a representational and computational gap between what is learnable in principle and what the model can express.

Finally, we interpret the forward pass of multilayer LSA models as performing a form of multi-objective optimization, which includes squared loss as one of the components. This formulation offers insight into why deeper architectures empirically outperform shallow ones when learning structured dynamical patterns.

**Limitation.** Our theoretical analysis focuses on 1-layer LSA models and first-order Markov chains. These choices allow us to isolate key structural effects and obtain analytically tractable results, such as characterizing the global minimum and proving NP-hardness of parameter recovery. However, this framework does not capture the full expressivity of modern transformers, which typically involve nonlinear attention mechanisms, multi-layer architectures, and learned positional encodings. Moreover, real-world dynamical processes often involve higher-order dependencies or more complex causal structures beyond simple Markovian assumptions. Extending our framework to handle richer classes of causal data, deeper networks, and nonlinear components such as softmax attention and MLP blocks remains an important direction for future work.

# 6    Acknowledgements

This work was supported by the IBM-Rensselaer Future of Computing Research Collaboration and the U.S. National Science Foundation project 2047488.

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

# Appendix

## A  Comparative Analysis of Setups

We train two variants of single-layer self-attention models to in-context learn length-2 Markov chains using gradient descent over 10K random prompts.

1. Variant 1 ($\mathrm{LSA}_{P,Q}^{(\mathrm{sparse})}$): LSA (Eq. 4) parameterized by sparse $P, Q$ (Eq. 7)

$$\begin{cases} Z_1 = Z_0 + \frac{1}{n}PZM(Z^\top QZ) \\ P,Q \in \{(\begin{bmatrix} 0 & 0 \\ b_1 & b_2 \end{bmatrix}, \begin{bmatrix} a_1 & 0 \\ a_2 & 0 \end{bmatrix}) \mid a_i, b_i \in \mathbb{R}\} \end{cases}$$

2. Variant 2 ($\mathrm{LSA}_{P,Q}$): LSA (Eq. 4) parameterized by dense $P, Q$

$$\begin{cases} Z_1 = Z_0 + \frac{1}{n}PZM(Z^\top QZ) \\ P,Q \in \mathbb{R}^{2\times 2} \end{cases}$$

To justify the choice of the sparse parameter space, we plot the training loss curve of the above three variants in Fig. 4. The loss value is computed as the mean squared error for the query sequence averaged over $B$ random prompts. We set $B = 100$ and use 30 in-context examples for each prompt. The in-context sequences are Markov chains with initial probability 0.3 and transition probabilities $p_{01}, p_{10}$ sampled from $U(0, 1)$. The results demonstrate that the loss curves under variant 1 and 2 converge to nearly the same value, indicating that the sparse and dense parameter matrices perform equivalently for LSA.

## B  Additional Experiments

In this section, we first illustrate the limitation of 1-layer LSA as the problem dimension $d$ increases, consistent with the NP-hardness analysis in Theorem 3.5. We then empirically explore the expressivity of deeper transformer architectures by evaluating GPT-2 models with varying numbers of layers. The results show that performance improves as the number of layers increases, suggesting that additional depth helps mitigate the limitations observed in shallow models.

### B.1  Limitation of 1-Layer LSA

Theoretically, we showed that while the reparameterized model admits an analytically computable global minimizer, recovering the corresponding LSA parameters is NP-hard due to bilinear coupling.

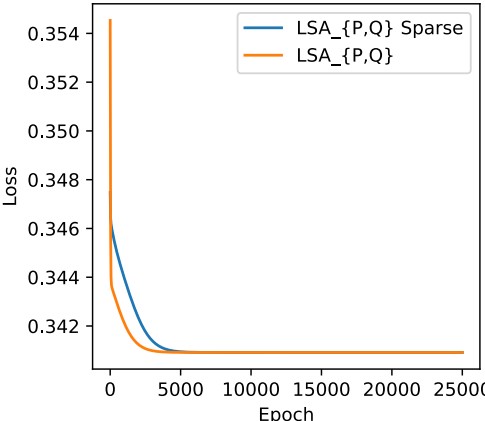

Figure 4: Training loss w.r.t epochs for variants of the self-attention models, evaluated on 100 random prompts, each containing 30 in-context samples and a query sequence.

The hardness stems from a bilinear feasibility program whose dimension scales polynomially with the dimension $d$ of the input Markov chain. To empirically verify our theoretical findings on the limitations of 1-layer LSA, we measure how the performance gap between the reparameterized model and the LSA model evolves with increasing dimension $d$.

For data, we construct each prompt with $n + 1 = 101$ sequences, where each sequence represents a binary Markov chain of length $d + 1$. The first $n$ sequences serve as in-context examples, and the final sequence $x_{n+1}$ is a query with its last token $y_{n+1}$ masked. The prediction task is to infer $y_{n+1}$ based on the shared temporal dynamics among chains. The initial states $x_{i,1}$ are independently sampled from $Bernouli(0.5)$. All transitions within a prompt are governed by a shared transition kernel $\mathsf{P} \in \mathbb{R}^{2 \times 2}$, sampled from the same distribution as the previous experiment. For each dimension $d \in \{1, 2, \ldots, 10\}$, we generate a batch of $B = 1000$ prompts. The LSA model is trained using the Adam optimizer for 1000 iterations with a learning rate of 0.01, minimizing the mean squared error between predicted and true labels. As a reference, we analytically compute the prediction from the reparameterized linear model using least squares regression. Accuracy is computed by rounding each predicted value to the nearest integer (0 or 1) and comparing it to the binary label. The experiment is repeated across 5 random seeds.

As shown in Fig. 5, the reparameterized model consistently outperforms the LSA model, with the accuracy gap grows as the dimension $d$ increases. Although both models attempt to fit the same underlying process, the hardness of parameter recovery suggests degraded performance at higher dimensions. In other words, the computational intractability has practical implications for expressivity and learning effectiveness in 1-layer LSA models.

## B.2 Capacity of Multi-Layer Transformers

To further investigate ICL for Markovian dynamics learning for practical models, we conduct empirical studies using GPT-2-based models to answer the following research questions[2].

*RQ1.* Does nonlinear attention in transformers functionally approximate classical models?
*RQ2.* How do attention patterns in transformers evolve across layers?

We adopt architectures based on GPT-2-blocks. We consider three configurations of (embedding dimension, number of transformer blocks, number of heads), inspired by [41]: (i) tiny: $(64, 3, 2)$, (ii) small: $(128, 6, 4)$, (iii) standard: $(256, 12, 8)$. The models are optimized by Adam over 50K epochs with learning rate 0.0001. For each epoch, we randomly generate 64 data samples to train the model parameters. To ensure high prediction performance given any length-$n'$ prompt ($n' \in [n]$), we train on the average of the error over different prompt lengths from 1 through $n$ and update $n$ from 26 to 101 during training. To generate data, we use the process described in Section B.1. We

---

[2]Our code is available at `https://github.com/dingyanna/ICL-Markov-Dynamics`

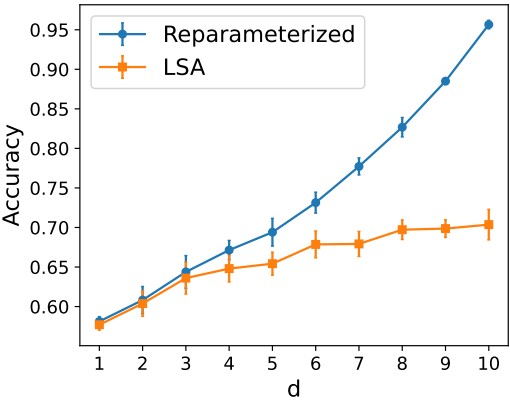

Figure 5: Accuracy versus dimension $d$ for the reparameterized model (blue) and the 1-layer LSA model (orange). Results show mean and standard deviation computed over 5 random seeds. The growing performance gap highlights the increasing difficulty for LSA to recover the optimal solution as the dimension increases.

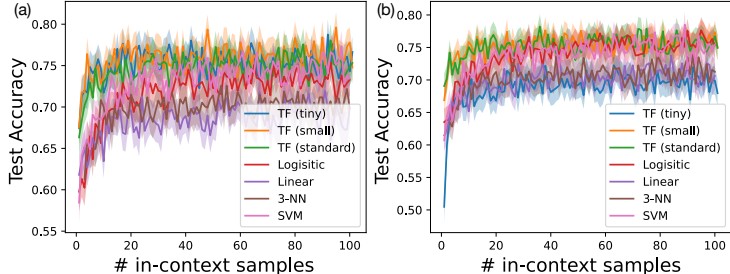

Figure 6: Testing accuracy for three model configurations, compared to baseline learning algorithms for (a) independent and (b) correlated initial conditions, respectively.

measure prediction accuracy by assigning the integer in $\{0, 1\}$ closest to the transformer's output as the predicted state. The experiments are conducted on an NVIDIA A40 GPU.

**Deeper transformers outperform classical models in predicting the next token for query sequence in-context (*RQ1*).** We investigate the performance of trained transformer compared to baseline learning algorithms, including logistic regression, linear regression, 3-Nearest Neighbors (3-NN), and Support Vector Machine (SVM), when the number of in-context samples vary from 1 to 100. Fig. 6 demonstrate the test accuracy for independent and correlated initial states. The accuracy is averaged over 1280 prompts, where the shaded region denotes 90% confidence intervals computed using 1000 bootstraps. The result implies that the trained transformers with small or standard size have comparable performance with SVM and logistic regression and better than the simple baseline 3-NN, while the test performance for tiny is slightly worse than its larger counterparts.

**Transformers capture similarities between in-context sequences (*RQ2*).** To investigate whether the attention mechanism captures structural similarities in in-context sequences with distinct transition kernels, we visualize the attention matrix and compare it to the sequence similarity matrix. We generate prompts by partitioning the in-context sequences into $k$ groups of equal size, each corresponding to a distinct transition kernel. Within each group, sequences are sampled from the same kernel, ensuring shared transition dynamics. To quantify sequence similarity, we compute a similarity score as: Similarity $= 1 - \frac{\text{Hamming distance}}{\text{Sequence length}}$. Fig. 7 presents the attention matrices across layers, illustrating a progressive emergence of structure that aligns with the sequence similarity matrix. In particular, by the third layer, sequences governed by the same transition kernel exhibit significantly stronger mutual attention, indicating that the model increasingly attends to structurally similar sequences. This

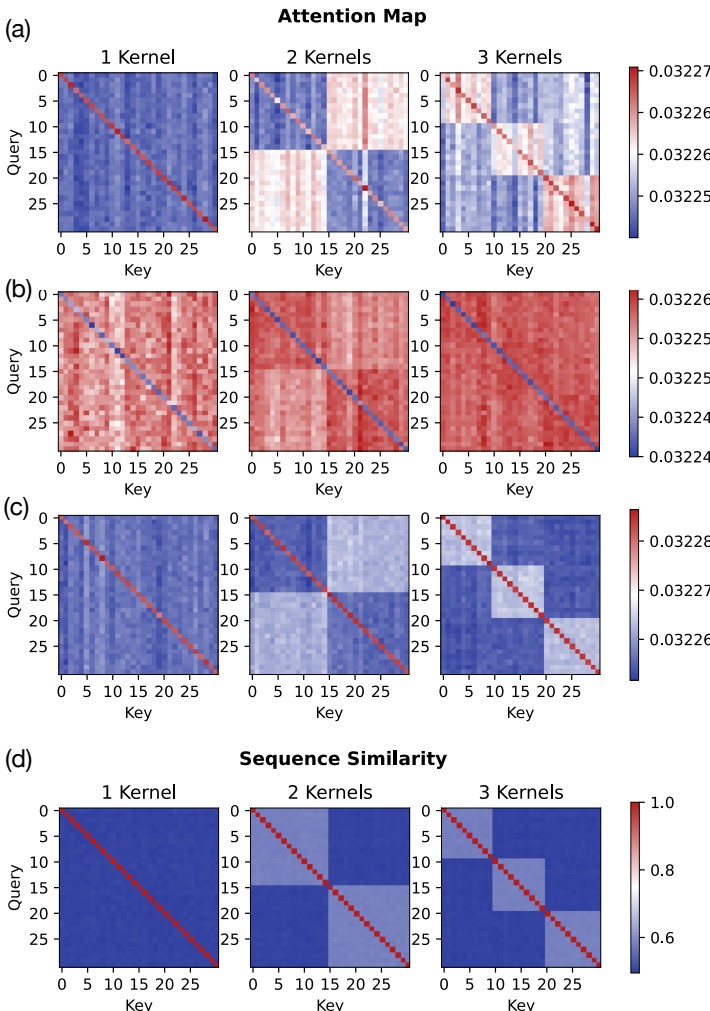

Figure 7: Attention patterns across 3 transformer layers and sequence similarity.(a)-(c) Attention maps from shallow to deeper layers. (d) Sequence similarity. In each subfigure, the three columns correspond to datasets with increasing numbers of transition kernels in the input prompt. The emergence of block-wise patterns becomes more pronounced in deeper layers.

layer-wise sharpening of attention suggests that transformers can progressively reflect the in-context Markovian dependencies.

In the following sections, we analyze the optimization landscape of the 1-layer LSA model for ICL over Markovian dynamics. We begin by characterizing the global minima of LSA through a reparameterized objective in low-dimensional settings (Sections C.1, C.2), deriving exact solutions for both i.i.d. and correlated initial-state cases. In Section C.3, we extend the analysis of the reparameterized objective to arbitrary-length sequences and general state spaces. Section C.4 then establishes a fundamental limitation that recovering transformer parameters from the global minimizer is NP-hard. Finally, Section D offers an interpretation of the forward computation in multi-layer LSA models as a multi-objective optimization process.

The theoretical results and their relationships are organized as follows:

1. In Section C.1, we assume that initial states are independently sampled.
   (a) Lemma C.1 derives the global minimizer of the reparameterized convex objective under this i.i.d. assumption.
   (b) Theorem C.2 (Theorem 3.3 restated) builds on Lemma C.1 to characterize the global minimizer of the original 1-layer LSA objective.

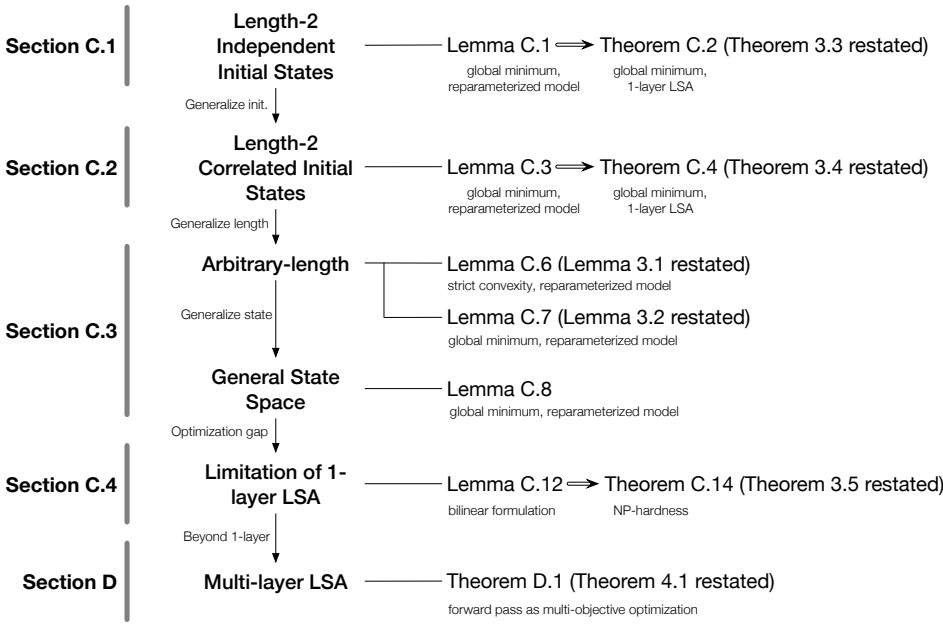

Figure 8: Structure of theoretical results in this section. Each row pair corresponds to a subsection, with the left showing the modeling case and the right listing the corresponding lemmas and theorems. Results build from short Markov chains with i.i.d. initial states to general, long sequences with general state space. This leads to a limitation result, showing that recovering transformer parameters from the global optimum is NP-hard in general. Finally, we provide a separate interpretation of the multi-layer LSA forward pass as a multi-objective optimization process.

2. In Section C.2, we extend the analysis to the setting where initial states may be correlated.
   (a) Lemma C.3 generalizes the convex minimization result to accommodate arbitrary initial-state distributions.
   (b) Theorem C.4 (Theorem 3.4 restated) then applies this generalization to characterize the global minimizer of the 1-layer LSA objective in the correlated setting.
3. In Section C.3, we consider Markov chains of arbitrary length and general (binary or nonbinary) state spaces.
   (a) Lemma C.6 (Lemma 3.1 restated) establishes that the reparameterized objective remains strictly convex in this general setting.
   (b) Lemma C.7 (Lemma 3.2 restated) and Lemma C.8 provide closed-form expressions for the global minimizer of the reparameterized objective for binary and nonbinary state spaces, respectively.
4. We then investigate whether such optimal solutions can be realized by any feasible choice of 1-layer transformer parameters in Section C.4.
   (a) Lemma C.12 recasts this recovery question as a bilinear feasibility problem.
   (b) Theorem C.14 (Theorem 3.5 restated) shows that this problem is NP-hard, revealing a fundamental limitation: 1-layer LSA may be unable to represent optimal solutions even when they exist in the reparameterized space.
5. Section D recasts the forward pass of a multi-layer LSA as a multi-objective optimization involving losses beyond squared error.

The logical flow of the main theoretical results is summarized in Figure 8.

## C  Proofs for Loss Landscape Characterization

### C.1  Global Optimum for Length-2 Markov Chains (i.i.d. Initialization; Theorem 3.3)

We begin by rewriting the loss by keeping parameters that affect the output prediction for the query $x_{n+1}$.

The input prompt is formatted as a $(d+1) \times (n+1)$ matrix:

$$Z_0 = \begin{bmatrix} x_1 & \cdots & x_n & x_{n+1} \\ y_1 & \cdots & y_n & 0 \end{bmatrix}.$$

We assume $x_i \overset{i.i.d.}{\sim} Bernoulli(p)$ and let $p_{ij}$ denote the transition probability from state $i$ to $j$ ($i, j \in \mathcal{X} = \{0, 1\}$). We define the label $y_i$ to be the next state. By definition of Markov chain, the expected value of $y_i$ given $x_i$ is

$$\mathbb{E}[y_i \mid x_i, p_{01}, p_{11}] = (1 - x_i)p_{01} + x_i p_{11} = p_{01} + (p_{11} - p_{01})x_i. \tag{12}$$

**Rewriting the objective function.** The in-context objective function for the single layer case is defined as:

$$f(P, Q) = \mathbb{E}_{\{x_i\}_{i=1}^{n+1}, p_{01}, p_{11}} \left[ \left( \left( Z_0 + \frac{1}{n} \mathrm{Attn}_{P,Q}(Z_0) \right)_{d+1,n+1} - y_{n+1} \right)^2 \right]. \tag{13}$$

By definition of attention (Eq. 4) (here $M = \begin{bmatrix} I_n & 0 \\ 0 & 0 \end{bmatrix} \in \mathbb{R}^{(n+1)\times(n+1)}$ is the mask matrix),

$$Z_0 + \frac{1}{n}\mathrm{Attn}_{P,Q}(Z_0) = Z_0 + \frac{1}{n}PZ_0M(Z_0^\top QZ_0) = Z_0 + \frac{1}{n}P(Z_0 M Z_0^\top)QZ_0$$

$$= Z_0 + \frac{1}{n}P\left( \begin{bmatrix} x_1 & \cdots & x^{(n)} & x_{n+1} \\ y_1 & \cdots & y_n & 0 \end{bmatrix} \begin{bmatrix} I_n & 0 \\ 0 & 0 \end{bmatrix} \begin{bmatrix} x_1 & y_1 \\ \vdots & \vdots \\ x_n & y_n \\ x_{n+1} & 0 \end{bmatrix} \right) QZ_0$$

$$= Z_0 + \frac{1}{n}P\left( \begin{bmatrix} x_1 & \cdots & x_n & 0 \\ y_1 & \cdots & y_n & 0 \end{bmatrix} \begin{bmatrix} x_1 & y_1 \\ \vdots & \vdots \\ x_n & y_n \\ x_{n+1} & 0 \end{bmatrix} \right) QZ_0$$

$$= Z_0 + P\left( \underbrace{\frac{1}{n}\sum_{i=1}^n \begin{bmatrix} x_i^2 & x_i y_i \\ x_i y_i & y_i^2 \end{bmatrix}}_{=:\mathsf{G}} \right) QZ_0.$$

The last column of the above matrix can be written as

$$\begin{bmatrix} x_{n+1} \\ 0 \end{bmatrix} + \frac{1}{n}P\mathsf{G}Q\begin{bmatrix} x_{n+1} \\ 0 \end{bmatrix}.$$

For the binary input case, $d = 1$ and $P, Q \in \mathbb{R}^{2\times 2}$. Let $b = [b_1; b_2]^\top$ ($b \in \mathbb{R}^2$) be the last row of $P$ and $a = [a_1; a_2] \in \mathbb{R}^2$ be the first column of $Q$. The bottom-right entry of $Z_0 + \frac{1}{n}\mathrm{Attn}_{P,Q}(Z_0)$ can be expressed as $b^\top \mathsf{G} a x_{n+1}$. Since $f(P, Q)$ only depends on parameters $b, a$, we rewrite the objective function as

$$f(P, Q) = \mathbb{E}_{\{x_i\}_{i=1}^{n+1}, p_{01}, p_{11}} \left[ \left( b^\top \mathsf{G} a x_{n+1} - y_{n+1} \right)^2 \right]. \tag{14}$$

**Reparameterization.** We further expand the term $b^\top \mathsf{G} a$ as

$$\begin{bmatrix} b_1 & b_2 \end{bmatrix} \left( \frac{1}{n}\sum_i \begin{bmatrix} x_i^2 & x_i y_i \\ x_i y_i & y_i^2 \end{bmatrix} \right) \begin{bmatrix} a_1 \\ a_2 \end{bmatrix}$$

$$= a_1 b_1 \frac{1}{n}\sum_{i=1}^n x_i^2 + (a_1 b_2 + a_2 b_1)\frac{1}{n}\sum_{i=1}^n x_i y_i + a_2 b_2 \frac{1}{n}\sum_{i=1}^n y_i^2.$$

Let $G_{xx}, G_{xy}, G_{yy}$ denote the top-left, top-right, and bottom-right entry, respectively. For any vector $X = [X_1; X_2; X_3]$ in $\mathbb{R}^3$, we consider the following loss function

$$\tilde{f}(X) = \mathbb{E}_{\{x_i\}_{i=1}^{n+1}, p_{01}, p_{11}} \left[ \left( (X_1 G_{xx} + X_2 G_{xy} + X_3 G_{yy}) x_{n+1} - y_{n+1} \right)^2 \right]. \tag{15}$$

We first derive the unique global minimum of the reparameterized loss function (Eq. 15) and then find the set of global minima for the original loss function (Eq. 13) over the space of $P, Q$.

**Lemma C.1.** *Consider the in-context learning of length-2 Markov chains $\{(x_i, y_i)\}_{i=1}^{n}$ ($x_i, y_i \in \{0, 1\}$) with transition probabilities $p_{01}, p_{11} \sim U(0, 1)$. Suppose the initial states $x_i$ are i.i.d. sampled from $Bernoulli(p)$ for some constant $p \in (0, 1)$. Consider the reparameterized objective*

$$\tilde{f}(X) = \mathbb{E}_{\{x_i, y_i\}_{i=1}^{n+1}, p_{01}, p_{11}} \left[ \left( (X_1 G_{xx} + X_2 G_{xy} + X_3 G_{yy}) x_{n+1} - y_{n+1} \right)^2 \right]. \tag{16}$$

*where $X = [X_1, X_2, X_3] \in \mathbb{R}^3$ and $y_i = (1 - x_{n+1}) p_{01} + x_{n+1} p_{11}$ denotes the conditional probability observing 1 at the next state given the current state.*

*(1) The objective function $\tilde{f}$ is strictly convex.*

*(2) The global minimum $X^*$ is given as $X^* = H^{-1} \begin{bmatrix} p^2/2 & p^2/3 & p^2/12 + p/4 \end{bmatrix}^\top$, where $H$ is a symmetric matrix defined as follows*

$$H := p \begin{bmatrix} \frac{p}{n} + \frac{n-1}{n} p^2 & \frac{p}{2n} + \frac{n-1}{2n} p^2 & \frac{p}{2} \\ & \frac{p}{2n} + \frac{(n-1)p^2}{3n} & \frac{p}{2n} + \frac{n-1}{n} \left( \frac{p}{4} + \frac{p^2}{12} \right) \\ & & \frac{1}{2n} + \frac{n-1}{n} \left( \frac{1}{3} - \frac{1}{6} p + \frac{1}{6} p^2 \right) \end{bmatrix} \tag{17}$$

*(omitting repeating entries in the lower half triangle).*

*Proof.* We defer the proof of *(1)* to Lemma 3.1. Since $\tilde{f}(X)$ is strictly convex, it has a unique global minimum that sets the gradient $\nabla \tilde{f}(X)$ to zero. To show *(2)*, we first set up the equation to evaluate the minimizer.

**Setting up equations to solve for minimizer.** The gradient of $\tilde{f}$ w.r.t. $X$ can be expressed as:

$$\nabla \tilde{f}(X) = 2 \begin{bmatrix} \mathbb{E} \left[ x_{n+1}^2 \left( G_{xx}^2 X_1 + G_{xy} G_{xx} X_2 + G_{yy} G_{xx} X_3 \right) - x_{n+1} y_{n+1} G_{xx} \right] \\ \mathbb{E} \left[ x_{n+1}^2 \left( G_{xx} G_{xy} X_1 + G_{xy}^2 X_2 + G_{yy} G_{xy} X_3 \right) - x_{n+1} y_{n+1} G_{xy} \right] \\ \mathbb{E} \left[ x_{n+1}^2 \left( G_{xx} G_{yy} X_1 + G_{xy} G_{yy} X_2 + G_{yy}^2 X_3 \right) - x_{n+1} y_{n+1} G_{yy} \right] \end{bmatrix}. \tag{18}$$

The global minimizer $X^*$ is the solution the following system:

$$\begin{bmatrix} \mathbb{E} \left[ x_{n+1}^2 G_{xx}^2 \right] & \mathbb{E} \left[ x_{n+1}^2 G_{xx} G_{xy} \right] & \mathbb{E} \left[ x_{n+1}^2 G_{xx} G_{yy} \right] \\ \mathbb{E} \left[ x_{n+1}^2 G_{xx} G_{xy} \right] & \mathbb{E} \left[ x_{n+1}^2 G_{xy}^2 \right] & \mathbb{E} \left[ x_{n+1}^2 G_{xy} G_{yy} \right] \\ \mathbb{E} \left[ x_{n+1}^2 G_{xx} G_{yy} \right] & \mathbb{E} \left[ x_{n+1}^2 G_{xy} G_{yy} \right] & \mathbb{E} \left[ x_{n+1}^2 G_{yy}^2 \right] \end{bmatrix} \begin{bmatrix} X_1^* \\ X_2^* \\ X_3^* \end{bmatrix} = \begin{bmatrix} \mathbb{E} \left[ x_{n+1} y_{n+1} G_{xx} \right] \\ \mathbb{E} \left[ x_{n+1} y_{n+1} G_{xy} \right] \\ \mathbb{E} \left[ x_{n+1} y_{n+1} G_{yy} \right] \end{bmatrix}. \tag{19}$$

Next, we compute the expected values in the linear system (19). The reason for each key derivation step is marked with labels such as (i), (ii), etc., and further explained at the end of the derivation.

**Computing RHS of Eq. 19.** We evaluate the three elements in RHS separately below.

1. For the first element, we have

$$\mathbb{E}_{\{x_i,y_i\}_{i=1}^{n+1},p_{01},p_{11}}\left[x_{n+1}y_{n+1}\mathsf{G}_{xx}\right]$$

$$=\mathbb{E}_{\{x_i,y_i\}_{i=1}^{n+1},p_{01},p_{11}}\left[x_{n+1}y_{n+1}\frac{1}{n}\left(\sum_{i=1}^n x_i^2\right)\right]$$

$$=\frac{1}{n}\sum_{i=1}^n\underbrace{\mathbb{E}_{\{x_i,y_i\}_{i=1}^{n+1},p_{01},p_{11}}\left[x_{n+1}y_{n+1}x_i^2\right]}_{\text{independent of }i}$$

$$=\mathbb{E}_{x_i,x_{n+1},y_{n+1}p_{01},p_{11}}\left[x_{n+1}y_{n+1}x_i^2\right]$$

$$=\mathbb{E}_{x_i,x_{n+1},p_{01},p_{11}}\left[\mathbb{E}_{y_{n+1}}\left[x_{n+1}y_{n+1}x_i^2\mid x_i,x_{n+1},p_{01},p_{11}\right]\right]$$

$$=\mathbb{E}_{x_i,x_{n+1},p_{01},p_{11}}\left[x_i^2\cdot x_{n+1}\mathbb{E}_{y_{n+1}}\left[y_{n+1}\mid x_i,x_{n+1},p_{01},p_{11}\right]\right]$$

$$\overset{(i)}{=}\mathbb{E}_{x_i,x_{n+1},p_{01},p_{11}}\left[x_i^2\cdot x_{n+1}\mathbb{E}_{y_{n+1}}\left[y_{n+1}\mid x_{n+1},p_{01},p_{11}\right]\right]$$

$$=\mathbb{E}_{x_i,x_{n+1},p_{01},p_{11}}\left[x_i^2\cdot(p_{01}x_{n+1}+(p_{11}-p_{01})x_{n+1}^2)\right]$$

$$\overset{(ii)}{=}\mathbb{E}_{x_i,x_{n+1},p_{01},p_{11}}\left[x_i^2\cdot p_{11}x_{n+1}\right]$$

$$\overset{(iii)}{=}\mathbb{E}_{p_{11}}\left[p_{11}\right]\cdot\mathbb{E}_{x_i}[x_i^2]\cdot\mathbb{E}_{x_{n+1}}[x_{n+1}]$$

$$\overset{(iv)}{=}\frac{1}{2}p^2.\tag{20}$$

2. Similarly, for the second element, we have

$$\mathbb{E}_{\{x_i,y_i\}_{i=1}^{n+1},p_{01},p_{11}}\left[x_{n+1}y_{n+1}\mathsf{G}_{xy}\right]$$

$$=\mathbb{E}_{x_i,y_i,x_{n+1},y_{n+1},p_{01},p_{11}}\left[x_iy_ix_{n+1}y_{n+1}\right]$$

$$\overset{(i)}{=}\mathbb{E}_{x_i,x_{n+1},p_{01},p_{11}}\left[x_i\mathbb{E}_{y_i}\left[y_i\mid x_i,p_{01},p_{11}\right]\cdot x_{n+1}\mathbb{E}_{y_{n+1}}\left[y_{n+1}\mid x_{n+1},p_{01},p_{11}\right]\right]$$

$$=\mathbb{E}_{x_i,x_{n+1},p_{01},p_{11}}\left[(p_{01}x_i+(p_{11}-p_{01})x_i^2)\cdot(p_{01}x_{n+1}+(p_{11}-p_{01})x_{n+1}^2)\right]$$

$$\overset{(ii)}{=}\mathbb{E}_{x_i,x_{n+1},p_{01},p_{11}}\left[p_{11}x_i\cdot p_{11}x_{n+1}\right]$$

$$\overset{(iii)}{=}\mathbb{E}_{p_{11}}\left[p_{11}^2\right]\cdot\mathbb{E}_{x_i}[x_i]\cdot\mathbb{E}_{x_{n+1}}[x_{n+1}]$$

$$\overset{(iv)}{=}\frac{1}{3}p^2.\tag{21}$$

3. The third element can be expanded as follows.

$$\mathbb{E}_{\{x_i,y_i\}_{i=1}^{n+1},p_{01},p_{11}}\left[x_{n+1}y_{n+1}\mathsf{G}_{yy}\right]$$

$$=\mathbb{E}_{x_i,y_i,x_{n+1},y_{n+1},p_{01},p_{11}}\left[x_{n+1}y_{n+1}y_i^2\right]$$

$$\overset{(i)}{=}\mathbb{E}_{x_i,x_{n+1},p_{01},p_{11}}\left[\mathbb{E}_{y_i}\left[y_i^2\mid x_i,p_{01},p_{11}\right]\cdot x_{n+1}\mathbb{E}_{y_{n+1}}\left[y_{n+1}\mid x_{n+1},p_{01},p_{11}\right]\right]$$

$$\overset{(ii)}{=}\mathbb{E}_{x_i,x_{n+1},p_{01},p_{11}}\left[\mathbb{E}_{y_i}\left[y_i\mid x_i,p_{01},p_{11}\right]\cdot(p_{11}x_{n+1})\right]$$

$$=\mathbb{E}_{x_i,x_{n+1},p_{01},p_{11}}\left[(p_{01}+(p_{11}-p_{01})x_i)\cdot(p_{11}x_{n+1})\right]$$

$$=\mathbb{E}_{x_i,x_{n+1},p_{01},p_{11}}\left[p_{01}p_{11}x_{n+1}+p_{11}^2x_ix_{n+1}-p_{01}p_{11}x_ix_{n+1}\right]$$

$$\overset{(iii)}{=}\mathbb{E}_{p_{01}}\left[p_{01}\right]\mathbb{E}_{p_{11}}\left[p_{11}\right]\mathbb{E}_{x_{n+1}}\left[x_{n+1}\right]$$

$$+\left(\mathbb{E}_{p_{11}}\left[p_{11}^2\right]-\mathbb{E}_{p_{01}}\left[p_{01}\right]\mathbb{E}_{p_{11}}\left[p_{11}\right]\right)\mathbb{E}_{x_i}\left[x_i\right]\mathbb{E}_{x_{n+1}}\left[x_{n+1}\right]$$

$$\overset{(iv)}{=}\frac{1}{4}p+\frac{1}{12}p^2.\tag{22}$$

**Computing LHS of Eq. 19.** We evaluate the expectation of the covariance of in-context examples: $\mathbb{E}[\mathsf{G}_{\cdot}^2]$.

1.

$$
\mathbb{E}_{\{x_i,y_i\}_{i=1}^{n+1},p_{01},p_{11}}\left[\mathsf{G}_{xx}^2\right]
$$

$$
=\mathbb{E}_{\{x_i,y_i\}_{i=1}^{n+1},p_{01},p_{11}}\left[\left(\frac{1}{n}\sum_{i=1}^n x_i^2\right)\left(\frac{1}{n}\sum_{i=1}^n x_i^2\right)\right]
$$

$$
=\frac{1}{n^2}\mathbb{E}_{\{x_i,y_i\}_{i=1}^n}\left[x_1{}^2 x_1{}^2+\cdots+x_1{}^2 x_n{}^2+\cdots+x_n{}^2 x_1{}^2+\cdots+x_n{}^2 x_n{}^2\right]
$$

$$
=\frac{1}{n^2}\left(n\mathbb{E}_{x_i}\left[x_i{}^4\right]+n(n-1)\mathbb{E}_{x_i,x_j}\underbrace{\left[x_i^2 x_j^2\right]}_{j\neq i}\right)
$$

$$
\stackrel{(i)}{=}\frac{1}{n^2}\left(np+n(n-1)\mathbb{E}_{x_i}\left[x_i^2\right]\mathbb{E}_{x_j}\left[x_j^2\right]\right)
$$

$$
\stackrel{(iv)}{=}\frac{1}{n^2}\left(np+n(n-1)p^2\right)
$$

$$
=\frac{p}{n}+\frac{n-1}{n}p^2. \tag{23}
$$

2.

$$
\mathbb{E}_{\{x_i,y_i\}_{i=1}^{n+1},p_{01},p_{11}}\left[\mathsf{G}_{xx}\mathsf{G}_{xy}\right]
$$

$$
=\mathbb{E}_{\{x_i\}_{i=1}^n,\{y_i\}_{i=1}^n,p_{01},p_{11}}\left[\left(\frac{1}{n}\sum_{i=1}^n x_i^2\right)\left(\frac{1}{n}\sum_{i=1}^n x_i y_i\right)\right]
$$

$$
\stackrel{(ii)}{=}\mathbb{E}_{\{x_i\}_{i=1}^n,p_{01},p_{11}}\left[\left(\frac{1}{n}\sum_{i=1}^n x_i\right)\mathbb{E}_{\{y_i\}_{i=1}^n}\left[\left(\frac{1}{n}\sum_{i=1}^n x_i y_i\right)\Big|\{x_i\}_{i=1}^n,p_{01},p_{11}\right]\right]
$$

$$
=\mathbb{E}_{\{x_i\}_{i=1}^n,p_{01},p_{11}}\left[\left(\frac{1}{n}\sum_{i=1}^n x_i\right)\left(\frac{1}{n}\sum_{i=1}^n x_i(p_{01}+(p_{11}-p_{01})x_i)\right)\right]
$$

$$
=\mathbb{E}_{\{x_i\}_{i=1}^n,p_{01},p_{11}}\left[p_{01}\left(\frac{1}{n}\sum_{i=1}^n x_i\right)\left(\frac{1}{n}\sum_{i=1}^n x_i\right)\right]
$$

$$
+\mathbb{E}_{\{x_i\}_{i=1}^n,p_{01},p_{11}}\left[(p_{11}-p_{01})\left(\frac{1}{n}\sum_{i=1}^n x_i\right)\left(\frac{1}{n}\sum_{i=1}^n x_i{}^2\right)\right]
$$

$$
\stackrel{(ii)}{=}\mathbb{E}_{\{x_i\}_{i=1}^n,p_{01},p_{11}}\left[p_{11}\left(\frac{1}{n}\sum_{i=1}^n x_i\right)\left(\frac{1}{n}\sum_{i=1}^n x_i\right)\right]
$$

$$
\stackrel{(iii,iv)}{=}\frac{1}{2}\mathbb{E}_{\{x_i,y_i\}_{i=1}^{n+1},p_{01},p_{11}}\left[\mathsf{G}_{xx}^2\right]
$$

$$
=\frac{p}{2n}+\frac{n-1}{2n}p^2. \tag{24}
$$

3.

$$\mathbb{E}_{\{x_i,y_i\}_{i=1}^{n+1},p_{01},p_{11}}\left[\mathsf{G}_{xx}\mathsf{G}_{yy}\right]$$

$$=\mathbb{E}_{\{x_i\}_{i=1}^{n},p_{01},p_{11}}\left[\mathbb{E}_{\{y_i\}_{i=1}^{n}}\left[\left(\frac{1}{n}\sum_{i=1}^{n}x_i^2\right)\left(\frac{1}{n}\sum_{i=1}^{n}y_i^2\right)\bigg|\{x_i\}_{i=1}^{n},p_{01},p_{11}\right]\right]$$

$$\overset{(ii)}{=}\mathbb{E}_{\{x_i\}_{i=1}^{n},p_{01},p_{11}}\left[\left(\frac{1}{n}\sum_{i=1}^{n}x_i^2\right)\mathbb{E}_{\{y_i\}_{i=1}^{n}}\left[\left(\frac{1}{n}\sum_{i=1}^{n}y_i\right)\bigg|\{x_i\}_{i=1}^{n},p_{01},p_{11}\right]\right]$$

$$=\mathbb{E}_{\{x_i\}_{i=1}^{n},p_{01},p_{11}}\left[\left(\frac{1}{n}\sum_{i=1}^{n}x_i\right)\left(\frac{1}{n}\sum_{i=1}^{n}(p_{01}+(p_{11}-p_{01})x_i)\right)\right]$$

$$=\mathbb{E}_{\{x_i\}_{i=1}^{n},p_{01},p_{11}}\left[p_{01}\left(\frac{1}{n}\sum_{i=1}^{n}x_i\right)+(p_{11}-p_{01})\left(\frac{1}{n}\sum_{i=1}^{n}x_i\right)^2\right]$$

$$\overset{(iii)}{=}\mathbb{E}_{p_{01}}\left[p_{01}\right]p+\mathbb{E}_{p_{01}}\left[(p_{11}-p_{01})\right]c$$

$$\overset{(iv)}{=}\frac{p}{2}. \tag{25}$$

4.

$$\mathbb{E}_{\{x_i,y_i\}_{i=1}^{n+1},p_{01},p_{11}}\left[\mathsf{G}_{xy}^2\right]$$

$$=\mathbb{E}_{\{x_i,y_i\}_{i=1}^{n},p_{01},p_{11}}\left[\left(\frac{1}{n}\sum_{i=1}^{n}x_iy_i\right)\left(\frac{1}{n}\sum_{i=1}^{n}x_iy_i\right)\right]$$

$$=\frac{1}{n^2}\mathbb{E}_{\{x_i,y_i\}_{i=1}^{n+1},p_{01},p_{11}}\left[\sum_{i=1}^{n}x_i^2y_i^2\right]+\frac{1}{n^2}\mathbb{E}_{\{x_i,y_i\}_{i=1}^{n+1},p_{01},p_{11}}\left[\sum_{i=1}^{n}\sum_{j=1,j\neq i}^{n}x_iy_ix_jy_j\right]$$

$$\overset{(ii)}{=}\frac{1}{n^2}\mathbb{E}_{\{x_i\}_{i=1}^{n},p_{01},p_{11}}\left[\sum_{i=1}^{n}p_{11}x_i\right]+\frac{1}{n^2}\mathbb{E}_{\{x_i\}_{i=1}^{n},p_{01},p_{11}}\left[\sum_{i=1}^{n}\sum_{j=1,j\neq i}^{n}p_{11}^2x_ix_j\right]$$

$$=\frac{p}{2n}+\frac{(n-1)p^2}{3n}. \tag{26}$$

5.

$$\mathbb{E}_{\{x_i,y_i\}_{i=1}^{n+1},p_{01},p_{11}}\left[\mathsf{G}_{xy}\mathsf{G}_{yy}\right]$$

$$=\mathbb{E}_{\{x_i,y_i\}_{i=1}^{n},p_{01},p_{11}}\left[\left(\frac{1}{n}\sum_{i=1}^{n}x_iy_i\right)\left(\frac{1}{n}\sum_{i=1}^{n}y_i^2\right)\right]$$

$$=\frac{1}{n^2}\mathbb{E}_{\{x_i,y_i\}_{i=1}^{n+1},p_{01},p_{11}}\left[\sum_{i=1}^{n}x_iy_i^3\right]+\frac{1}{n^2}\mathbb{E}_{\{x_i,y_i\}_{i=1}^{n+1},p_{01},p_{11}}\left[\sum_{i=1}^{n}\sum_{j=1,j\neq i}^{n}x_iy_iy_j^2\right]$$

$$\overset{(ii)}{=}\frac{1}{n^2}\mathbb{E}_{\{x_i\}_{i=1}^{n},p_{01},p_{11}}\left[\sum_{i=1}^{n}p_{11}x_i\right]$$

$$\quad+\frac{1}{n^2}\mathbb{E}_{\{x_i\}_{i=1}^{n},p_{01},p_{11}}\left[\sum_{i=1}^{n}\sum_{j=1,j\neq i}^{n}p_{11}x_i(p_{01}+(p_{11}-p_{01})x_j)\right]$$

$$\overset{(iv)}{=}\frac{p}{2n}+\frac{n-1}{n}\left(\frac{p}{4}+\frac{p^2}{12}\right). \tag{27}$$

6.

$$\mathbb{E}_{\{x_i,y_i\}_{i=1}^{n+1},p_{01},p_{11}}\left[\mathsf{G}_{yy}^2\right]$$

$$=\mathbb{E}_{\{x_i,y_i\}_{i=1}^{n},p_{01},p_{11}}\left[\left(\frac{1}{n}\sum_{i=1}^{n}y_i^2\right)\left(\frac{1}{n}\sum_{i=1}^{n}y_i^2\right)\right]$$

$$=\frac{1}{n^2}\mathbb{E}_{\{x_i,y_i\}_{i=1}^{n+1},p_{01},p_{11}}\left[\sum_{i=1}^{n}y_i^4\right]+\frac{1}{n^2}\mathbb{E}_{\{x_i,y_i\}_{i=1}^{n+1},p_{01},p_{11}}\left[\sum_{i=1}^{n}\sum_{j=1,j\neq i}^{n}y_i^2 y_j^2\right]$$

$$\stackrel{(ii)}{=}\frac{1}{n^2}\mathbb{E}_{\{x_i\}_{i=1}^{n},p_{01},p_{11}}\left[\sum_{i=1}^{n}p_{01}+(p_{11}-p_{01})x_i\right]$$

$$+\frac{1}{n^2}\mathbb{E}_{\{x_i\}_{i=1}^{n},p_{01},p_{11}}\left[\sum_{i=1}^{n}\sum_{j=1,j\neq i}^{n}(p_{01}+(p_{11}-p_{01})x_i)(p_{01}+(p_{11}-p_{01})x_j)\right]$$

$$\stackrel{(iv)}{=}\frac{1}{2n}+\frac{n-1}{n}\left(\frac{1}{3}-\frac{1}{6}p+\frac{1}{6}p^2\right). \tag{28}$$

Throughout the derivation, $(i)$ uses the fact that $\{x_j, y_j\}$ and $\{x_{j'}, y_{j'}\}$ $(j' \neq j)$ are conditionally independent given $p_{01}, p_{11}$; $(ii)$ holds since $x_i$, $y_i$ are binary random variables and $x_i^k = x_i$, $y_i^k = y_i$ for any integer $k$; $(iii)$ follows from the fact that $p_{01}, p_{11}$ and $x_j$ $(j \in [n+1])$ are jointly independent; $(iv)$ holds because the $k$th moments of uniform distribution $U(0,1)$ and Bernoulli distribution $Bernoulli(p)$ are $\frac{1}{k+1}$ and $p$, respectively.

Since $x_{n+1}$ and $x_i$ $(i \in [n])$ are independent, we have $\mathbb{E}[x_{n+1}^2 \mathsf{G}_\cdot^2] = \mathbb{E}[x_{n+1}^2]\mathbb{E}[\mathsf{G}_\cdot^2] = p\mathbb{E}[\mathsf{G}_\cdot^2]$. Hence we have the expression for $H$.

Since $\tilde{f}(X)$ is strictly convex, Eq. 19 has a unique solution $X^* = H^{-1}\begin{bmatrix} p^2 & p^2/3 & p^2/12 + p/4 \end{bmatrix}$.
□

**Theorem C.2** (**Theorem 3.3 restated**). *Consider the in-context learning of length-2 Markov chains $\{(x_i, y_i)\}_{i=1}^{n}$ $(x_i, y_i \in \{0,1\})$ with transition probabilities $p_{01}, p_{11} \sim U(0,1)$. Suppose the initial states $x_i$ are i.i.d. sampled from $Bernoulli(p)$ for some constant $p \in (0,1)$.*

*Let $X^* := H^{-1}\begin{bmatrix} p^2/2 & p^2/3 & p^2/12 + p/4 \end{bmatrix}^\top$, where $H$ is a symmetric matrix defined as follows*

$$H := p\begin{bmatrix} \frac{p}{n}+\frac{(n-1)p^2}{n} & \frac{p}{2n}+\frac{(n-1)p^2}{2n} & \frac{p}{2} \\ & \frac{p}{2n}+\frac{(n-1)p^2}{3n} & \frac{p}{2n}+\frac{n-1}{n}\left(\frac{p}{4}+\frac{p^2}{12}\right) \\ & & \frac{1}{2n}+\frac{n-1}{n}\left(\frac{1}{3}-\frac{p}{6}+\frac{p^2}{6}\right) \end{bmatrix}. \tag{29}$$

*Then the following choice of parameters*

$$P = \begin{bmatrix} 0 & 0 \\ 1 & \frac{X_2^* \pm \sqrt{X_2^{*\,2}-4X_1^* X_3^*}}{2} \end{bmatrix} \quad Q = \begin{bmatrix} X_1^* & 0 \\ X_2^* - \frac{X_1^* X_2^* \pm X_1^* \sqrt{X_2^{*\,2}-4X_1^* X_3^*}}{2} & 0 \end{bmatrix} \tag{30}$$

*is a global minimizer of $f(P, Q)$.*

### C.2 Global Optimum for Length-2 Markov Chains (Correlated Initialization; Theorem 3.3)

**Lemma C.3.** *Consider the in-context learning of length-2 Markov chains $\{(x_i, y_i)\}_{i=1}^{n}$ $(x_i, y_i \in \{0,1\})$ with transition probabilities $p_{01}, p_{11} \sim U(0,1)$. Suppose the initial states $x_i$ are sampled from $Bernoulli(p)$ for some constant $p \in (0,1)$. Let $c_1 = \sum_{i=1}^{n}\mathbb{E}[x_i x_{n+1}], c_2 = \sum_{i=1}^{n}\sum_{j=1,j\neq i}^{n}\mathbb{E}[x_i x_j x_{n+1}]$.*

*Consider the reparameterized objective*

$$\tilde{f}(X) = \mathbb{E}_{\{x_i,y_i\}_{i=1}^{n+1},p_{01},p_{11}}\left[\left((X_1\mathsf{G}_{xx}+X_2\mathsf{G}_{xy}+X_3\mathsf{G}_{yy})x_{n+1}-y_{n+1}\right)^2\right]. \tag{31}$$

where $X = [X_1, X_2, X_3] \in \mathbb{R}^3$ and $y_i = (1 - x_{n+1})p_{01} + x_{n+1}p_{11}$ denotes the conditional probability observing 1 at the next state given the current state.

*Then a global minimum is given as*

$$X^* = H^{-1} \begin{bmatrix} c_1/2n \\ c_1/3n \\ p/4 + c_1/12n \end{bmatrix},$$ (32)

*where*

$$H = \begin{bmatrix} \frac{c_1}{n^2} + \frac{c_2}{n^2} & \frac{c_1}{2n^2} + \frac{c_2}{2n^2} & \frac{c_1}{2n} \\ & \frac{c_1}{2n^2} + \frac{c_2}{3n^2} & \frac{(n+1)c_1}{4n^2} + \frac{c_2}{12n^2} \\ & & \frac{(2n+1)p}{6n} - \frac{(n-1)c_1}{6n^2} + \frac{c_2}{6n^2} \end{bmatrix}$$ (33)

*(omitting repeating entries in the lower half triangle).*

*Proof.* Since the objective function remains the same, the derivation for the equations follows from the independent in-context example case (Eq. 19). Similarly, we label the key steps with (i), (ii), etc., and defer the explanation at the end of the derivation.

**Computing RHS of Eq. 19 w/o assuming independence of $\{x_i\}_{i\in[n+1]}$.**

1. For the first element, we have

$$\mathbb{E}_{\{x_i,y_i\}_{i=1}^{n+1},p_{01},p_{11}} \left[ x_{n+1}y_{n+1}\mathsf{G}_{xx} \right]$$

$$\overset{(i)}{=} \frac{1}{n} \sum_{i=1}^{n} \mathbb{E}_{x_i,x_{n+1},p_{01},p_{11}} \left[ x_{n+1}x_i \mathbb{E}_{y_{n+1}} \left[ y_{n+1} \mid x_{n+1}, p_{01}, p_{11} \right] \right]$$

$$\overset{(i)}{=} \frac{1}{n} \sum_{i=1}^{n} \mathbb{E}_{x_i,x_{n+1},p_{11}} \left[ p_{11}x_ix_{n+1} \right]$$

$$\overset{(iii)}{=} \frac{1}{n} \sum_{i=1}^{n} \mathbb{E}_{p_{11}} \left[ p_{11} \right] \mathbb{E}_{x_i,x_{n+1}} \left[ x_ix_{n+1} \right]$$

$$= \frac{1}{2n}c_1.$$ (34)

2. Similarly, for the second element, we have

$$\mathbb{E}_{\{x_i,y_i\}_{i=1}^{n+1},p_{01},p_{11}} \left[ x_{n+1}y_{n+1}\mathsf{G}_{xy} \right]$$

$$\overset{(ii)}{=} \frac{1}{n} \sum_{i=1}^{n} \mathbb{E}_{x_i,x_{n+1},p_{01},p_{11}} \left[ x_{n+1}\mathbb{E}_{y_{n+1}} \left[ y_{n+1} \mid x_{n+1}, p_{01}, p_{11} \right] \cdot x_i \mathbb{E}_{y_i} \left[ y_i \mid x_i, p_{01}, p_{11} \right] \right]$$

$$\overset{(i)}{=} \frac{1}{n} \sum_{i=1}^{n} \mathbb{E}_{x_i,x_{n+1},p_{01},p_{11}} \left[ (x_{n+1}p_{11})(x_ip_{11}) \right]$$

$$\overset{(iii)}{=} \frac{1}{3n} \sum_{i=1}^{n} \mathbb{E}[x_ix_{n+1}] = \frac{1}{3n}c_1.$$ (35)

3. The third element can be expanded as follows.

$$\mathbb{E}_{\{x_i,y_i\}_{i=1}^{n+1},p_{01},p_{11}} \left[ x_{n+1}y_{n+1}\mathsf{G}_{xy} \right]$$

$$\overset{(i)}{=} \frac{1}{n} \sum_{i=1}^{n} \mathbb{E}_{x_i,y_i,x_{n+1},y_{n+1},p_{01},p_{11}} \left[ x_{n+1}y_{n+1}y_i \right]$$

$$\overset{(ii)}{=} \frac{1}{n} \sum_{i=1}^{n} \mathbb{E}_{x_i,x_{n+1},p_{01},p_{11}} \left[ x_{n+1}\mathbb{E}_{y_{n+1}} \left[ y_{n+1} \mid x_{n+1}, p_{01}, p_{11} \right] \cdot \mathbb{E}_{y_i} \left[ y_i \mid x_i, p_{01}, p_{11} \right] \right]$$

$$\overset{(i)}{=} \frac{1}{n} \sum_{i=1}^{n} \mathbb{E}_{x_i,x_{n+1},p_{01},p_{11}} \left[ (x_{n+1}p_{11})(p_{01} + (p_{11} - p_{01})x_i)) \right]$$

$$= \frac{1}{4}p + \frac{1}{12n}c_1.$$ (36)

The derivation holds due the following facts: $(i)$ the states are binary, $(ii)$ $y_j$ and $y_i$ are conditionally independent for $j \neq i$, $(iii)$ independence between $x_i$ and $p_{01}, p_{11}$.

**Computing LHS of Eq. 19 w/o assuming independence of $\{x_i\}_{i \in [n+1]}$.** We directly present the results for the other terms, as their derivation is similar to that of the RHS in the independent case.

$$\mathbb{E}\left[x_{n+1}^2 \mathsf{G}_{xx}^2\right] = \frac{1}{n^2} \sum_{i=1}^{n} \mathbb{E}[x_i x_{n+1}] + \frac{1}{n^2} \sum_{i=1}^{n} \sum_{\substack{j \neq i \\ j=1}}^{n} \mathbb{E}[x_i x_j x_{n+1}]$$

$$= \frac{1}{n^2} c_1 + \frac{1}{n^2} c_2, \tag{37}$$

$$\mathbb{E}\left[x_{n+1}^2 \mathsf{G}_{xx} \mathsf{G}_{xy}\right] = \frac{1}{2n^2} \sum_{i=1}^{n} \mathbb{E}[x_i x_{n+1}] + \frac{1}{2n^2} \sum_{i=1}^{n} \sum_{\substack{j \neq i \\ j=1}}^{n} \mathbb{E}[x_i x_j x_{n+1}]$$

$$= \frac{1}{2n^2} c_1 + \frac{1}{2n^2} c_2, \tag{38}$$

$$\mathbb{E}\left[x_{n+1}^2 \mathsf{G}_{xx} \mathsf{G}_{yy}\right] = \frac{1}{2n} \sum_{i=1}^{n} \mathbb{E}[x_i x_{n+1}]$$

$$= \frac{1}{2n} c_1, \tag{39}$$

$$\mathbb{E}\left[x_{n+1}^2 \mathsf{G}_{xy}^2\right] = \frac{1}{2n^2} \sum_{i=1}^{n} \mathbb{E}[x_i x_{n+1}] + \frac{1}{3n^2} \sum_{i=1}^{n} \sum_{\substack{j \neq i \\ j=1}}^{n} \mathbb{E}[x_i x_j x_{n+1}]$$

$$= \frac{1}{2n^2} c_1 + \frac{1}{3n^2} c_2, \tag{40}$$

$$\mathbb{E}\left[x_{n+1}^2 \mathsf{G}_{xy} \mathsf{G}_{yy}\right] = \frac{1}{2n^2} \sum_{i=1}^{n} \mathbb{E}[x_i x_{n+1}] + \frac{1}{n^2} \sum_{i=1}^{n} \sum_{\substack{j \neq i \\ j=1}}^{n} \frac{1}{4} \mathbb{E}[x_i x_{n+1}] + \frac{1}{12} \mathbb{E}[x_i x_j x_{n+1}]$$

$$= \frac{n+1}{4n^2} c_1 + \frac{1}{12n^2} c_2, \tag{41}$$

$$\mathbb{E}\left[x_{n+1}^2 \mathsf{G}_{yy}^2\right] = \frac{p}{2n} + \frac{(n-1)p}{3n} + \frac{1}{n^2} \sum_{i=1}^{n} \sum_{\substack{j \neq i \\ j=1}}^{n}$$

$$- \frac{1}{12} \mathbb{E}[x_i x_{n+1}] - \frac{1}{12} \mathbb{E}[x_j x_{n+1}] + \frac{1}{6} \mathbb{E}[x_i x_j x_{n+1}]$$

$$= \frac{(2n+1)p}{6n} - \frac{n-1}{6n^2} c_1 + \frac{1}{6n^2} c_2. \tag{42}$$

$\square$

**Theorem C.4** (***Theorem 3.4 restated***). *Consider the in-context learning of length-2 Markov chains $\{(x_i, y_i)\}_{i=1}^{n}$ ($x_i, y_i \in \{0,1\}$) with transition probabilities $p_{01}, p_{11} \sim U(0,1)$. Suppose the initial states $x_i$ are sampled from $Bernoulli(p)$ for some constant $p \in (0,1)$. Let $c_1 = \sum_{i=1}^{n} \mathbb{E}[x_i x_{n+1}], c_2 = \sum_{i=1}^{n} \sum_{j=1, j \neq i}^{n} \mathbb{E}[x_i x_j x_{n+1}]$.*

*We define $X^*$ as $X^* := H^{-1} [c_1/2n \quad c_1/3n \quad p/4 + c_1/12n]$, where $H$ is a symmetric matrix defined as follows*

$$H := \begin{bmatrix} \frac{c_1}{n^2} + \frac{c_2}{n^2} & \frac{c_1}{2n^2} + \frac{c_2}{2n^2} & \frac{c_1}{2n} \\ & \frac{c_1}{2n^2} + \frac{c_2}{3n^2} & \frac{(n+1)c_1}{4n^2} + \frac{c_2}{12n^2} \\ & & \frac{(2n+1)p}{6n} - \frac{(n-1)c_1}{6n^2} + \frac{c_2}{6n^2} \end{bmatrix} \tag{43}$$

*(repeating entries in the lower half triangle are omitted).*

*Then by substituting $X^*$ into Eq. 10 gives a global minimizer of $f(P, Q)$.*

*Example* C.5. Suppose $x_{n+1} \sim Bernoulli(p)$ and $x_i \mid x_{n+1} \sim Bernoulli(g(x_{n+1}))$ for some function $g : \{0, 1\} \to [0, 1]$. For example, when $g(x) = (x - p)^2$, the expected values can be computed as follows.

For $i \in [n]$, $j = n + 1$,

$$
\begin{aligned}
\mathbb{E}[x_i x_{n+1}] &= \mathbb{E}_{x_{n+1}}\left[x_{n+1}\mathbb{E}_{x_i}[x_i|x_{n+1}]\right] \\
&= \mathbb{E}_{x_{n+1}}\left[x_{n+1}^3 - 2px_{n+1}^2 + p^2 x_{n+1}\right] \\
&= p - 2p^2 + p^3.
\end{aligned}
$$

Therefore $c_1 = n(p - 2p^2 + p^3)$.

$$
\begin{aligned}
\frac{c_2}{n(n-1)} &= \mathbb{E}[x_i x_j x_{n+1}] \\
&\overset{(i)}{=} \mathbb{E}_{x_{n+1}}\left[x_{n+1}\mathbb{E}_{x_i}[x_i|x_{n+1}]\mathbb{E}_{x_j}[x_j|x_{n+1}]\right] \\
&= \mathbb{E}_{x_{n+1}}\left[x_{n+1}(x_{n+1} - p)^2(x_{n+1} - p)^2\right] \\
&= \mathbb{E}_{x_{n+1}}\left[x_{n+1}(x_{n+1}^2 - 2px_{n+1} + p^2)(x_{n+1}^2 - 2px_{n+1} + p^2)\right] \\
&\overset{(ii)}{=} \mathbb{E}_{x_{n+1}}\left[x_{n+1}((1 - 2p)x_{n+1} + p^2)^2\right] \\
&= \mathbb{E}_{x_{n+1}}\left[(1 - 4p + 4p^2)x_{n+1}^3 + 2(1 - 2p)p^2 x_{n+1}^2 + p^4 x_{n+1}\right] \\
&= p - 4p^2 + 4p^3 + 2p^3 - 4p^4 + p^5 \\
&= p^5 - 4p^4 + 6p^3 - 4p^2 + p.
\end{aligned}
$$

The above derivation holds because $(i)$ $x_i, x_j$ are conditionally ind. given $x_{n+1}$, and $(ii)$ the states of Markov chain are binary.

## C.3   Reparameterized Optimum in the General Case (Lemma 3.2)

We recall $(x_i, y_i)$ form a binary Markov chain of length $d + 1$. Assuming the initial states are sampled from $Bernoulli(p)$, the probability of $x_{i,1}$ being 1 is $p$. For $1 < j \leq d$, the probability of $x_{i,j}$ being 1, given $x_{i,j-1}$, is $p_{11}x_{i,j-1} + (1 - x_{i,j-1})p_{01}$. The probability of $y_i$ being 1, given $x_{i,d}$, is $p_{11}x_{i,d} + (1 - x_{i,d})p_{01}$.

**Reparameterization.**   For general $d \geq 1$, the projection matrix $P$ and attention weight matrix $Q$ are of size $(d + 1) \times (d + 1)$. We write

$$
P = \begin{bmatrix} 0_{d \times (d+1)} \\ b^\top \end{bmatrix} \quad Q = \begin{bmatrix} A & 0_{d+1} \end{bmatrix}, \tag{44}
$$

where $b^\top \in \mathbb{R}^{1 \times (d+1)}$ denote the last row of $P$ and $A \in \mathbb{R}^{(d+1) \times d}$ ($j \in [d]$) represent the first $d$ columns of $Q$. The objective function can be rewritten as:

$$
f(P, Q) = \mathbb{E}_{\{x_i, y_i\}_{i=1}^{n+1}, p_{01}, p_{11}}\left[\left(\sum_{j=1}^{d} b^\top \mathsf{G} A_{\cdot,j} x_{n+1,j} - y_{n+1}\right)^2\right], \tag{45}
$$

where $x_{n+1,j}$ ($j \in [d]$) denotes the $j$th element of $x_{n+1}$ and $A_{\cdot,j}$ denote the $j$th column of $A$. The $i$-$j$ entry of $\mathsf{G}$ ($\mathsf{G}_{i,j}$) has the following expression:

$$
\mathsf{G}_{i,j} = \begin{cases} 1/n \sum_{k=1}^{n} x_{k,i}x_{k,j} & \text{if } i, j \in [d] \\ 1/n \sum_{k=1}^{n} x_{k,j}y_k & \text{if } i \in [d], j = d + 1 \text{ or } i = d + 1, j \in [d] \\ 1/n \sum_{k=1}^{n} y_k^2 & \text{if } i, j = d + 1 \end{cases}. \tag{46}
$$

Since $\mathsf{G}$ is symmetric, to obtain an objective function with a unique global minimum, we collect model parameters that share the same coefficients $\mathsf{G}_{i,j} = \mathsf{G}_{j,i}$. We introduce a reparameterization $\phi$ which maps from the model parameter space to $\mathbb{R}^{dm}$, where $m = \frac{(d+2)(d+1)}{2}$:

$$
\phi(P, Q)_r = X_r = \begin{cases} A_{i,j}b_{j'} + A_{j',j}b_{i'} & \text{for } i' \in [d + 1], j' > i' \\ A_{i',j}b_{j'} & \text{for } i' \in [d + 1], j' = i' \end{cases}. \tag{47}
$$

Here $\phi(\cdot)_r$ is the $r$th entry of the resulting vector, with $r = (j-1)m + i'(d+1) + j'$ and $A_{i,j}$ denotes the $(i,j)$-th entry of $A$ and $b_i$ denotes the $i$th element of $b$.

To simplify notation, we collapse the unique elements in $\mathsf{G}$ into a vector:

$$\mathsf{g} = [\mathsf{G}_{1,1} \quad \mathsf{G}_{1,2} \quad \cdots \quad \mathsf{G}_{1,d+1} \quad \mathsf{G}_{2,2} \quad \cdots \quad \mathsf{G}_{d,d} \quad \mathsf{G}_{d,d+1} \quad \mathsf{G}_{d+1,d+1}]^\top . \tag{48}$$

We concatenate the parameters $X^{(j)}$ ($j \in [d]$) into a vector $X = [X^{(1)}; \ldots; X^{(d)}] \in \mathbb{R}^{dm}$ and consider the following reparameterized objective function

$$\tilde{f}(X) = \mathbb{E}_{\{x_i,y_i\}_{i=1}^{n+1}, p_{01}, p_{11}} \left[ \left( (x_{n+1} \otimes \mathsf{g})^\top X - y_{n+1} \right)^2 \right] . \tag{49}$$

Building on the formulation of the reparameterized objective for arbitrary-length Markov chains, we next show that this objective is strictly convex.

**Lemma C.6 (** *Lemma 3.1 restated*). *Suppose the initial probability of the Markov chains is $\pi_1 = [1 - p, p]$ with $p \in (0,1)$ and the transition probabilities are sampled from $U(0,1)$. The reparameterized objective function Eq. 49 is strictly convex w.r.t. $X \in \mathbb{R}^{dm}$.*

*Proof.* We show the Hessian of $\tilde{f}$ w.r.t. $X$, $\mathbb{E}[x_{n+1}x_{n+1}^\top \otimes \mathsf{g}\mathsf{g}^\top]$, is positive definite. Let $w \neq 0$ be an arbitrary nontrivial vector in $\mathbb{R}^{dm}$. Let $z := x_{n+1} \otimes \mathsf{g}$. Then for any $x_{n+1} \in \{0,1\}^d$ and $\mathsf{g} \in [0,1]^m$, $w^\top \mathbb{E}[x_{n+1}x_{n+1}^\top \otimes \mathsf{g}\mathsf{g}^\top]w = w^\top \mathbb{E}[(x_{n+1} \otimes \mathsf{g})(x_{n+1} \otimes \mathsf{g})^\top]w = w^\top z z^\top w = |w^\top z|^2 \geq 0$. Since $w \neq 0$, at least one of its entry is nonzero and this entry is multiplied by one of $\{x_{n+1,j}\mathsf{G}_{i',j'} : j \in [d], i', j' \in [d+1]\}$ in the expression $w^\top z$. Take $j = \alpha, i' = \beta, j' = \gamma$. Then it suffices to find specific $\{x_i, y_i\}_{i=1}^n$ and $x_{n+1}$ s.t. $x_{n+1}[\alpha]\mathsf{G}_{\beta,\gamma} > 0$ with positive probability, i.e., $\mathbb{P}[x_{n+1,\alpha}\mathsf{G}_{\beta,\gamma}] > 0$. Since the initial probability $p \in (0,1)$ and the transition probabilities $p_{ij}$ are nonzero, by definition of Markov chains, $\mathbb{P}[x_{n+1,\alpha}\mathsf{G}_{\beta,\gamma}]$ is the product of $p$ (or $1-p$) and $p_{ij}$s and therefore is nonzero. Now because $w^\top(zz^\top)w \geq 0$ for all $z$ in its support and there exists at least one $z \in \mathbb{R}^{dm}$ s.t. $w^\top(zz^\top)w > 0$ and $\mathbb{P}[z] > 0$, we have $w^\top \mathbb{E}[zz^\top]w > 0$. Hence the matrix $\mathbb{E}[x_{n+1}x_{n+1}^\top \otimes \mathsf{g}\mathsf{g}]$ is positive definite and it follows that $\tilde{f}$ is strictly convex. $\square$

Leveraging the strict convexity of the objective, we proceed to derive its global minimizer in the case of binary Markov chains with arbitrary length.

**Lemma C.7 (***Lemma 3.2 restated***).** *Consider the in-context learning of length-$d+1$ ($d \geq 1$) Markov chains $\{(x_i, y_i)\}_{i=1}^n$ ($x_i, y_i \in \{0,1\}$) with transition kernel $\mathsf{P} = \begin{bmatrix} p_{00} & p_{01} \\ p_{10} & p_{11} \end{bmatrix} \in (0,1)^2$. Suppose the initial states $x_i$ are i.i.d. sampled from $Bernoulli(p)$ for some constant $p \in (0,1)$. Consider indices $i, j \in [d], i', j', k', l' \in [d+1]$ with $i' \leq j', k' \leq l'$. We denote $t_1 \leq t_2 \leq t_3 \leq t_4$ as the sorted version of $(i', j', k', l')$. Define $H \in \mathbb{R}^{dm \times dm}$ as*

$$H_{r,c} = \frac{1}{n}\mathbb{E}\left[ \left( p(\mathsf{P}^{t_1-1})_{11} + (1-p)(\mathsf{P}^{t_1-1})_{01} \right)(\mathsf{P}^{t_2-t_1})_{11}(\mathsf{P}^{t_3-t_2})_{11}(\mathsf{P}^{t_4-t_3})_{11} \right]$$

$$+ \frac{n-1}{n}\mathbb{E}\left[ \left( p(\mathsf{P}^{i'-1})_{11} + (1-p)(\mathsf{P}^{i'-1})_{01} \right)(\mathsf{P}^{j'-i'})_{11} \right.$$

$$\left. \left( p(\mathsf{P}^{k'-1})_{11} + (1-p)(\mathsf{P}^{k'-1})_{01} \right)(\mathsf{P}^{l'-k'})_{11} \right], \tag{50}$$

*where $r = (i-1)m + j' + \sum_{\tau=0}^{i'-2} d + 1 - \tau$, $c = (j-1)m + l' + \sum_{\tau=0}^{k'-2} d + 1 - \tau$.*

*Define $b \in \mathbb{R}^{dm}$ as*

$$b_{(j-1)m+j'+\sum_{\tau=0}^{i'-2} d+1-\tau} = \mathbb{E}\left[ \left( p(\mathsf{P}^{j-1})_{11} + (1-p)(\mathsf{P}^{j-1})_{01} \right)(\mathsf{P}^{d+1-j})_{11} \right.$$

$$\left. \left( p(\mathsf{P}^{i'-1})_{11} + (1-p)(\mathsf{P}^{i'-1})_{01} \right)(\mathsf{P}^{j'-i'})_{11} \right]. \tag{51}$$

*The global minimum $X^* \in \mathbb{R}^{dm}$ of the objective function described in Eq. 49 equals $X^* = H^{-1}b$.*

*Proof.* Setting the gradient of Eq. 49 w.r.t. $X$ to zero, we have

$$\mathbb{E}\left[ x_{n+1}x_{n+1}^\top \otimes \mathsf{g}\mathsf{g}^\top \right]X = \mathbb{E}\left[ y_{n+1}(x_{n+1} \otimes \mathsf{g}) \right] \tag{52}$$

where $\otimes$ denotes the Kronecker product.

**Evaluating LHS of Eq. 52:** $\mathbb{E}\left[x_{n+1,i}x_{n+1,j}\mathsf{G}_{i',j'}\mathsf{G}_{k',l'}\right]$ **with** $i,j \in [d]$, $i \leq j$, $i',j',k',l' \in [d+1]$, **and** $i' \leq k', j' \leq l'$. Let $\mathsf{P} = \begin{bmatrix} p_{00} & p_{01} \\ p_{10} & p_{11} \end{bmatrix}$ denote the transition probability matrix and $\pi = [1-p, p]$ the initial marginal probability. Further, $(\mathsf{P}^k)_{ij}$ $(i,j \in \{0,1\})$ denotes the specific entry of $\mathsf{P}$ raised to the power of $k$. Then

$$\mathbb{E}\left[x_{n+1,i}x_{n+1,j}\mathsf{G}_{i',j'}\mathsf{G}_{k',l'}\right] = \mathbb{E}\left[x_{n+1,i}x_{n+1,j}\right]\mathbb{E}\left[\mathsf{G}_{i',j'}\mathsf{G}_{k',l'}\right], \tag{53}$$

due to the fact that $x_i$ $(i \in [n])$ and $x_{n+1}$ are independent and $\mathsf{G}$ contains in-context samples only. We then evaluate the two terms $\mathbb{E}\left[x_{n+1,i}x_{n+1,j}\right]$, $\mathbb{E}\left[\mathsf{G}_{i',j'}\mathsf{G}_{k',l'}\right]$ separately.

- For $i \leq j$, the probability of $x_{n+1,i} = x_{n+1,j} = 1$ is equivalent to that of $x_{n+1,i} = 1$ conditioned on $x_{n+1,1}$ multiplied by the probability of $x_{n+1,j} = 1$ conditioned on $x_{n+1,i} = 1$. Therefore,

$$
\begin{aligned}
&\mathbb{E}\left[x_{n+1,i}x_{n+1,j}\right] \\
={}&\mathbb{E}\left[\mathbb{P}[x_{n+1,i} = x_{n+1,j} = 1]\right] \\
={}&\mathbb{E}\left[\left(p(\mathsf{P}^{i-1})_{11} + (1-p)(\mathsf{P}^{i-1})_{01}\right)(\mathsf{P}^{j-i})_{11}\right].
\end{aligned} \tag{54}
$$

- We temporarily let $x_{k,d+1} = y_k$ for $k \in [n]$.
  For $i',j',k',l' \in [d+1]$ and $i' \leq j', k' \leq l'$, we have

$$
\begin{aligned}
&\mathbb{E}\left[\mathsf{G}_{i',j'}\mathsf{G}_{k',l'}\right] \\
={}&\mathbb{E}\left[\left(\frac{1}{n}\sum_{k=1}^{n}x_{k,i'}x_{k,j'}\right)\left(\frac{1}{n}\sum_{k=1}^{n}x_{\kappa,k'}x_{\kappa,l'}\right)\right] \\
={}&\frac{1}{n^2}\mathbb{E}\left[\left(\sum_{k=1}^{n}x_{k,i'}x_{k,j'}\right)\left(\sum_{\kappa=1}^{n}x_{\kappa,k'}x_{\kappa,l'}\right)\right] \\
={}&\frac{1}{n^2}\mathbb{E}\left[\sum_{k=1}^{n}\sum_{\kappa=1}^{n}x_{k,i'}x_{k,j'}x_{\kappa,k'}x_{\kappa,l'}\right] \\
={}&\frac{1}{n^2}\mathbb{E}\left[\sum_{k=1}^{n}x_{k,i'}x_{k,j'}x_{k,k'}x_{k,l'}\right] \\
&+ \frac{1}{n^2}\mathbb{E}\left[\sum_{k=1}^{n}\sum_{\kappa=1,\kappa\neq k}^{n}x_{k,i'}x_{k,j'}x_{\kappa,k'}x_{\kappa,l'}\right].
\end{aligned} \tag{55}
$$

The summands in the first term, in the case of $j' \leq k'$, has the following form. The remaining orderings of $i',j',k',l'$ can be computed in a similar manner as follows.

$$
\begin{aligned}
&\mathbb{E}\left[\sum_{k=1}^{n}x_{k,i'}x_{k,j'}x_{k,k'}x_{k,l'}\right] \\
={}&\mathbb{E}\left[\sum_{k=1}^{n}\mathbb{P}\left[x_{k,i'} = x_{k,j'} = x_{k,k'} = x_{k,l'} = 1\right]\right] \\
={}&n\mathbb{E}\left[\left(p(\mathsf{P}^{i'-1})_{11} + (1-p)(\mathsf{P}^{i'-1})_{01}\right)(\mathsf{P}^{j'-i'})_{11}(\mathsf{P}^{k'-j'})_{11}(\mathsf{P}^{l'-k'})_{11}\right].
\end{aligned} \tag{56}
$$

Each summand in the second term of Eq. 55 contains a product of coactivations from two distinct chains, $z_k$ and $z_\kappa$ $(k \neq \kappa)$. Because the chains are independently drawn from the Markov kernel,

the terms $x_{k,i'}x_{k,j'}$ and $x_{\kappa,k'}x_{\kappa,l'}$ are statistically independent.

$$\mathbb{E}\left[\sum_{k=1}^{n}\sum_{\kappa=1,\kappa\neq k}^{n} x_{k,i'}x_{k,j'}x_{\kappa,k'}x_{\kappa,l'}\right]$$

$$= \mathbb{E}\left[\sum_{k=1}^{n}\sum_{\kappa=1,\kappa\neq k}^{n} \mathbb{P}\left[x_{k,i'}=x_{k,j'}=1\right]\mathbb{P}\left[x_{\kappa,k'}=x_{\kappa,l'}=1\right]\right]$$

$$= n(n-1)\mathbb{E}\left[\left(p(\mathsf{P}^{i'-1})_{11}+(1-p)(\mathsf{P}^{i'-1})_{01}\right)(\mathsf{P}^{j'-i'})_{11}\right.$$
$$\left.\left(p(\mathsf{P}^{k'-1})_{11}+(1-p)(\mathsf{P}^{k'-1})_{01}\right)(\mathsf{P}^{l'-k'})_{11}\right]. \tag{57}$$

**Evaluating RHS of Eq. 52:** $\mathbb{E}[x_{n+1,j}y_{n+1}\mathsf{G}_{i',j'}]$ **with** $i'\leq j'$.

$$\mathbb{E}[x_{n+1,j}y_{n+1}\mathsf{G}_{i',j'}] = \frac{1}{n}\sum_{k=1}^{n}\mathbb{E}\left[x_{n+1,j}y_{n+1}x_{k,i'}x_{k,j'}\right]$$

$$= \frac{1}{n}\sum_{k=1}^{n}\mathbb{E}\left[\mathbb{P}\left[x_{n+1,j}=y_{n+1}=1\right]\mathbb{P}\left[x_{k,i'}=x_{k,j'}=1\right]\right]$$

$$= \mathbb{E}\left[\left(p(\mathsf{P}^{j-1})_{11}+(1-p)(\mathsf{P}^{j-1})_{01}\right)(\mathsf{P}^{d+1-j})_{11}\right.$$
$$\left.\left(p(\mathsf{P}^{i'-1})_{11}+(1-p)(\mathsf{P}^{i'-1})_{01}\right)(\mathsf{P}^{j'-i'})_{11}\right]. \tag{58}$$

$\square$

We next extend the global minimization result to the case of a non-binary state space. The derivation follows a similar strategy as in the binary case, but now incorporates a Dirichlet prior over the Markov transition kernel P, where each row $\mathsf{P}_{s,:}$ ($s\in\mathcal{S}$) is independently sampled from $\mathrm{Dir}(\alpha\cdot\mathbf{1}_S)$. The concentration parameter $\alpha>0$ controls the variability of the transitions: smaller values encourage sparse, deterministic dynamics, while larger values yield more uniform behavior.

**Lemma C.8.** *Assumptions:*

- *Each row of* P *is sampled from the Dirichlet distribution of order $S$ with parameters $\alpha_s$ with $s\in\mathcal{S}=\{0,\ldots,|\mathcal{S}|\}$.*
- *The initial states are independently and uniformly sampled from $\mathcal{S}$.*

*Define $H\in\mathbb{R}^{dm\times dm}$ as*

$$H_{r,c} = \frac{1}{|\mathcal{S}|}\mathbb{E}\left[\sum_{s\in\mathcal{S}}\sum_{s'\in\mathcal{S}}\sum_{s''\in\mathcal{S}} ss'(\mathsf{P}^{j-i})_{ss'}(\mathsf{P}^{i-1})_{ss'}\right]$$

$$\left(\frac{1}{n|\mathcal{S}|}\mathbb{E}\left[\sum_{s_1,s_2,s_3,s_4,s_5\in\mathcal{S}} s_1s_2s_3s_4s_5(P^{i'-1})_{s_1s_2}(P^{j'-i'})_{s_2s_3}(P^{k'-j'})_{s_3s_4}(P^{l'-k'})_{s_4s_5}\right]\right.$$

$$\frac{n-1}{n|\mathcal{S}|}\mathbb{E}\left[\left(\sum_{s_1,s_2,s_3\in\mathcal{S}} s_1s_2s_3(P^{i'-1})_{s_1s_2}(P^{j'-i'})_{s_2s_3}\right)\right.$$

$$\left.\left.\left(\sum_{s_1,s_2,s_3\in\mathcal{S}} s_1s_2s_3(\mathsf{P}^{k'-1})_{s_1s_2}(P^{l'-k'})_{s_2s_3}\right)\right]\right], \tag{59}$$

*where* $r = (i-1)m+j'+\sum_{\tau=0}^{i'-2}d+1-\tau$, $c = (j-1)m+l'+\sum_{\tau=0}^{k'-2}d+1-\tau$.

*Define $h \in \mathbb{R}^{dm}$ as*

$$h_{(j-1)m+j'+\sum_{\tau=0}^{i'-2} d+1-\tau} = \mathbb{E}\left[\left(\sum_{s_1,s_2,s_3\in\mathcal{S}} s_1 s_2 s_3 (\mathsf{P}^{j-1})_{s_1 s_2}(\mathsf{P}^{d+1-j})_{s_2 s_3}\right)\right.$$
$$\left.\left(\sum_{s_1,s_2,s_3\in\mathcal{S}} s_1 s_2 s_3 (\mathsf{P}^{i'-1})_{s_1 s_2}(\mathsf{P}^{j'-i'})_{s_2 s_3}\right)\right]. \tag{60}$$

*The global minimum $X^* \in \mathbb{R}^{dm}$ of the objective function described in Eq. 49 equals $X^* = H^{-1}h$.*

*Proof.* **Evaluating entries in $H$:** $\mathbb{E}\left[x_{n+1,i}x_{n+1,j}\mathsf{G}_{i',j'}\mathsf{G}_{k',l'}\right]$ **with** $i,j \in [d]$, $i \leq j$, $i',j',k,l' \in [d+1]$, **and** $i' \leq k', j' \leq l'$**.** Since $x_{n+1}$ and $x_i$ $(i \in [n])$ are independent, as before, we evaluate $\mathbb{E}\left[x_{n+1,i}x_{n+1,j}\right]$ and $\mathbb{E}\left[\mathsf{G}_{i',j'}\mathsf{G}_{k',l'}\right]$ separately.

The first term captures the expected coactivation between two positions in the query chain.

$$\mathbb{E}\left[x_{n+1,i}x_{n+1,j}\right] = \mathbb{E}\left[\sum_{s\in\mathcal{S}}\sum_{s'\in\mathcal{S}} ss'\mathbb{P}\left[x_{n+1,i}=s, x_{n+1,j}=s'\right]\right]$$

$$= \mathbb{E}\left[\sum_{s\in\mathcal{S}}\sum_{s'\in\mathcal{S}} ss'(\mathsf{P}^{j-i})_{ss'}\mathbb{P}[x_{n+1,i}=s]\right]$$

$$= \frac{1}{|\mathcal{S}|}\mathbb{E}\left[\sum_{s\in\mathcal{S}}\sum_{s'\in\mathcal{S}}\sum_{s''\in\mathcal{S}} ss'(\mathsf{P}^{j-i})_{ss'}(\mathsf{P}^{i-1})_{ss'}\right]. \tag{61}$$

We proceed to evaluate the second term, which denotes the aggregated coactivation over the in-context samples.

$$\mathbb{E}\left[\mathsf{G}_{i',j'}\mathsf{G}_{k',l'}\right] = \frac{1}{n^2}\mathbb{E}\left[\sum_{k=1}^{n} x_{k,i'}x_{k,j'}x_{k,k'}x_{k,l'}\right]$$

$$+ \frac{1}{n^2}\mathbb{E}\left[\sum_{k=1}^{n}\sum_{\kappa=1,\kappa\neq k}^{n} x_{k,i'}x_{k,j'}x_{\kappa,k'}x_{\kappa,l'}\right]. \tag{62}$$

The summands in the first term, in the case of $j' \leq k'$, has the following form. The remaining orderings of $i',j',k',l'$ can be computed in a similar manner.

$$\mathbb{E}\left[\sum_{k=1}^{n} x_{k,i'}x_{k,j'}x_{k,k'}x_{k,l'}\right]$$
$$= \frac{n}{|\mathcal{S}|}\mathbb{E}\sum_{s_1,s_2,s_3,s_4,s_5\in\mathcal{S}} s_1 s_2 s_3 s_4 s_5 (P^{i'-1})_{s_1 s_2}(P^{j'-i'})_{s_2 s_3}(P^{k'-j'})_{s_3 s_4}(P^{l'-k'})_{s_4 s_5}. \tag{63}$$

In the summands of the second term in Eq. 62 contains the product of the coactivation from two distinct chains $z_k, z_\kappa$ $(k \neq \kappa)$. Since the two chains are independently sampled from the Markovian kernel, the two products $x_{k,i'}x_{k,j'}$ and $x_{\kappa,k'}x_{\kappa,l'}$ are independent.

$$\mathbb{E}\left[\sum_{k=1}^{n}\sum_{\kappa=1,\kappa\neq k}^{n} x_{k,i'}x_{k,j'}x_{\kappa,k'}x_{\kappa,l'}\right]$$

$$= n(n-1)\mathbb{E}\left[\left(\sum_{s_1,s_2,s_3\in\mathcal{S}} s_1 s_2 s_3 (\mathsf{P}^{i'-1})_{s_1 s_2}(\mathsf{P}^{j'-i'})_{s_2 s_3}\right)\right.$$

$$\left.\left(\sum_{s_1,s_2,s_3\in\mathcal{S}} s_1 s_2 s_3 (\mathsf{P}^{k'-1})_{s_1 s_2}(\mathsf{P}^{l'-k'})_{s_2 s_3}\right)\right]. \tag{64}$$

**Evaluating entries in** $h$**:** $\mathbb{E}[x_{n+1,j}y_{n+1}\mathsf{G}_{i',j'}]$ **with** $i' \leq j'$**.** We utilize the independence of the query and in-context chains to separate the expectation of $x_{n+1,j}y_{n+1}$ and $\mathsf{G}_{i',j'}$. Specifically, we have

$$
\begin{aligned}
\mathbb{E}[x_{n+1,j}y_{n+1}\mathsf{G}_{i',j'}] &= \frac{1}{n}\sum_{k=1}^{n}\mathbb{E}\left[x_{n+1,j}y_{n+1}x_{k,i'}x_{k,j'}\right] \\
&= \frac{1}{n}\sum_{k=1}^{n}\mathbb{E}\left[x_{n+1,j}y_{n+1}\right]\mathbb{E}\left[x_{k,i'}x_{k,j'}\right] \\
&= \frac{1}{n}\sum_{k=1}^{n}\mathbb{E}\left[\left(\sum_{s_1,s_2,s_3\in\mathcal{S}}s_1s_2s_3(\mathsf{P}^{j-1})_{s_1s_2}(\mathsf{P}^{d+1-j})_{s_2s_3}\right)\right. \\
&\qquad\qquad\left.\left(\sum_{s_1,s_2,s_3\in\mathcal{S}}s_1s_2s_3(\mathsf{P}^{i'-1})_{s_1s_2}(\mathsf{P}^{j'-i'})_{s_2s_3}\right)\right].
\end{aligned}
\tag{65}
$$

$\square$

Although we obtain a closed-form expression for the global minimum $X^* = H^{-1}h$, the entries of both $H$ and $h$ involve high-order interactions across the kernel. In particular, each entry entails expectations of products of entries from powers of a random transition matrix $\mathsf{P}$, where each row of $\mathsf{P}$ is independently sampled from a Dirichlet prior. Evaluating such expectations is analytically challenging due to the nonlinear dependencies introduced by matrix multiplication, especially when powers of $\mathsf{P}$ couple multiple rows. However, for fixed values of the Dirichlet concentration parameter $\alpha$, these expectations are tractable via Monte Carlo sampling, enabling empirical evaluation of $H$ and $h$.

Moreover, the functional form of these entries suggests that $H$ is dense in general, as most entries involve sums over all state triplets and nontrivially depend on all entries of $\mathsf{P}$. As a consequence, the inverse $H^{-1}$ is also expected to be dense, indicating that the global optimum $X^*$ integrates information across all positions.

### C.4 Optimization Limitation of Single-Layer LSA (Theorem 3.5)

We have found the global minimizer of the reparameterized objective $\tilde{f}$. The goal is to find a transformer parameter $(P,Q)$ such that $\phi(P,Q) = X^*$. We show that in general this problem is NP-hard in Theorem 3.5. We formally state the original problem to solve the reparameterization equation for transformer parameters $(b, A)$ below.

**Problem C.9** (Transformer parameter reconstruction.). *Let* $d \in \mathbb{Z}_{\geq 1}$ *denote the dimension of the in-context data, i.e., the length of the in-context Markov chain minus one. Given* $X \in \mathbb{R}^{dm}$ *($m = (d+1)(d+2)/2$), solve the following system for* $b \in \mathbb{R}^{d+1}$, $A \in \mathbb{R}^{(d+1)\times d}$

$$
\begin{cases} b_i A_{k,j} = X_r & \text{if } i = k \\ b_i A_{k,j} + b_k A_{i,j} = X_r & \text{if } i \neq k \end{cases},
\tag{66}
$$

*where* $i, k \in [d+1]; j \in [d]; r = (j-1)m + i(d+1) + k - \sum_{i'\leq i}(i'-1)$.

We introduce a broader class of bilinear feasibility problem that incorporates the above problem as follows.

**Problem C.10** (Bilinear feasibility.). *Let* $d \in \mathbb{Z}_{\geq 1}$ *denote the dimension of the in-context data, i.e., the length of the in-context Markov chain minus one. Define* $m = (d+1)(d+2)/2$. *Let* $X \in \mathbb{R}^{dm}$ *and* $D^{(r)} \in \mathbb{R}^{(d+1)\times(d+1)}$ *($r \in [dm]$) be given. Solve the following system for* $b \in \mathbb{R}^{d+1}$, $A \in \mathbb{R}^{(d+1)\times d}$.

$$
b^\top D^{(r)} A_{:,j} = X_r,
\tag{67}
$$

*where* $j \in [d]; r = (j-1)m + i(d+1) + k - \sum_{i'\leq i}(i'-1)$ *with* $i, k \in [d+1]$.

Next, we define a bilinear program and establish its equivalence with the above bilinear feasibility problem. Each equality in Eq. 67 is equivalent to two inequalities. We hold one inequality out and treat it as the objective. We state the bilinear program below.

**Problem C.11** (Bilinear Program). *Let $X$ be some given vector in $\mathbb{R}^{dm}$ where $d \in \mathbb{Z}_{\geq 1}$ is the problem dimension and $m = (d+1)(d+2)/2$. Let $D^{(r)} \in \mathbb{R}^{(d+1)\times(d+1)}$ with $r \in [dm]$ be some given matrices. Find the optimal solution $b^* \in \mathbb{R}^d$, $A^* \in \mathbb{R}^{(d+1)\times d}$ for the following problem:*

$$\min_{b\in\mathbb{R}^d, A\in\mathbb{R}^{(d+1)\times d}} \quad b^\top D^{(1)} A_{:,1} - X_1$$

$$s.t. \qquad b^\top D^{(1)} A_{:,1} \geq X_1,$$
$$b^\top D^{(r)} A_{:,j} \leq X_r,$$
$$b^\top D^{(r)} A_{:,j}, \geq X_r, \tag{68}$$

*where $j \in [d], r = (j-1)m + i(d+1) + k - \sum_{i'\leq i}(i'-1)$ with $i,k \in [d+1]$.*

We now show the equivalence between solving the bilinear feasibility problem and finding a zero minimum in the bilinear program.

**Lemma C.12** (Equivalence between bilinear feasibility and bilinear program). *The bilinear feasibility problem (67) has a solution if and only if the bilinear program (68) has a feasible solution whose objective function value is zero.*

*Proof.* Before showing the equivalence, we note that the minimal objective function value achievable by the feasible variables for the program (68) is greater than or equal to zero. Because for a solution to be feasible, it must satisfy $b^\top D^{(1)} A_{:,1} \geq X_1$ and hence $b^\top D^{(1)} A_{:,1} - X_1$ must be greater than or equal to zero.

($\Rightarrow$) Suppose the bilinear feasibility problem has a solution $b, A$. Then they must satisfy the inequality constraints in the program (68). Moreover, the objective function value would also evaluate to zero by definition of $b, A$. Therefore, $b, A$ is an optimal solution of the bilinear program (68).

($\Leftarrow$) Suppose $b^*, A^*$ is the optimal solution of the bilinear program (68) whose objective function value is zero. Then they must satisfy all inequality constraints, which are equivalent to $dm - 1$ equations in the bilinear system (67). Additionally, the optimal solution guarantees the objective being zero and hence the remaining one equation also holds. □

**Bilinear separability program.** By Lemma C.12, it suffices to show finding a feasible solution achieving zero minimum for the bilinear program (68) is NP-hard. To do this, we reduce from bilinear program (13) in Theorem 3.1 in [48]. This bilinear program has been shown to be NP-hard by reducing from the problem of bilinear separability: determining whether two disjoint sets of points can be strictly separated by two planes such that one set occupies exactly three of the four regions defined by the planes. Let $n'$ denote dimension of the points and $m', k'$ are the number of points in the two sets. Let $A' \in \mathbb{R}^{m'\times n'}$ and $B \in \mathbb{R}^{k'\times n'}$ store the points in the two disjoint sets, where each row corresponds to a point. Let $e$ denote a vector of ones with arbitrary dimension. We follow their notation, and in cases of conflict, we use a prime symbol to indicate their variables. Their original system to determine the bilinear separability is:

$$\min_{z^1,z^2,w^1,w^2,\gamma^1,\gamma^2} \quad z^1 z^2$$

$$\text{s.t.} \quad \begin{array}{ll} -A'w^1 + \gamma^1 e + e \leq 0, & \quad -A'w^2 + \gamma^2 e + e \leq 0, \\ Bw^1 - \gamma^1 e + e \leq z^1, & \quad Bw^2 - \gamma^2 e + e \leq z^2, \\ 0 \leq z^1, & \quad 0 \leq z^2. \end{array} \tag{69}$$

Here, $z^1, z^2 \in \mathbb{R}^{k'}$ are decision variables, $w^1, w^2 \in \mathbb{R}^n$ and $\gamma^1, \gamma^2 \in \mathbb{R}$ are the remaining auxiliary variables; $e$ is some given constant vector in $\mathbb{R}^{n'}$. The above system has zero minimum if and only if the two sets are separable in the above mentioned way.

Program (69) is equivalent to the following problem after splitting the decision variables into two nonnegative parts.

$$\min_{\substack{z^1_+,z^1_-,z^2_+,z^2_-, \\ w^1,w^2,\gamma^1,\gamma^2}} \quad (z^1_+ - z^1_-)(z^2_+ - z^2_-)$$

$$\text{s.t.} \quad \begin{array}{ll} -A'w^1 + \gamma^1 e + e \leq 0, & \quad -A'w^2 + \gamma^2 e + e \leq 0, \\ Bw^1 - \gamma^1 e + e \leq z^1_+ - z^1_-, & \quad Bw^2 - \gamma^2 e + e \leq z^2_+ - z^2_-, \\ 0 \leq z^1_+, z^1_-, & \quad 0 \leq z^2_+, z^2_-. \end{array} \tag{70}$$

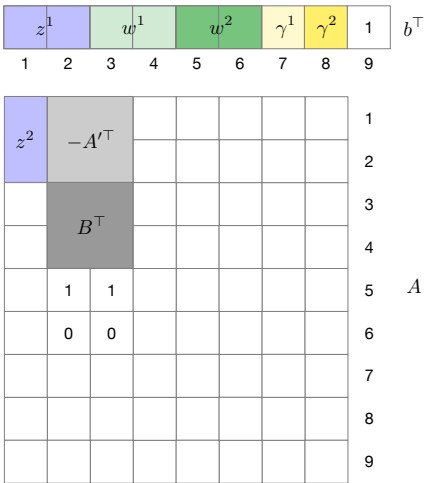

Figure 9: How each variable in the LSA parameter recovery problem is utilized to construct a feasibility program that contains the objective and all constraints in the bilinear separability program (70) for the case of $m' = k' = 2, n' = 2$, and $d = 8$.

The reduction idea is to map decision variables to the first few entries of $(b, A_{\cdot,1})$, auxiliary variables to entries in $b$, and as many constants as possible to entries in $A$. We first give a reduction example (C.13) and then formalize the NP-hardness of solving the bilinear program (68) corresponding to recovering transformer parameters from reparameterization.

*Example* C.13. As an example, we consider the case for $m' = k' = 2, n' = 2$, and $d = 8$. We detail the variable assignment below and visualize it in Fig. 9.

In this example, we explicitly map each constant, decision, and auxiliary variable from the bilinear separability program (70) to the corresponding parameter entries in the bilinear feasibility program (68). We assign variables to the following matrices and vectors in the separability program: (i) *constant*: $A' \in \mathbb{R}^{2 \times 2}, B \in \mathbb{R}^{2 \times 2}, e = [1 \quad 1]^\top$. (ii) decision variables: $z_+^1, z_-^1, z_+^2, z_-^2 \in \mathbb{R}^2$; (iii) auxiliary variables: $w^1, w^2 \in \mathbb{R}^2, \gamma^1, \gamma^2 \in \mathbb{R}^2$.

For indexing, we use $b_{i:i'}$ to denote the subvector consisting of entries $i$ through $i'$ of $b$, and $A_{i:i',j}$ to denote the corresponding rows $i$ through $i'$ of the $j$th column of $A$. Additionally, $A_{i:i',j:j'}$ denotes the submatrix of $A$ spanning rows $i$ to $i'$ and columns $j$ to $j'$, inclusive. We use $(z_+^1)_i$ to denote the $i$th entry of $z_+^1$.

**Decision variables and objective for** $m' = k' = 2$. We map the decision variables and objective from the separability program into the feasibility program framework. To do so, we split $b_{1:2}, A_{1:2,1}$ into two nonnegative components and keep the rest of the variables as it is. Let $\tilde{b}$ denote the resulting variable:

$$\tilde{b} := \begin{bmatrix} b_1^+ - b_1^- & b_2^+ - b_2^- & b_3 & b_4 & b_5 & b_6 & b_7 & b_8 & b_9 \end{bmatrix}^\top. \tag{71}$$

Likewise, we denote a partially split version of the first column of $A$ as follows

$$\tilde{A}_{:,1} := \begin{bmatrix} A_{1,1}^+ - A_{1,1}^- & A_{2,1}^+ - A_{2,1}^- & A_{3,1} & A_{4,1} & A_{5,1} & A_{6,1} & A_{7,1} & A_{8,1} & A_{9,1} \end{bmatrix}^\top. \tag{72}$$

*Variable assignment:* We map decision variables as follows:

$$z_+^1 \mapsto b_{1:2}^+, \quad z_-^1 \mapsto b_{1:2}^-, \quad z_+^2 \mapsto A_{1:2,1}^+, \quad z_-^2 \mapsto A_{1:2,1}^-. \tag{73}$$

$D^{(1)}$'s *construction:* We set $D^{(1)}$'s entry to 1 at the following positions and 0 otherwise:

$$\{(i,i) : i \in [2]\}. \tag{74}$$

Then the objective in the feasibility program becomes

$$\min (b_{1:2}^+ - b_{1:2}^-) \cdot (A_{1:2,1}^+ - A_{1:2,1}^-) - X_1. \tag{75}$$

Since $X_1$ is fixed, minimizing the above expression is equivalent to minimizing the objective in (70). Moreover, because we apply the same variable splitting to the corresponding entries of $b$ and $A$, the nonnegativity constraints on $z_+^1, z_-^1, z_+^2, z_-^2$ are preserved in the feasibility formulation.

**Linear constraints for $m' = k' = n' = 2$.** We then encode the linear constraints into the feasibility program setup.

*Variable assignment:* 1. We map the given constant matrices $A', B$ to sub-matrices of $A$. Since the first column of $A$ is reserved for decision variables, we begin assigning the remaining constants starting from the second column.

$$A' \mapsto -A_{1:2,2:3}^\top, \quad B \mapsto A_{3:4,2:3}^\top. \tag{76}$$

2. We assign the entries in $\tilde{b}$ after the decision variables to the auxiliary variables:

$$w^1 \mapsto b_{3:4}, \quad w^2 \mapsto b_{5:6}, \quad \gamma^1 \mapsto b_7, \quad \gamma^2 \mapsto b_8. \tag{77}$$

3. We further require a few entries to be fixed at 0 or 1:

$$b_9 = A_{5,2:3} := 1, \quad A_{6,2:3} := 0. \tag{78}$$

*Counting constraints in both programs:* In the separability program, the constraints $-A'w^1 + \gamma^1 e + e \leq 0$ and $-A'w^2 + \gamma^2 e + e \leq 0$ each consist of 2 scalar inequalities. Similarly, the constraints $Bw^1 - \gamma^1 e + e \leq z_+^1 - z_-^1$ and $Bw^2 - \gamma^2 e + e \leq z_+^2 - z_-^2$ also contribute 2 inequalities each. In total, there are 8 scalar constraints in the separability program, and we construct 8 corresponding $D^{(r)}$ matrices in the feasibility program to match these.

In the feasibility program, the right-hand side index $r$ ranges from 1 to $dm = 8 \cdot 45 = 360$. Due to the indexing convention, incrementing the column index $j$ of $A$ increases $r$ by $m = 45$. Specifically, for each $j$, the set $r \in \{(j-1)m+1, (j-1)m+2, \ldots, jm\} = \{45(j-1)+1, 45(j-1)+2, \ldots, 45j\}$ corresponds to 45 constraints involving the expression $\tilde{b}^\top D^{(r)} A_{:,j}$.

*$D^{(r)}$'s construction:* For each linear constraint in the separability program, we proceed as follows: (a) express the constraint in its full matrix form; (b) encode it using the variables in the feasibility program based on the earlier assignments (Eqs. 76-78); and (c) configure the corresponding $D^{(r)}$ to implement the constraint within the feasibility setup. We set each $D^{(r)}$ based on the terms $b_i A_{k,j}$ that appear in the constraint written in step (b). For instance, if a term $b_i A_{k,j}$ appears with coefficient 1, we set the corresponding entry $D_{i,k}^{(r)} = 1$.

1. $-A'w^1 + \gamma^1 e + e \leq 0$.
   (a) Full matrix:

$$-\begin{bmatrix} A'_{1,1} & A'_{1,2} \\ A'_{2,1} & A'_{2,2} \end{bmatrix} \begin{bmatrix} w_1^1 \\ w_2^1 \end{bmatrix} + \gamma^1 \begin{bmatrix} 1 \\ 1 \end{bmatrix} + \begin{bmatrix} 1 \\ 1 \end{bmatrix} \leq \begin{bmatrix} 0 \\ 0 \end{bmatrix}. \tag{79}$$

   (b) Mapping to feasibility program's constraints:

$$\begin{bmatrix} A_{1,2} & A_{2,2} \\ A_{1,3} & A_{2,3} \end{bmatrix} \begin{bmatrix} b_3 \\ b_4 \end{bmatrix} + b_7 \begin{bmatrix} A_{5,2} \\ A_{5,3} \end{bmatrix} + b_9 \begin{bmatrix} A_{5,2} \\ A_{5,3} \end{bmatrix} \leq b_9 \begin{bmatrix} A_{6,2} \\ A_{6,3} \end{bmatrix}. \tag{80}$$

   (c) We use the first index $r$ corresponding to $j = 2$ to encode the first row of inequality (80), and the first $r$ corresponding to $j = 3$ to encode the second row. This leads us to construct $D^{(46)}$ and $D^{(91)}$. Based on the terms in (80), we set the following entries of $D^{(45)}, D^{(91)}$ as 1 and others 0:

$$\{(3,1), (4,2)\} \cup \{(7,5)\} \cup \{(9,5)\} \cup \{(9,6)\}. \tag{81}$$

2. $-A'w^2 + \gamma^2 e + e \leq 0$.
   (a) Full matrix:

$$-\begin{bmatrix} A'_{1,1} & A'_{1,2} \\ A'_{2,1} & A'_{2,2} \end{bmatrix} \begin{bmatrix} w_2^1 \\ w_2^2 \end{bmatrix} + \gamma^2 \begin{bmatrix} 1 \\ 1 \end{bmatrix} + \begin{bmatrix} 1 \\ 1 \end{bmatrix} \leq \begin{bmatrix} 0 \\ 0 \end{bmatrix}. \tag{82}$$

(b) Mapping to feasibility program's constraints:

$$\begin{bmatrix} A_{1,2} & A_{2,2} \\ A_{1,3} & A_{2,3} \end{bmatrix} \begin{bmatrix} b_5 \\ b_6 \end{bmatrix} + b_8 \begin{bmatrix} A_{5,2} \\ A_{5,3} \end{bmatrix} + b_9 \begin{bmatrix} A_{5,2} \\ A_{5,3} \end{bmatrix} \leq b_9 \begin{bmatrix} A_{6,2} \\ A_{6,3} \end{bmatrix}. \tag{83}$$

(c) We use the second index $r$ corresponding to $j = 2$ to encode the first row of inequality (83), and the second $r$ corresponding to $j = 3$ to encode the second row, which leads us to construct $D^{(47)}$ and $D^{(92)}$. Based on the terms in (83), we set the following entries of $D^{(47)}, D^{(92)}$ as 1 and others 0:

$$\{(5,1),(6,2)\} \cup \{(8,5)\} \cup \{(9,5)\} \cup \{(9,6)\}. \tag{84}$$

3. $Bw^1 - \gamma^1 e + e \leq z_+^1 - z_-^1$.
   (a) Full matrix:

$$\begin{bmatrix} B_{1,1} & B_{1,2} \\ B_{2,1} & B_{2,2} \end{bmatrix} \begin{bmatrix} w_1^1 \\ w_2^1 \end{bmatrix} - \gamma^1 \begin{bmatrix} 1 \\ 1 \end{bmatrix} + \begin{bmatrix} 1 \\ 1 \end{bmatrix} \leq \begin{bmatrix} (z_+^1)_1 - (z_-^1)_1 \\ (z_+^1)_2 - (z_-^1)_2 \end{bmatrix}. \tag{85}$$

   (b) Mapping to feasibility program's constraints:

$$\begin{bmatrix} A_{3,2} & A_{4,2} \\ A_{3,3} & A_{4,3} \end{bmatrix} \begin{bmatrix} b_3 \\ b_4 \end{bmatrix} - b_7 \begin{bmatrix} A_{5,2} \\ A_{5,3} \end{bmatrix} + b_9 \begin{bmatrix} A_{5,2} \\ A_{5,3} \end{bmatrix} \leq \begin{bmatrix} (b_1^+ - b_1^-)A_{5,2} \\ (b_2^+ - b_2^-)A_{5,3} \end{bmatrix}. \tag{86}$$

   (c) We use the third index $r$ corresponding to $j = 2$ to encode the first row of inequality (86), and the third $r$ corresponding to $j = 3$ to encode the second row. We set the following entries of $D^{(48)}, D^{(93)}$ as 1

$$\{(3,3),(4,4)\} \cup \{(9,5)\}, \tag{87}$$

   the following entries to $-1$

$$\{(7,5)\} \cup \{(j-1,5)\}, \tag{88}$$

   and the rest to 0, where $j = 2$ for $D^{(48)}$ and $j = 3$ for $D^{(93)}$.
4. $Bw^2 - \gamma^2 e + e \leq z_+^2 - z_-^2$.
   (a) Full matrix:

$$\begin{bmatrix} B_{1,1} & B_{1,2} \\ B_{2,1} & B_{2,2} \end{bmatrix} \begin{bmatrix} w_1^2 \\ w_2^2 \end{bmatrix} - \gamma^2 \begin{bmatrix} 1 \\ 1 \end{bmatrix} + \begin{bmatrix} 1 \\ 1 \end{bmatrix} \leq \begin{bmatrix} (z_+^2)_1 - (z_-^2)_1 \\ (z_+^2)_2 - (z_-^2)_2 \end{bmatrix}. \tag{89}$$

   (b) Mapping to feasibility program's constraints:

$$\begin{bmatrix} A_{3,2} & A_{4,2} \\ A_{3,3} & A_{4,3} \end{bmatrix} \begin{bmatrix} b_5 \\ b_6 \end{bmatrix} - b_8 \begin{bmatrix} A_{5,2} \\ A_{5,3} \end{bmatrix} + b_9 \begin{bmatrix} A_{5,2} \\ A_{5,3} \end{bmatrix} \leq \begin{bmatrix} (b_1^+ - b_1^-)A_{5,2} \\ (b_2^+ - b_2^-)A_{5,3} \end{bmatrix}. \tag{90}$$

   (c) We use the forth index $r$ corresponding to $j = 2$ to encode the first row of inequality (90), and the forth $r$ corresponding to $j = 3$ to encode the second row. We set the following entries of $D^{(49)}, D^{(94)}$ as 1

$$\{(5,3),(6,4)\} \cup \{(9,5)\}, \tag{91}$$

   the following entries to $-1$

$$\{(8,5)\} \cup \{(j-1,5)\}, \tag{92}$$

   and the rest to 0, where $j = 2$ for $D^{(49)}$ and $j = 3$ for $D^{(94)}$.

In this example, we matched each constraint by configuring the corresponding $D^{(r)}$ matrices along with the variable assignments for $b$ and $A$. Additionally, we explicitly set $X_{46}, X_{91}$, $X_{47}, X_{92}$, $X_{48}, X_{93}, X_{49}, X_{94}$ to zero.

Building on the illustrative reduction above, we now formalize the general reduction. The variable assignment patterns and matrix constructions introduced earlier naturally extend to arbitrary problem sizes, as detailed below.

**Theorem C.14** (*Theorem 3.5 restated*). *Let $d \in \mathbb{Z}_{\geq 1}$ denote the dimension of the in-context data (the length of the Markov chain minus one), and let $m = \frac{(d+1)(d+2)}{2}$. Given $X \in \mathbb{R}^{dm}$ representing the global minimizer of the reparameterized loss, solving for 1-layer LSA parameters $(b, A)$ that satisfy the reparameterization equation Eq. 8 is NP-hard with respect to $d$.*

*Proof.* We map each constant, decision and auxiliary variable from the bilinear program for the separability problem (70) to the variables in that for the bilinear feasibility problem (68). By freezing the variables that were assigned a constant, we obtain a program for bilinear feasibility that subsumes the objective and constraints in the separability program.

Specifically, we make assignment for the following entities in the separability program (70): (i) *constant*: $A' \in \mathbb{R}^{m' \times n'}$, $B \in \mathbb{R}^{k' \times n'}$, $e = \mathbf{1}$ whose shape is dependent on context; (ii) decision variables: $z_+^1, z_-^1, z_+^2, z_-^2 \in \mathbb{R}^{k'}$; (iii) auxiliary variables: $w^1, w^2 \in \mathbb{R}^{n'}$, $\gamma^1, \gamma^2 \in \mathbb{R}^{n'}$.

**Decision variables and objective.** We split $b_{1:k'}, A_{1:k',1}$ into two nonnegative components and keep the rest of the variables as it is. Let $\tilde{b}$ denote the resulting variable:

$$\tilde{b} := \begin{bmatrix} b_1^+ - b_1^- & \cdots & b_{k'}^+ - b_{k'}^- & b_{k'+1} & \cdots & b_{d+1} \end{bmatrix}^\top. \tag{93}$$

Likewise, we denote a partially split version of the first column of $A$ as follows

$$\tilde{A}_{:,1} := \begin{bmatrix} A_{1,1}^+ - A_{1,1}^- & \cdots & A_{k',1}^+ - A_{k',1}^- & A_{k'+1,1} & \cdots & A_{d+1,1} \end{bmatrix}^\top. \tag{94}$$

*Variable assignment:* We map decision variables as follows:

$$z_+^1 \mapsto b_{1:k'}^+, \quad z_-^1 \mapsto b_{1:k'}^-, \quad z_+^2 \mapsto A_{1:k',1}^+, \quad z_-^2 \mapsto A_{1:k',1}^-. \tag{95}$$

$\underline{D^{(1)}\text{'s construction:}}$ We set $D^{(1)}$'s entry to 1 at the following positions and 0 otherwise:

$$\{(i,i) : i \in [k']\}. \tag{96}$$

Then the objective in the feasibility program becomes

$$\min \, (b_{1:k'}^+ - b_{1:k'}^-) \cdot (A_{1:k',1}^+ - A_{1:k',1}^-) - X_1. \tag{97}$$

Since $X_1$ is constant by construction, minimizing the bilinear term in the feasibility program directly corresponds to minimizing the objective in the separability program (70). The same variable splitting applied to the relevant entries of $b$ and $A$ ensures that the nonnegativity constraints on $z_+^1, z_-^1, z_+^2, z_-^2$ are preserved.

**Linear constraints.** To encode each linear constraint in program (70), we construct $D^{(r)}$s and make appropriate assignments for constants and auxiliary variables.

*Variable assignment:* 1. We embed the constant matrices $A'$ and $B$ into submatrices of $A$, starting from the second column.

$$A' \mapsto -A_{1:n',2:1+m'}^\top, \quad B \mapsto A_{n'+1:2n',2:1+k'}^\top. \tag{98}$$

2. We assign the entries in $\tilde{b}$ after the decision variables to the auxiliary variables:

$$w^1 \mapsto b_{k'+1:k'+n'}, \quad w^2 \mapsto b_{k'+n'+1:k'+2n'}, \quad \gamma^1 \mapsto b_{k'+2n'+1}, \quad \gamma^2 \mapsto b_{k'+2n'+2}. \tag{99}$$

3. We further require a few entries to be fixed at 0 or 1:

$$b_{k'+2n'+3} = A_{2n'+1,2:1+\max(m',k')} := 1, \quad A_{2n'+2,2:1+\max(m',k')} := 0. \tag{100}$$

*Counting constraints in both programs:* For constraints $-A'w^1 + \gamma^1 e + e \le 0$ and $-A'w^2 + \gamma^1 e + e \le 0$, they each contain $m'$ inequalities. Similarly, for constraints $Bw^1 - \gamma^1 e + e \le z_+^1 - z_-^1$ and $Bw^2 - \gamma^2 e + e \le z_+^2 - z_-^2$, they each contain $k'$ inequalities. In total, we set $2(m' + k')$ $D^{(r)}$s to encode the linear constraints.

In the feasibility program, the index for the RHS entries ($r$) ranges from 1 to $dm$. By the relationship between $r$ and the index for $b$ and $A$, if the column index of $A$ is incremented by 1, $r$ would increase by $m$. In particular, any $r \in \{(j-1)m+1, (j-1)m+2 \ldots, jm\}$ corresponds to a constraint in the feasibility program with LHS of the form $\tilde{b}^\top D^{(r)} A_{:,j}$.

$\underline{D^{(r)}\text{'s construction:}}$ For each $D^{(r)}$, we set four parts of its entries to be nonzero, in order to match with each term in the linear constraint.

Take the first group of constraints $-A'w^1 + \gamma^1 e + e \leq 0$ as an example, the first row is

$$-A'_{1,:} \cdot w^1 + \gamma^1 + 1 \leq 0. \tag{101}$$

By the variable assignment, the desired inequality in the feasibility program would utilize partial entries in the first *column* of $A$:

$$b_{k'+1:k'+n'} \cdot A_{1:n',2} + b_{k'+2n'+1} \underbrace{A_{2n'+1,2}}_{=1} + \underbrace{b_{k'+2n'+3}A_{2n'+1,2}}_{=1} \leq -b_{k'+2n'+3} \underbrace{A_{2n'+2,2}}_{=0}. \tag{102}$$

The above inequality can be achieved by setting $n' + 3$ entries in $D^{(jm+1)}$ accordingly, where blue and red indicate the row and column to be set to nonzero, respectively. The rest of the rows in the first group of constraints utilizes the next $m' - 1$ columns of $A$. Since moving column index $j$ corresponds to shifting $r$ by $m$, the corresponding inequalities are of the form: $\tilde{b}^\top D^{jm+1} A_{:,j+1} \leq X_{jm+1}$ for $j \in [m']$.

Using the above method, we map each linear constraint below.

1. $-A'w^1 + \gamma^1 e + e \leq 0$. For $j \in [m']$, we set the following entries of $D^{(jm+1)}$ as 1 and others 0:

$$\{(k' + k, k) : k \in [n']\} \cup \{(k' + 2n' + 1, 2n' + 1)\}$$
$$\cup \{(k' + 2n' + 3, 2n' + 1)\} \cup \{(k' + 2n' + 3, 2n' + 2)\}. \tag{103}$$

2. $-A'w^2 + \gamma^2 e + e \leq 0$. For $j \in [m']$, we set the following entries of $D^{(jm+2)}$ as 1 and others 0:

$$\{(k' + n' + k, k) : k \in [n']\} \cup \{(k' + 2n' + 2, 2n' + 1)\}$$
$$\cup \{(k' + 2n' + 3, 2n' + 1)\} \cup \{(k' + 2n' + 3, 2n' + 2)\}. \tag{104}$$

3. $Bw^1 - \gamma^1 e + e \leq z_+^1 - z_-^1$. For $j \in [k']$, we set the following entries of $D^{(jm+3)}$ as 1

$$\{(k' + k, n' + k) : k \in [n']\} \cup \{(k' + 2n' + 3, 2n' + 1)\}, \tag{105}$$

the following entries to $-1$

$$\{(k' + 2n' + 1, 2n' + 1)\} \cup \{(j - 1, 2n' + 1)\}, \tag{106}$$

and the remaining entries to 0.

4. $Bw^2 - \gamma^2 e + e \leq z_+^2 - z_-^2$. For $j \in [k']$, we set the following entries of $D^{(jm+4)}$ as 1

$$\{(k' + n' + k, n' + k) : k \in [n']\} \cup \{(k' + 2n' + 3, 2n' + 1)\}, \tag{107}$$

the following entries to $-1$

$$\{(k' + 2n' + 2, 2n' + 1)\} \cup \{(j - 1, 2n' + 1)\}, \tag{108}$$

and the remaining entries to 0.

We solely rely on the configuration of $D^{(r)}$ and the $b, A$ to match with the constraints and set $X_{jm+1}, X_{jm+2}, X_{j'm+3}, X_{j'm+4}$ to zero for all $j \in [m']$ and $j' \in [k']$. $\qquad\square$

The dimension $d$ of the feasibility problem depends polynomially on the problem dimension of the separability program parameters $m', n', k'$. Specifically, the reduction requires that

$$d \geq \max\left(k' + 2n' + 2,\ 1 + \max(m', k')\right).$$

This reduction preserves the scaling of the problem size, and thus the feasibility problem remains NP-hard with respect to its dimension $d$. In particular, the computational hardness increases with the problem dimension. This trend is also reflected empirically in Fig. 5, where the performance gap between LSA and the reparameterized model widens as $d$ increases.

Since it is potentially impossible to find a transformer parameter that achieves the global minimum of the reparameterized results, we observe the following performance boundaries for a single-layer LSA.

*Remark* C.15. We define a mapping $\psi$ that projects $X \in \mathbb{R}^{dm}$ to the parameter space:

$$\psi(X) = \operatorname{argmin}_{P,Q} \|\phi(P,Q) - X\|_2^2. \tag{109}$$

Here, $\psi$ finds a parameter set that maps to the closest point to $X$ under $\phi$. $\psi(X)$ is the preimage of $X$ under $\phi$, if such a preimage exists. Let $f^*$ be the global minimum of $f$. Then $\tilde{f}(X^*) \leq f^* \leq f(\psi(X^*))$. Let $P^*, Q^*$ denote the global minimizer corresponding to $f^*$. Since $\tilde{f}$ is strictly convex w.r.t $X \in \mathbb{R}^{dm}$, it follows that $\tilde{f}(X^*)$ is the lower bound for any $\tilde{f}(\phi(P,Q))$, including $f^* = f(P^*, Q^*) = \tilde{f}(\phi(P^*, Q^*))$. Therefore $\tilde{f}(X^*) \leq f^*$. Similarly, since $f^*$ is smaller than any $f(P, Q)$, we have $f^* \leq f(\psi(X^*))$.

*Example* C.16. As an example, for $d = 2$, $\mathbf{gg}^\top$ becomes

$$\begin{bmatrix} \mathsf{G}_{11}^2 & \mathsf{G}_{11}\mathsf{G}_{12} & \mathsf{G}_{11}\mathsf{G}_{13} & \mathsf{G}_{11}\mathsf{G}_{22} & \mathsf{G}_{11}\mathsf{G}_{23} & \mathsf{G}_{11}\mathsf{G}_{33} \\ & \mathsf{G}_{12}^2 & \mathsf{G}_{12}\mathsf{G}_{13} & \mathsf{G}_{12}\mathsf{G}_{22} & \mathsf{G}_{12}\mathsf{G}_{23} & \mathsf{G}_{12}\mathsf{G}_{33} \\ & & \mathsf{G}_{13}^2 & \mathsf{G}_{13}\mathsf{G}_{22} & \mathsf{G}_{13}\mathsf{G}_{23} & \mathsf{G}_{13}\mathsf{G}_{33} \\ & & & \mathsf{G}_{22}^2 & \mathsf{G}_{22}\mathsf{G}_{23} & \mathsf{G}_{22}\mathsf{G}_{33} \\ & & & & \mathsf{G}_{23}^2 & \mathsf{G}_{23}\mathsf{G}_{33} \\ & & & & & \mathsf{G}_{33}^2 \end{bmatrix} \tag{110}$$

(omitting the index-separating comma and the repeating entries in the lower half triangle).

After reparameterization, the objective function can be rewritten as

$$\tilde{f}(X) = \mathbb{E}_{\{x_i, y_i\}_{i=1}^{n+1}, p_{01}, p_{11}} \left[ \left( \sum_{j=1}^{2} \mathbf{g}^\top X^{(j)} x_{n+1,j} - y_{n+1} \right)^2 \right].$$

where $X \in \mathbb{R}^{12}$ denotes the concatenation of the two vectors $X^{(1)}, X^{(2)} \in \mathbb{R}^6$. The gradient of $\tilde{f}$ w.r.t. $X$ is

$$\nabla \tilde{f}(X) = \mathbb{E} \begin{bmatrix} (x_{n+1,1})^2 \mathbf{gg}^\top & x_{n+1,1} x_{n+1,2} \mathbf{gg}^\top \\ x_{n+1,2} x_{n+1,2} \mathbf{gg}^\top & (x_{n+1,2})^2 \mathbf{gg}^\top \end{bmatrix} X - \mathbb{E}[y_{n+1}(x_{n+1} \otimes \mathbf{g})].$$

We obtain the global minimizer $X^*$ by solving $\nabla \tilde{f}(X^*) = 0$:

$$X^{(1)^*} = \begin{bmatrix} -0.15 & 0.39 & 0.15 & 0.12 & 2.40 & -0.09 \end{bmatrix},$$
$$X^{(2)^*} = \begin{bmatrix} 0.07 & -0.19 & -0.07 & -0.06 & -1.20 & 0.04 \end{bmatrix}.$$

We project $X^{(1)}, X^{(2)}$ into the model parameter space.

Since the entires of $X^{(1)}$ are nonzero, we have $b_1 \neq b_2$.

To verify the derivation, we plot the loss function w.r.t $X_i$, indicating the global optimizer $X_i^*$ using red dashed line in Fig. 10. The theoretical global minimizer aligns with the lowest error.

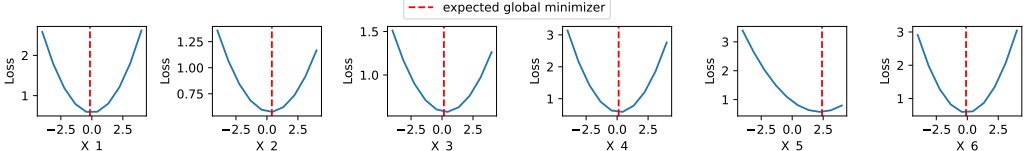

Figure 10: Loss function w.r.t. the first six parameters after reparameterization.

# D  Proof for Forward Pass as Multi-Objective Optimization (Theorem 4.1)

To demonstrate the equivalence between the forward pass and preconditioned gradient descent, we aim to express the iterative definition of LSA as an update of weight vectors, drawing inspiration from [19]. However, unlike their approach, our proof diverges because the update formula for LSA cannot be simplified due to the presence of nonzero entries in $b_l$.

**Theorem D.1** (***Theorem 4.1 restated***). *Consider the L-layer transformer parameterzed by* $b_l, A_l = -\begin{bmatrix} \bar{A}_l \\ a_l^\top \end{bmatrix}$ *where* $b_l \in \mathbb{R}^{d+1}, \bar{A}_l \in \mathbb{R}^{d \times d}, a_l \in \mathbb{R}^d$ *for* $l \in [L]$. *The forward pass of the transformer is equivalent to optimizing two groups of objectives* $R_1, R_2 : \mathbb{R}^d \to \mathbb{R}^{d+1}$. *Specifically, let* $y_{n+1}^{(l)}$ *be the bottom-right entry of the lth layer output. Then* $y_{n+1}^{(l)} = \langle w_l, x_{n+1} \rangle$ *where* $w_l$ *is iteratively defined as follows:* $w_0 = 0$ *and*

$$w_{l+1}^\top = w_l^\top - b_l^\top \left( \nabla R_1(w_l) \bar{A}_l + \nabla R_2(w_l) \begin{bmatrix} 0_{(d-1) \times d} \\ a_l^\top \end{bmatrix} \right) \tag{111}$$

$$\text{where } R_1(w) = \frac{1}{n} \sum_{j=1}^n \begin{bmatrix} -\langle w, x_j \rangle x_j \\ 0.5(y_j - \langle w, x_j \rangle)^2 \end{bmatrix}, \tag{112}$$

$$R_2(w) = \frac{1}{n} \sum_{j=1}^n \begin{cases} \begin{bmatrix} (-w_d(y_j - \langle w_{:d-1}, x_{j,:d-1} \rangle) + \frac{w_d^2}{2} x_{j,d}) x_j \\ \frac{1}{3x_{j,d}} (y_j - w^\top x_j)^3 \end{bmatrix} & \text{if } x_{j,d} \neq 0 \\ \begin{bmatrix} (-w_d(y_j - \langle w_{:d-1}, x_{j,:d-1} \rangle) + \frac{w_d^2}{2} x_{j,d}) x_j \\ -(y_j - \langle w_{:d-1}, x_{j,:d-1} \rangle)^2 w_d \end{bmatrix} & \text{if } x_{j,d} = 0 \end{cases}. \tag{113}$$

*In the above expression,* $w_{:d-1}, x_{j,:d-1}$ *denote the first* $d-1$ *entries of* $w$ *and* $x_j$, *respectively.*

*Proof.* Let $y_i^{(k)}$ denote the $(d+1)$-$i$ entry of the embdding $Z_k$ and $x_i^{(k)}$ is the first $d$ entries of the $i$th column in $Z_k$. Since the first $d$ rows of $P$ is zero, the first $d$ rows of $Z_k$ is the same as $Z_0$. Therefore $x_i^{(k)} = x_i^{(0)} = x_i, \forall i \in [n+1]$.

We define a mapping to represent applying $k$ transformer layers to the bottom right entry of an embedding matrix $Z_0$ with $[Z_0]_{d+1,n+1} = y$: $g(x, y, k) : \mathbb{R}^d \times \mathbb{R} \times \mathbb{Z} \to \mathbb{R}$. When $x = x_{n+1}, y = y_{n+1}^{(0)} = 0, g(x, y, k) = g(x, 0, k) = y_{n+1}^{(k)}$. We establish two claims for $g(x, y, k)$ when $x = x_{n+1}$.

**Claim 1:** $g(x, y, k) = g(x, 0, k) + y$. The equation implies that applying the transfomer $k$ times on $Z_0$ with $[Z_0]_{d+1,n+1} = y$ is equivalent to applying the transformer $k$ times on $Z_0'$ with $[Z_0']_{d+1,n+1} = 0$ and then add the resulting bottom-right entry with $y$.

By definition of LSA, the iterative definition of $y_i^{(k)}$ $(i \in [n+1])$ is given by:

$$y_i^{(k+1)} = y_i^{(k)} - b_k^\top \underbrace{\frac{1}{n} \sum_{j=1}^n \begin{bmatrix} x_j x_j^\top & y_j^{(k)} x_j \\ y_j^{(k)} x_j^\top & y_j^{(k)2} \end{bmatrix}}_{:= \mathsf{G}^{(k)}} A_k x_i. \tag{114}$$

Since $y_i^{(k)}$ is independent of $y_{n+1}^{(k')}$ for any $k'$, and $y_{n+1}^{(k)}$ depends on $y_{n+1}^{(k)}$ additively, one can show inductively that $g(x, y, k)$ and $g(x, 0, k)$ always differ by $y$, i.e., $g(x, y, k) = g(x, 0, k) + y$.

**Claim 2:** $g(x, 0, k)$ **is linear in** $x$. We prove the claim inductively. When $k = 0$, $g(x, 0, 0) = y_{n+1}^{(0)} - b_k^\top \mathsf{G}^{(k)} A_k x_{n+1}$ is linear in $x = x_{n+1}$. For $k \geq 0$, suppose $g(x, 0, k)$ is linear in $x$, then $g(x, 0, k+1) = y_{n+1}^{(k+1)} = y_{n+1}^{(k)} - b_k^\top \mathsf{G}^{(k)} A_k x_{n+1} = g(x, 0, k) - b_k^\top \mathsf{G}^{(k)} A_k x_{n+1}$. The first term $g(x, 0, k)$ is linear in $y$. The term $y_j^{(k)}$ with $j \neq n+1$ does not depend on $x_{n+1}$ according to Eq. 114. Hence $b_k^\top \mathsf{G}^{(k)} A_k x_{n+1}$ is also linear in $x_{n+1}$.

Combining the two claims, we have

$$g(x, y, k) = g(x, 0, k) + y = \langle \theta_k, x \rangle + y, \tag{115}$$

for some $\theta_k \in \mathbb{R}^d$ with $\theta_0 = 0$. One can copy the values in the $i$th column to the $n+1$th column and adopt the previous arguments to show that $g(x_i, y_i, k) = \langle \theta_k, x_i \rangle + y_i$. By substituting

$y_i = \langle \theta_k, x_i \rangle + y_i$ into Eq. 114, we have

$$y_{n+1}^{(k+1)} = y_{n+1}^{(k)} - \frac{1}{n}\sum_{j=1}^{n} b_k^\top \begin{bmatrix} x_j x_j^\top & (y_j + \theta_k^\top x_j)x_j \\ (y_j + \theta_k^\top x_j)x_j^\top & (y_j + \theta_k^\top x_j)^2 \end{bmatrix} A_k x_{n+1} \tag{116}$$

$$\Rightarrow \langle \theta_{k+1}, x_{n+1} \rangle = \langle \theta_k, x_{n+1} \rangle - \frac{1}{n}\sum_{j=1}^{n} b_k^\top \begin{bmatrix} x_j x_j^\top & (y_j + \theta_k^\top x_j)x_j \\ (y_j + \theta_k^\top x_j)x_j^\top & (y_j + \theta_k^\top x_j)^2 \end{bmatrix} A_k x_{n+1}. \tag{117}$$

Since the above equation holds for any $x_{n+1}$, we obtain

$$\theta_{k+1}^\top = \theta_k^\top - \frac{1}{n}\sum_{j=1}^{n} b_k^\top \begin{bmatrix} x_j x_j^\top & (y_j + \theta_k^\top x_j)x_j \\ (y_j + \theta_k^\top x_j)x_j^\top & (y_j + \theta_k^\top x_j)^2 \end{bmatrix} A_k$$

$$= \theta_k^\top - b_k^\top \underbrace{\frac{1}{n}\sum_{j=1}^{n} \begin{bmatrix} x_j x_j^\top \\ (y_j + \theta_k^\top x_j)x_j^\top \end{bmatrix}}_{\bar{G}_1 \in \mathbb{R}^{(d+1)\times d}} \bar{A}_k - \left( b_k^\top \frac{1}{n}\sum_{j=1}^{n} \underbrace{\begin{bmatrix} (y_j + \theta_k^\top x_j)x_j \\ (y_j + \theta_k^\top x_j)^2 \end{bmatrix}}_{\in \mathbb{R}^{d+1}} \right) a_k^\top$$

$$= \theta_k^\top - b_k^\top \frac{1}{n}\sum_{j=1}^{n} \begin{bmatrix} x_j x_j^\top \\ (y_j + \theta_k^\top x_j)x_j^\top \end{bmatrix} \bar{A}_k - b_k^\top \underbrace{\bar{G}_2}_{\in \mathbb{R}^{(d+1)\times d}} \begin{bmatrix} 0_{(d-1)\times d} \\ a_k^\top \end{bmatrix}, \tag{118}$$

where if $x_{j,d} \neq 0$, then

$$\bar{G}_2 := \begin{bmatrix} x_{j,1}\theta_{k,d}x_j & \cdots & \theta_{k,d}x_{j,d-1}x_j & (y_j + \theta_k^\top x_j)x_j \\ (y_j + \theta_k^\top x_j)^2 \frac{x_{j,1}}{x_{j,d}} & \cdots & (y_j + \theta_k^\top x_j)^2 \frac{x_{j,d-1}}{x_{j,d}} & (y_j + \theta_k^\top x_j)^2 \end{bmatrix}, \tag{119}$$

otherwise if $x_{j,d} \neq 0$, then

$$\bar{G}_2 := \begin{bmatrix} x_{j,1}\theta_{k,d}x_j & \cdots & x_{j,d-1}\theta_{k,d}x_j & (y_j + \theta_k^\top x_j)x_j \\ 2(y_j + \theta_k^\top x_j)x_{j,1}\theta_{k,d} & \cdots & 2(y_j + \theta_k^\top x_j)x_{j,d-1}\theta_{k,d} & (y_j + \underbrace{\theta_k^\top x_j}_{(i)})^2 \end{bmatrix}. \tag{120}$$

In the above expression, $\theta_{k,i}, x_{j,i}$ denote the $i$th element of $\theta_k, x_j$, respectively. We treat $b_k, \bar{A}_k, \begin{bmatrix} 0_{(d-1)\times d} \\ a_k \end{bmatrix}$ as preconditioners. We construct two sets of muli-objectives $\tilde{R}_1, \tilde{R}_2 : \mathbb{R}^d \to \mathbb{R}^{d+1}$ as follows. We drop the index for layer on $\theta$ and use $\theta_d$ to denote the $d$th entry of $\theta$. Let $x_{j,:d-1}, \theta_{:d-1}$ denote the first $d-1$ entries of the vectors $x_j, \theta$ respectively.

$$\tilde{R}_1(\theta) := \frac{1}{n}\sum_{j=1}^{n} \begin{bmatrix} x_{j,1}\theta^\top x_j \\ x_{j,2}\theta^\top x_j \\ \vdots \\ x_{j,d}\theta^\top x_j \\ 0.5(\theta^\top x_j + y_j)^2 \end{bmatrix} = \frac{1}{n}\sum_{j=1}^{n} \begin{bmatrix} \langle \theta, x_j \rangle x_j \\ 0.5(\langle \theta, x_j \rangle + y_j)^2 \end{bmatrix}, \tag{121}$$

$$\tilde{R}_2(\theta) := \frac{1}{n}\sum_{j=1}^{n} \begin{cases} \begin{bmatrix} (\theta_d(y_j + \langle \theta_{:d-1}, x_{j,:d-1} \rangle) + \frac{\theta_d^2}{2}x_{j,d})x_j \\ \frac{1}{3x_{j,d}}(y_j + \theta^\top x_j)^3 \end{bmatrix} & \text{if } x_{j,d} \neq 0 \\ \begin{bmatrix} (\theta_d(y_j + \langle \theta_{:d-1}, x_{j,:d-1} \rangle) + \frac{\theta_d^2}{2}x_{j,d})x_j \\ (y_j + \underbrace{\langle \theta_{:d-1}, x_{j,:d-1} \rangle}_{(i)})^2 \theta_d \end{bmatrix} & \text{if } x_{j,d} = 0 \end{cases}. \tag{122}$$

The derivation of $\tilde{R}_2(\theta)$ for the case $x_{j,d} = 0$ utilizes the fact that the terms marked by $(i)$ are equivalent: $\langle \theta, x_j \rangle = \langle \theta_{:d-1}, x_{j,:d-1} \rangle$ when $x_{j,d} = 0$.

Then $\nabla R_1(\theta) = \bar{G}_1$ and $\nabla R_2(\theta) = \bar{G}_2$ yield

$$\theta_{k+1}^\top = \theta_k^\top - b_k^\top \nabla \tilde{R}_1(\theta)\bar{A}_k - b_k^\top \nabla \tilde{R}_2(\theta) \begin{bmatrix} 0_{(d-1)\times d} \\ a_k^\top \end{bmatrix}$$

$$= \theta_k^\top - b_k^\top \left( \nabla \tilde{R}_1(\theta)\bar{A}_k + \nabla \tilde{R}_2(\theta) \begin{bmatrix} 0_{(d-1)\times d} \\ a_k^\top \end{bmatrix} \right). \tag{123}$$

By letting $w = -\theta$ and $R_i(w) = \tilde{R}_i(-\theta)$ ($i \in [2]$), we obtain the desired result. $\qquad\square$

The first $d$ terms in $R_1(w)$ $(-x_{j,k}w^\top x_j$ for $k \in [d])$ capture the interaction between the $k$-th state of $x_j$ and the linear combination of states determined by $w$, emphasizing their individual roles within the sequence. The final term, $(w^\top x_j - y_j)^2$ ensures alignment between the linear model's prediction and the target state $y_j$.

$R_2(w)$ emphasizes the role of $w_d$, since the first $d-1$ rows of the last preconditioning matrix are zeros. The first $d$ objectives in $R_2(w)$ scale the target state $y_j$ by $x_j$ and $w_d$, with additional quadratic terms like $\frac{w_d^2}{2}x_{j,d}$. These terms capture the influence of $w_d, x_{j,d}$, the alignment between $y_j$ and partial prediction $\langle w_{:d-1}, x_{j,:d-1}\rangle$. The final objective is in cubic penalty form $\frac{1}{3x_{j,d}}(y_j - w^\top x_j)^3$ when $x_{j,d} \neq 0$, magnifying sequences with smaller $x_{j,d}$. When $x_{j,d} = 0$, the penalty changes to a quadratic term $-(y_j - \langle w_{:d-1}, x_{j,:d-1}\rangle)^2 w_d$, focusing solely on aligning $y_j$ with the partial prediction based on $w_{:d-1}$. Furthermore, when $x_{j,d} = 0$, the final objective becomes convex if $w_d < 0$ and concave otherwise.

# E  Additional Related Works

A growing line of research investigates the structural and computational limitations of shallow transformer models. Rajaraman et al. [24] examine ICL on Markovian data and show that while 1-layer transformers may struggle to capture high-order dependencies, adding one or two layers enables accurate modeling of $k$-th order Markov processes. Olsson et al. [45] demonstrate that induction heads do not arise in 1-layer transformers, and that their emergence and the corresponding gains in ICL performance are only observed in models with two or more layers. Separately, Sanford et al. [37, 38] study the representational and computational limits of transformers from a theoretical perspective. While not directly focused on ICL, they characterize tasks that require deeper models or higher attention capacity to solve, and establish lower bounds by connecting transformer depth to parallel computation. Our work complements these studies from an optimization perspective, where we show that even when a global minimizer exists in a relaxed formulation, recovering a feasible parameterization in the 1-layer transformer space is NP-hard.

