# OpenReview forum: "Optimality and NP-Hardness of Transformers in Learning  Markovian Dynamical Functions"
_NeurIPS.cc/2025/Conference — NeurIPS 2025 poster_

### Official Review · Reviewer_88Bq · 2025-07-02

**Clarity:** 2
**Significance:** 4
**Originality:** 4
**Rating:** 5
**Confidence:** 3

**Summary:**

This paper studies the in-context learning (in)ability of transformers for Markovian dynamical functions where the first $d$ $x_i$ variables are drawn from a Markov process and the $y_i$ is the $d+1$ element of the Markov chain. Each context is a distinct set of transition probabilities. The authors show that under a reparameterization of one layer linear self attention, they can identify a global minimizer for the ICL population risk. They then show that it is NP hard for one to find a one layer transformer configuration that achieves a given reparameterized configuration $X_\star$. They do this by mapping the reparameterization to an NP hard bilinear optimization problem. They then show that deeper transformers with linear self attention can be interpreted as gradient descent on a mixture of two objective functions for any given context. They conclude with empirical simulations showing that performance improves as the depth of a model increases.

**Questions:**

1. Would the improvements to the performance on this task continue if you keep increasing the depth? Have the authors tried training models deeper than 3 layers? Is the global minimizing loss / Bayes error /  best possible accuracy computed?
2. Could the authors clarify how what the NP hardness result indicates about learnability of this ICL problem in one layer LSA? The authors state "the 1-layer transformer’s representational capacity may be insufficient to express dynamical functions induced by structured inputs, such as those governed by shared Markovian dynamics." I understand that finding a $P,Q$ for a given $X$ may be NP hard, but couldn't GD find a solution that may be very close to $X_\star$ in performance in principle? I am not an expert in this style of result so any additional clarification would be useful to me.

**Ethical Concerns:**

["NO or VERY MINOR ethics concerns only"]

**Limitations:**

The authors do a good job addressing potential limitations.

**Quality:**

3

**Strengths And Weaknesses:**

Strengths

1. This paper introduces a very novel and interesting setting for ICL that involves dense correlations across tokens.
2. The authors provide several interesting results about LSA transformers on this problem.
3. They also provide an interesting simulation in the experimental validation section.

Weaknesses
1. It is unclear if the NP hardness result necessarily implies that the one layer transformer can or cannot learn the task (see Questions).
2. The experiments are fairly limited, including those in the Appendix. More explanation of the solution structure in the deep case and the GD implementation would be helpful to the reader.

---

> ### Author Rebuttal · Authors · 2025-07-31
>
> We appreciate the reviewer’s encouraging comments as well as the suggestions for improving clarity and expanding the experimental section. We respond below to the specific questions about the NP-hardness and generalization to deeper layers.
>
> ## 1. NP-hardness interpretation
> > **W1:** It is unclear if the NP hardness result necessarily implies that the one layer transformer can or cannot learn the task.
> > **Q2:** Can gradient descent find a solution close to the global minimum performance?
>
> The NP-hardness result does not mean 1-layer transformer can’t learn Markov chains, but finding the best possible parameters is computationally hard.
> - **For 1-layer LSA:** As shown in Fig. 5 (Section B.1), as the sequence length d increases, the performance gap between the 1-layer LSA trained with gradient descent with 1000 epochs and the reparameterized optimum X* becomes larger. This aligns with our NP-hardness result, which suggests that recovering the global optimum becomes intractable as $d$ increases.
> - **For deeper LSA:** Fig. 3 in the main text shows that adding layers improves accuracy, suggesting that increased expressivity helps overcome the limitations we identify for shallow architectures. For the same task and number of training epochs, standard training can succeed in practice when the model class is sufficiently expressive to begin with.
>
> ## 2. Generalization to multi-layers
> > **W2**: Limited experiments, including those in the appendix. Lack of explanation of deep case solution structure and GD implementation
> - **Explanation of deep-case solution**.
>     - **Parameter assumption.** The parameter assumption in Eq. 11 requires certain entries in the LSA weights $P, Q$ to be nonzero, with their locations matching the global minimum structure in Theorems 3.3 and 3.4, though their values remain unrestricted. We then interpret the forward computation of this family of LSA parameters as iterative steps of preconditioned gradient descent. The emergence of multiple objective types is due to the nonzero entries in $P,Q$.
>     - **Empirical verification (Fig. 2).** To verify the multi-layer interpretation, we fixed the LSA parameter obtained by gradient descent training. The objectives are evaluated for this fixed LSA at each layer.  We observed that initially, the model makes progress on both $R_1$ and $R_2$, but beyond a certain depth, optimization deteriorates, all losses begin increasing. The non-monotonic trend is likely due to deeper LSA preconditioners  negatively affecting the performance.
>
> > **Q1:** Would the improvements to the performance on this task continue if you keep increasing the depth? Have the authors tried training models deeper than 3 layers? Is the global minimizing loss / Bayes error / best possible accuracy computed?
>
> We extended our experiments to deeper models and observed that performance continues to improve up to 6 layers. We fix the total number of prompts as 1000, number of in-context examples as 100 and consider binary Markov chains with prefix length $d=5$. We perform gradient descent for 5000 epochs for all layers. The accuracy and MSE w.r.t. # layers is shown below. These results support the intuition that additional layers can mitigate the expressivity limitation. We hypothesize that the decrease at 5 layers may be due to optimization instability. Nonetheless, the overall trend supports the benefit of increased depth.
>
> | # Layers | Accuracy   | MSE  |
> | --- | --- | --- |
> | 1 | 0.681 | 0.225 |
> | 2 | 0.687 | 0.216 |
> | 3 | 0.713 | 0.200 |
> | 4 | 0.747 | 0.188 |
> | 5 | 0.681 | 0.218 |
> | 6 | 0.784  | 0.166 |
> - **Baseline.** For the same dataset, the Bayes error is 0.315, computed by first estimating the transition kernel from in-context example and the first 5 tokens of the query sequence, then using this estimated kernel to make a Bayes-optimal prediction of the next token. The Bayes error is then computed as the fraction of prompts for which this prediction is incorrect.

---

> > ### Comment · Reviewer_88Bq · 2025-08-04
> >
> > I thank the authors for their detailed responses and new experiments varying the depth. I am impressed by the depth separation results for this Markovian dynamical function learning, both in theory and experiments. I stand by my positive evaluation of this work and support acceptance.

---

> > > ### Author Response · Authors · 2025-08-07
> > >
> > > Thank you for your time and thoughtful review. We sincerely appreciate your positive evaluation and insightful comments, which have helped strengthen the paper.

---

### Official Review · Reviewer_Fq3M · 2025-07-02

**Clarity:** 3
**Significance:** 3
**Originality:** 3
**Rating:** 4
**Confidence:** 3

**Summary:**

This paper investigates how transformers perform in-context learning when the data arises from structured Markovian dynamics, rather than i.i.d. distributions. Focusing on linear self-attention (LSA), the authors derive a closed-form global minimizer for short Markov sequences and show that finding transformer parameters that realize this optimum is NP-hard in general—highlighting a fundamental limitation of one-layer LSA models. They further show that multilayer LSA models can overcome this expressivity barrier by implicitly performing a preconditioned multi-objective optimization. The theoretical results are supported by empirical validation, offering a novel perspective on how architectural depth enables transformers to generalize across structured, temporally dependent inputs.

**Questions:**

I would consider increasing my score if most of the following questions are clearly addressed. Please also refer to the weaknesses section.

Questions:

- Your analysis assumes linear self-attention and binary first-order Markov chains. How do your results generalize to softmax attention, multi-head architectures, or higher-order/non-binary dynamics?
- The NP-hardness result is worst-case. Do you expect this computational hardness to appear in typical learning scenarios? Are there cases where parameter recovery remains tractable?
- The multi-layer analysis relies on a structured parameter space. Do real-world trained transformers actually follow this structure? Can you justify that the multi-objective interpretation holds more generally?
- The experiments focus on simple synthetic tasks. Can you share more details or actually main-text results for the GPT-2 experiments, and can you evaluate on more realistic sequence data and modern models?

**Ethical Concerns:**

["NO or VERY MINOR ethics concerns only"]

**Final Justification:**

After considering the rebuttal and discussion, I have decided to maintain my original evaluation. This paper offers a rigorous and well-presented theoretical analysis of in-context learning for Markovian dynamics, including an NP-hardness result for recovering the optimal one-layer LSA parameters and an interpretation of multi-layer models as performing preconditioned multi-objective optimization. The authors provided thoughtful clarifications and additional experiments. Nonetheless, several points remain only partially addressed: the generalization of the analysis beyond one-layer LSA and binary Markov chains is left for future work, the practical implications of the NP-hardness result are not fully explored, and the multi-layer interpretation depends on a structured parameterization that may not emerge in real-world settings. The contributions are technically solid and of theoretical interest, but the remaining scope limitations lead me to keep my borderline accept rating.

**Limitations:**

Yes.

**Paper Formatting Concerns:**

The itemize environment on the second page seems to be modified to preserve more space.

**Quality:**

3

**Strengths And Weaknesses:**

Strengths:

- The paper provides a rigorous analytical characterization of in-context learning for Markovian dynamical tasks.
- The authors prove that recovering the transformer parameters that achieve the optimal in-context performance on general Markovian functions is an NP-hard problem (Theorem 3.5). This is a notable finding that reveals a structural limitation of single-layer transformers, as there exists a gap between the optimal structured predictor and what a one-layer LSA can represent or learn in practice.
- The paper presents a novel interpretation of multi-layer transformers as performing a preconditioned multi-objective optimization in their forward pass (Theorem 4.1). In essence, a deeper LSA model is shown to implicitly minimize a main squared-loss objective while simultaneously maximizing several linear objectives.
- Rigor and clarity of the analysis overall.

Weaknesses:

- The theoretical analysis is confined to a highly simplified setting – one-layer linear self-attention models and first-order binary Markov chains. While this abstraction is necessary for analytical tractability, it limits the direct applicability of the results.
- The NP-hardness result, while theoretically important, is a worst-case complexity statement. It shows that exactly recovering the optimal parameter solution in the general Markovian setting is intractable, but this doesn’t necessarily mean that learning such tasks is impossible in practice, for example heuristic training or approximate solutions might perform well on typical instances. The paper could discuss more about the practical implications of this hardness.
- The multi-layer LSA analysis (Theorem 4.1) is conducted under a special parameterization that “mirrors” the single-layer optimum’s structure. Essentially, the authors restrict the model to a certain subspace of parameters (Eq. 11 in the paper) to derive the multi-objective interpretation. This raises a question of how general the insight is as real transformers trained without such constraints might deviate from this idealized behavior.
- While the paper does include experiments, they are relatively limited in scope. The main text only reports synthetic data results for small state spaces, and points to additional experiments with standard transformer architectures (GPT-2) in the appendix.

---

> ### Author Rebuttal · Authors · 2025-07-31
>
> We thank the reviewer for the constructive feedback and careful reading of our theoretical results. Below, we discuss the  relevance of our analysis to more general settings and  clarify the intuition behind our NP-hardness result and multi-objective formulation.
>
> ## 1. Restricted assumptions in problem setup
> > **W1**: Analysis assumes 1-layer linear attention and binary first-order Markov chains with limiting direct applicability to richer architectures or dynamics.
>
> - **Theoretical tractability.** We would like to point out that one-layer linear self-attention models are widely adopted architectures in prior ICL studies [1,2,3], enabling clear theoretical insights.
> - **Generalizability.** Moreover, the limitation result of 1-layer LSA is not restricted to the current data distribution, i.e., binary state space or memory order of Markov chains. As shown in Lemma C.8, the reparameterized global optimum exists for Markov chains with arbitrary state spaces and trajectory lengths, yet shallow Transformers still fail to represent this optimum.
>
> > **Q1**: How do results generalize to softmax attention, multi-head architectures, or higher-order/non-binary dynamics?
> - **Scope of our study.** Our analysis focuses on LSA and first-order binary Markov dynamics, which offer an analytically tractable setting to study expressivity and optimization barriers in ICL.
> - **Novelty of our analysis.** To the best of our knowledge, this work takes an initial step toward understanding how ICL models handle structured function classes. Our work provides a theoretical lens by framing expressivity as an optimization problem, explaining why shallow models struggle to learn certain function classes in context.
> - **Extensibility.** This line of analysis offers a potential starting point for understanding expressivity limitations in broader function classes.  Extending our framework to cover these richer settings is an exciting direction we hope to pursue.
>
> ## 2. NP-hard result interpretation
> > **W2 & Q2**: Heuristic training or approximate solutions may work well in typical cases. Will computational hardness show up in typical learning scenarios? Any tractable cases?
> - **For 1-layer LSA:** As shown in Fig. 5 (Section B.1), as the sequence length d increases, the performance gap between the 1-layer LSA trained with gradient descent with 1000 epochs and the reparameterized optimum X* becomes larger. This aligns with our NP-hardness result, which suggests that recovering the global optimum becomes intractable as $d$ increases.
> - **For deeper LSA:** Fig. 3 in the main text shows that adding layers improves accuracy, suggesting that increased expressivity helps overcome the limitations we identify for shallow architectures. For the same task and number of training epochs, standard training can succeed in practice when the model class is sufficiently expressive to begin with.
> - **Tractable cases:** Theorems 3.3 and 3.4 show that for d=1, the inverse mapping (b, A) that achieves a given X* can be computed in closed form. However, for larger d, $\phi(b, A)$ becomes increasingly entangled due to bilinear interactions, and recovering parameters consistent with a given X* becomes intractable in general (Theorem 3.5).
>
> ## 3. Generality of multi-Layer analysis
> > **W3**: Limited generality of the multi-layer analysis due to structured parameterization.
> > **Q3**: Do real-world trained transformers actually follow this structure? Can you justify that the multi-objective interpretation holds more generally?
> - **Parameter assumption justification.** The parameter assumption in Eq. 11 requires certain entries in the LSA weights $P, Q$ to be nonzero, with their locations matching the global minimum structure in Theorems 3.3 and 3.4, though their values remain unrestricted. This construction allows us to interpret the forward computation as iterative steps of preconditioned gradient descent, where the emergence of multiple objectives arises from the structured sparsity in $P$ and $Q$.
> - **Generality.** Our analysis provides an analytically tractable case that highlights how deeper layers can improve performance by coordinating optimization across objectives using each layer's parameters. While we do not claim that real-world trained transformers follow this specific parameterization, we hope this controlled setting offers some intuition that may transfer to more realistic architectures.
>
> ## 4. Experimental scope
> > **W4**: Limited experimental scope and missing main-text results for larger models
> > **Q4**: Can you share more details or actually main-text results for the GPT-2 experiments, and can you evaluate on more realistic sequence data and modern models?
>
> Due to space constraints, we included the GPT-2 experiments in Appendix B.2. These experiments are designed to complement our theoretical analysis by evaluating how deeper, nonlinear transformer models perform on the same ICL task.
> - **GPT-2 Experiments.** We compare GPT-2 variants (with 3, 6, and 12 layers, respectively) on our synthetic benchmark. Fig. 6 shows deeper models consistently achieve higher accuracy in predicting the final token of a Markov chain, confirming the theoretical insight that multi-layer attention improves performance. Notably, performance with 6 and 12 layers becomes comparable to classical models like SVM and logistic regression and outperforms 3-NN baselines.
>     - **Attention maps.** To probe model behavior, we visualize attention maps across layers. Fig. 7 shows the emergence of block-wise attention, with deeper layers increasingly focusing on structurally similar in-context sequences governed by the same transition kernel.
> - **Additional experiments.** We extended our experiments to deeper models and observed that performance continues to improve up to 6 layers. We fix the total number of prompts as 1000, number of in-context examples as 100 and consider binary Markov chains with prefix length $d=5$. We perform gradient descent for 5000 epochs for all layers. The accuracy and MSE w.r.t. # layers is shown below. These results support the intuition that additional layers can mitigate the expressivity limitation. We hypothesize that the decrease at 5 layers may be due to optimization instability. Nonetheless, the overall trend supports the benefit of increased depth.
>
> | # Layers | Accuracy   | MSE  |
> | --- | --- | --- |
> | 1 | 0.681 | 0.225 |
> | 2 | 0.687 | 0.216 |
> | 3 | 0.713 | 0.200 |
> | 4 | 0.747 | 0.188 |
> | 5 | 0.681 | 0.218 |
> | 6 | 0.784  | 0.166 |
>
> - **Relationship to modern LLMs phenomenon.** Empirical works [4] have observed the emergence of ICL behavior in deeper transformers trained on language datasets, up to 40 layers and 13B parameters, while such capabilities are largely absent in shallow models with fewer than 8 layers. Our work provides a theoretical lens for this type of observation by framing expressivity as an optimization problem, explaining why shallow models struggle to represent certain function classes in context.
>
>
> **References**
>
> [1] Kwangjun Ahn, et al., Transformers learn to implement preconditioned gradient descent for in-context learning. NeurIPS 2023.
>
> [2] Ruiqi Zhang, et al., Trained Transformers Learn Linear Models In-Context. J. Mach. Learn. Res. 25: 49:1-49:55 2024.
>
> [3] Jianhao Huang, et al., Transformers Learn to Implement Multi-step Gradient Descent with Chain of Thought. ICLR 2025.
>
> [4] Catherine Olsson et al., In-context Learning and Induction Heads. 2022

---

> > ### Comment · Reviewer_Fq3M · 2025-08-03
> >
> > Thank you to the authors for the detailed and thoughtful response. I appreciate the clarifications and the additional experiments, particularly the GPT-2 results.
> >
> > That said, several concerns remain only partially addressed. In particular, the generalization beyond one-layer LSA and binary Markov settings is left for future work, the practical implications of the NP-hardness result are not fully explored, and the multi-layer analysis relies on a structured parameterization that may not hold in practice. The experimental validation, while helpful, remains limited to synthetic tasks.
> >
> > Based on the rebuttal and the current scope of the work, I will maintain my original score.

---

> > > ### Author Response · Authors · 2025-08-07
> > >
> > > Thank you for your constructive feedback. We appreciate your acknowledgment of the additional experiments.
> > >
> > > We would like to respectfully clarify some remaining concerns:
> > > - **"generalization beyond one-layer LSA and binary Markov settings":**  While our setup adopts simplifying assumptions, these follow well-established practices in theoretical ICL research [1-6]. Such scientific models are widely adopted in theoretical analysis of the principles of the attention modules, for instance, single-layer transformers have been extensively studied across numerous theoretical works [1-5] because they capture the essential attention mechanisms while remaining analytically tractable.
> > > - **"practical implications of the NP-hardness result":** Our NP-hardness result is consistent with both existing theoretical [7] and empirical [8] works on shallow transformer limitations, offering a new theoretical lens for why shallow attention struggles with complex dependencies.
> > > - **"multi-layer analysis with structured parameterization":**  The parameterization assumptions (Eq. 11), widely adopted in theoretical analyses [2,4,9], enable analytical tractability while preserving essential attention mechanisms. In particular, the multi-layer analysis helps reveal how transformers adapt to in-context examples of structured functions with non-iid inputs by balancing multiple objectives. Moreover, this parameterization configuration (Eq. 11) utilizes more attention parameters that were previously restricted to zero in prior works [2,4,9], revealing more of the attention mechanism's expressivity.
> > > - **"experimental validation on synthetic tasks":** As the first systematic attempt to understand these limitations theoretically, our synthetic validation provides the necessary foundation for future extensions to more complex settings.
> > >
> > > We sincerely thank you again for your valuable comments, which have helped improve our work.
> > >
> > > **References**
> > >
> > > [1] Johannes von Oswald et al., Transformers Learn In-Context by Gradient Descent. ICML 2023
> > >
> > > [2] Ruiqi Zhang, et al., Trained Transformers Learn Linear Models In-Context. J. Mach. Learn. Res. 25: 49:1-49:55 2024
> > >
> > > [3] Arvind V. Mahankali et al., One Step of Gradient Descent is Provably the Optimal In-Context Learner with One Layer of Linear Self-Attention. ICLR 2024
> > >
> > > [4] Kwangjun Ahn et al., Transformers learn to implement preconditioned gradient descent for in-context learning. NeurIPS 2023
> > >
> > > [5] Hongkang Li et al., How Do Nonlinear Transformers Learn and Generalize in In-Context Learning? ICML 2024
> > >
> > > [6] Ashok Vardhan Makkuva et al., Local to Global: Learning Dynamics and Effect of Initialization for Transformers. NeurIPS 2024
> > >
> > > [7] Clayton Sanford et al., Representational Strengths and Limitations of Transformers. NeurIPS 2023
> > >
> > > [8] Catherine Olsson et al., In-context Learning and Induction Heads. 2022
> > >
> > > [9] Yingcong Li et al., Gating is Weighting: Understanding Gated Linear Attention through In-context Learning, 2025

---

### Official Review · Reviewer_SbDi · 2025-07-02

**Clarity:** 3
**Significance:** 3
**Originality:** 2
**Rating:** 4
**Confidence:** 3

**Summary:**

The paper studies how do transformers in-context learning markovian chains by proving that there exists (reparametrized) transformer parameters which are global optima of the objective loss function. Additionally, it shows that for an actual one-layer LSA to achieve the reparametrized parameter is NP-hard. Moreover, it demonstrates that an n L-layer LSA can be interpreted as preconditioned multi-objective optimization.

**Questions:**

Q1: I do not fully comprehend the motivation behind studying the ICL ability on Markov chains. If the authors want to study non IID tasks, dynamical systems would have been enough, so could the authors clarify on the necessity to study Markov chains?

Q2: While Theorem 3.3 & 3.4 states the global minima under the reparametrized objective, Theorem 3.5 admits the NP-hardness to reconstruct transformer parameters from reparametrization. Does it mean that a one-layer transformer can or cannot learn Markov chain?

Q3: For the multi-layer setting (Theorem 4.1), why does the $(x,y)$ pair suddenly change to linear regression, and what does the two objectives $R_1$ and $R_2$ mean from a high level?

**Ethical Concerns:**

["NO or VERY MINOR ethics concerns only"]

**Final Justification:**

During the rebuttal period my main questions are solved and it appears that I have misunderstood the data representation in Q3 and thank you for the clarification. Also I have surveyed works on the explanation of ICL Markovian process and admits its significance.  Overall the paper is a good and natural extension from previous works of in-context gradient descent on linear regression to Markov chains.

**Limitations:**

yes

**Quality:**

3

**Strengths And Weaknesses:**

## Strengths

1. The paper studies the ICL ability of transformers on Markov chains which distinguishes itself with previous works on IID tasks like linear regression.

2. The theoretical aspect of the paper is solid, and some experiments are provided to support the conclusions despite being a theoretical paper.

## Weaknesses

Please see Questions.

---

> ### Author Rebuttal · Authors · 2025-07-31
>
> We are grateful to the reviewer for raising insightful questions. We address the key concerns about the role of Markov chains,  the interpretation of our NP-hardness result and multi-layer analysis in the following responses.
>
> ## 1. Setup clarification
> > **Q1**: Why study Markov chains instead of general dynamical systems for non-i.i.d. ICL
>
> Thank you for your suggestion on general dynamical system models. Indeed, it has been our general purpose to replace IID model with a more realistic and structured stochastic dynamical system model for analyzing transforms whose inputs and outputs can be characterized as probability measures.
> - **Modeling power of Markov models.**  Markov processes, due to its strength in modelling capability/flexibility and extensive reservoirs of theory and technical methods developed, has been an important model for stochastic dynamical systems, see e.g. references [1] and [2] for their treatment in widely used textbooks in the area. Hence, it is natural to introduce a Markov chain, a Markov processes with discrete time and finite state space, in this study. This choice provides a controlled way to examine how Transformers approximate latent temporal structure, making it a natural step toward understanding ICL on more general function classes.
>
> ## 2. Global minima to NP-hard results
> > **Q2**: While Theorem 3.3 and 3.4 states the global minima under the reparametrized objective, Theorem 3.5 admits the NP-hardness to reconstruct transformer parameters from reparametrization. Does NP-hardness mean 1-layer transformers can’t learn Markov chains?
>
> The NP-hardness result does not mean 1-layer transformer can’t learn Markov chains, but finding the best possible parameters is computationally hard in the worst-case.
> - **Tractable vs NP-hard case.** Theorems 3.3 and 3.4 provide a closed-form solution for recovering (b, A) from X* when d = 1, but this tractability breaks down for larger d, where the bilinear structure of $\phi(b, A)$ induced by the 1-layer LSA becomes more complex.
> - **NP-Hardness' implication for performance.** As shown in Fig. 5 (Section B.1), when the sequence length $d$ increases, the performance gap between the 1-layer LSA trained with gradient descent (1K epochs) and the reparameterized optimum $X^*$ becomes larger. While the reparameterized model achieved 95% accuracy for $d=10$, the corresponding 1-layer LSA trained with 1K epochs only reached ~70%. This aligns with our NP-hardness result, which suggests that recovering the global optimum becomes intractable as $d$ increases.
>
> ## 3. Multi-objective interpretation
> > **Q3**: In Theorem 4.1, why does the forward pass reduce to a linear regression problem, and what do the two objectives R_1 and R_2 represent at a high level?
> - **Linear model origin.** As detailed in Appendix Section D, we show that for an $L$-layer LSA, the final prediction $\hat{y}\_{n+1}^{(L)}$ can be expressed as a linear function of $x\_{n+1}$, i.e., $\hat{y}\_{n+1}^{(l)} = \langle w\_{l}, x\_{n+1} \rangle$. This holds for any prompt instance and follows directly from unrolling the forward pass. Given this linear form, we analyze how $w_l$ evolves across layers and show that it follows a preconditioned gradient descent path on two sets of objectives, $R_1$ and $R_2$.
> - **Objective interpretation.** At a high level, $R_1$ captures standard regression behavior, encouraging the linear model to align with the training data through squared error and weighted contributions from each input coordinate. $R_2$ introduces a cubic-order correction that places additional emphasis on the last input dimension.
>
> **References**
>
> [1] L. Arnold, Random dynamical systems, Springer, 1998.
>
> [2] R. Bhattacharya and M. Majumdar, Random dynamical systems: theory and applications, CUP, 2007.

---

> > ### Comment · Reviewer_SbDi · 2025-08-05
> >
> > Thank you for your response. My main questions are solved and it appears that I have misunderstood the data representation in Q3 and thank you for the clarification. Also I have surveyed works on the explanation of ICL Markovian process and admits its significance. Thus I will raise my score to 4.

---

> > > ### Author Response · Authors · 2025-08-07
> > >
> > > We are grateful for your thoughtful feedback and for taking the time to revisit our work. We’re glad that our clarifications regarding the problem setup were helpful, and we truly appreciate your recognition and the reevaluation.

---

### Official Review · Reviewer_MTmD · 2025-07-03

**Clarity:** 3
**Significance:** 2
**Originality:** 2
**Rating:** 4
**Confidence:** 4

**Summary:**

The paper investigates how transformers exhibit in-context learning (ICL) when trained on data generated from Markovian processes. It also identifies the global minima for such data sources in the case of a one-layer linear self-attention model. Additionally, the authors demonstrate that recovering the transformer's parameters could be NP-hard. Finally, they show that multiple layers can be interpreted as preconditioned gradient descent operating with multiple objectives.

**Questions:**

**Questions:**

1. Could the authors please clarify what the solutions in Theorem 3.3 and subsequently in Theorem 3.4 represent? For example, assuming the data generation process described between lines 133–139, it seems that estimating the correct downstream output, even in a regression settings, should ideally involve predicting the mean (if I’m not wrong). Is that what the solution actually learns? If not, could the discrepancy stem from the formulation, the nature of the solution, or some other factor? A clearer interpretation would help in understanding the practical implications of these results.

2. In the proof of Theorem 3.3 (and by extension Theorem 3.4), the authors simplify the LSA considerably and eventually reduce the problem to one of linear regression. While this helps in obtaining a tractable solution, the actual optimization landscape of $P$ and $Q$ appears quite different from the simplified setting (even if the global optima coincide). Understanding the behavior of the original optimization, particularly in terms of $P$ and $Q$, could provide additional insights. Would it be possible to characterize the loss landscape with respect to $P$ and $Q$ more directly? Moreover, how might the optimization dynamics change in the original setup compared to the simplified one?

3. I found the discussion around the problem being NP-hard a bit difficult to follow. It would be helpful if the authors could reiterate the main arguments more clearly. Also, please correct me if I'm misunderstanding, but the proof seems to present a generalization of Theorem 3.3 with a larger number of parameters. If that's the case, is the idea that we can reduce the problem to solving a linear regression, much like in Theorem 3.3, but in order to recover $P$ and $Q$, we ultimately need to solve a system of polynomial equations (and solving such systems is known to be NP-hard, as discussed in [E])? If this interpretation is correct, wouldn’t that offer a more straightforward sketch of the hardness argument?

4. As a follow-up to Problem 3, I remain a bit skeptical about the result. If my interpretation of why Problem 3 is NP-hard is accurate, then it seems that the hardness stems from the specific way the authors define optimality. I'm asking this because many problems in machine learning are technically NP-hard due to non-convexity [F], yet we still manage to find approximate solutions using gradient-based methods in practice. So even if the problem is NP-hard in the worst case, is it possible to converge to a reasonable solution using standard descent algorithms? Some clarification on this point would be helpful.

5. Could the authors elaborate on what the multiple objectives in Theorem 4.1 are, and what they intuitively represent? Understanding the nature and interpretation of these objectives would help clarify the broader implications of the result. Additionally, it would be helpful if the authors could walk through the intuitions behind the explanations outlined in lines 328–332


**Note:** I find the analysis in this work to be interesting. However, my main concern is regarding the applicability of the results. Much of the analysis appears to depend on very simplified settings, and the assumptions made are quite strong and seem to diverge significantly from practical scenarios. Based on these concerns, I am currently assigning a lower score. That said, I am open to revisiting my evaluation during the rebuttal phase, depending on the authors' clarifications and the views expressed by other reviewers.

**Minor Typos:**

1. Line before line 996 - the underbace should not be over the $1/n$ term.

2. Line 1045 - missing bracket in $U(0, 1)$

---

**References:**

[A] Alberto Bietti, Vivien Cabannes, Diane Bouchacourt, Herve Jegou, and Leon Bottou. Birth of a Transformer: A Memory Viewpoint. In Thirty-seventh Conference on Neural Information Processing Systems, 2023.

[B] Ezra Edelman, Nikolaos Tsilivis, Benjamin Edelman, Eran Malach, and Surbhi Goel. The Evolution of Statistical Induction Heads: In-Context Learning Markov Chains. In Advances in Neural Information Processing Systems, 37:64273–64311, 2024.

[C] Eshaan Nichani, Alex Damian, and Jason D. Lee. How Transformers Learn Causal Structure with Gradient Descent. In Forty-first International Conference on Machine Learning, 2024. URL

[D] Nived Rajaraman, Marco Bondaschi, Ashok Vardhan Makkuva, Kannan Ramchandran, and Michael Gastpar. Transformers on Markov Data: Constant Depth Suffices. In Advances in Neural Information Processing Systems, 37:137521–137556, 2024.

[E] https://bpb-us-e1.wpmucdn.com/sites.harvard.edu/dist/7/800/files/2024/10/spring2012-st12-scribe-lect14.pdf

[F] Blum, Avrim, and Ronald Rivest. "Training a 3-node neural network is NP-complete." Advances in neural information processing systems 1 (1988).

**Ethical Concerns:**

["NO or VERY MINOR ethics concerns only"]

**Final Justification:**

I thank the authors for their comments and efforts throughout the rebuttal process. After carefully weighing the pros and cons, I have decided to increase my rating from **3** to **4**.

**Pros:**
1. The paper is generally well-written and includes a substantial number of experimental results.
2. The authors did an excellent job during the rebuttal in clarifying my concerns, particularly regarding the proofs and the NP-hardness result.

**Cons:**
1. The setting feels somewhat artificial and does not align well with common practices for training on sequential data.
2. To the best of my knowledge, the paper does not address learning dynamics, which is a concern given the reliance on the NP-hardness argument.
3. The multi-objective argument supporting Theorem 4 is relatively weak, and the preconditioner argument is also unconvincing.

**Overall Assessment:**

Having said this, making progress in theoretical machine learning is inherently challenging and is also important if we are to deepen our understanding of large language models in the future. Moreover, to reiterate, the authors did a phenomenal job during the rebuttal, addressing concerns with clarity and professionalism. Hence, I recommend acceptance.

**Limitations:**

None.

**Paper Formatting Concerns:**

No.

**Quality:**

3

**Strengths And Weaknesses:**

**Strengths:**

1. The paper tries to tackle a very interesting problem which is ICL on Markov chains.

2. The authors have tried to contextualize their work very well with regards to other work.

3. The authors have also tried to extend the conventional regression-based theory for ICL tasks to a new domain which is Markov chains and this indeed is a challenging problem.

**Weaknesses:**
1. I feel somewhat uncertain about the applicability of this work. The problem setting is certainly interesting, especially in light of prior studies such as [A], [B], [C], and [D], which explore similar questions using the actual architectures employed by transformers in practice. In contrast, the current setup comes across as a bit contrived. The paper appears to build on the idea that some existing studies interpret in-context learning (ICL) through regression-based tasks and then explores how this perspective might extend to Markov chains. However, when focusing on Markov chains, particularly in the context of ICL, the primary objective typically involves next-token prediction. In that regard, a more natural setup already appears in works like [A], [B], [C], and [D], which the authors have appropriately cited.

2. The authors have clearly invested substantial effort in characterizing optimal solutions. However, they introduce several simplifications that make it harder to assess how applicable the findings are in practice. Additionally, the paper does not clearly convey the intuitive meaning of the global solution. Please see the questions for a follow-up

3. The paper seems to leave out the intuitions behind many of its theorems and propositions. While this is not necessarily a shortcoming, it gives the impression that the focus leans more toward mathematical formalism than toward building a qualitative understanding of the results. For instance, in Theorem 4, prior work such as [A], [B], [C], and [D] might lead one to expect that adding multiple transformer layers would yield a counting-based estimator (even though the settings are different). In this setup, however, it's not immediately clear why multiple layers of LSA would result in solving a multi-objective optimization problem. It would be helpful to develop more intuition around why this behavior arises and what gives rise to the presence of multiple objectives.

---

> ### Author Rebuttal · Authors · 2025-07-31
>
> We sincerely thank the reviewer for the thoughtful and detailed feedback. We aim to clarify the concerns including the motivation behind our design choices and the interpretation of our results below.
>
> ## 1. Setup clarification
> > **W1**: Concerns about the applicability of the problem setup compared to prior works [A] [B] [C] [D] that study next-token prediction for Markov chains.
> - **Relationship with prior studies**
>     - **Extending regression-style tasks.** We extend the ICL framework applied in [1–7] to study a new class of functions that has not been previously considered. In these works, the underlying function class is often linear and the input data is assumed to be i.i.d. and zero-centered. In contrast, we focus on settings where the input-output relationship is governed by a dynamical function with non-iid input.
>     - **Differences within Markov chain studies.** While studies in [A-D] (as cited by the reviewer) focus on next token prediction for probabilistic sequences, they also extend to model dependence between many tokens and within sequences with different Markov chain mechanisms, such as higher order Markov chains and Markov chains with causality.
>     - **Incomparable approaches.** We note that [1–7] and [A–D] approach the ICL problem from distinct perspectives, and their setups are not directly comparable. Specifically, [A–D] aim to predict the next token in a sequence whereas our goal is to recover a function using in-context examples.
> - **Reusable framework for probing expressivity.**  We offer a general approach to quantify and understand the limitation of 1-layer LSA. We reparameterize the LSA objective $f$ into a linear model $\tilde{f}$ over an enlarged input space (Eq. 8) and characterize its global optimum X* (Lemma 3.2). We then show that bridging the gap between 1-layer LSA and the best achievable predictor under $\tilde{f}$ is computationally hard (Theorem 3.5). This analysis is applicable to other ICL function classes that induce a dense structure on the optimum X*, where most entries in X* are nonzero, which might lead to NP-hardness in finding the corresponding optimal LSA parameters. This insight helps clarify what kinds of structured inference tasks that are challenging for shallow LSAs.
>
> ## 2. Global minima and reparameterization
>
> > **W2**: Unclear interpretation of global minimum and its relation to mean prediction.
> > **Q1**: What the solutions in Theorem 3.3 and 3.4 represent?
> - **Role of global min derivation.**  Theorems 3.3 and 3.4 characterize the global minimum of the 1-layer LSA in the tractable case $d=1$. They show that mapping from a dense reparameterized optimum X* to transformer parameters (b*,A*) results in dense solutions, in contrast to the sparse structures typical in ICL for linear tasks, thereby motivating the construction of a broader parameter class (Eq.11) for which we derive the multi-layer LSA interpretation.
> - **Empirically,** we showed in Fig. 5 that for binary state space with 100 in-context examples and $d=1$, the accuracy of this global minimum is ~60%. This relatively low accuracy is due to the limited input dimension, and it improves as $d$ increases.
>
> > **W2**:  Introduced several simplifications (reparameterization) that make it harder to assess how applicable the findings are in practice.
> > **Q2**: Is it possible to characterize the loss landscape more directly? Moreover, how might the optimization dynamics change in the original setup compared to the simplified one?
> - **Usage of reparameterization.** The reparameterization simplifies the optimization while preserving expressivity, as the linear model fully captures the 1-layer LSA. Such technique has been applied in recent studies [5,8] to study the loss landscape for  ICL of linear tasks.
>     - **New discovery under this reparameterization.** Under linear tasks, the global minimum of the reparameterization can be mapped back to the transformer parameter space. For more complex function class and non-iid input data, finding the corresponding transformer parameters becomes NP-hard, which sets our task apart from their setup.
> - **Impact on training dynamics.** The reparameterized objective $\tilde{f}$ is strictly convex (Lemma 3.1), ensuring convergence to its unique global optimum via gradient descent. In contrast, the original LSA objective $f(b, A)$ is non-convex due to bilinear parameter coupling, which can lead to suboptimal convergence depending on initialization. Empirically, we observe that training over $\tilde{f}$ reaches lower loss more quickly, likely because it avoids saddle points and plateaus present in $f$’s landscape.
>
>
> ## 3. NP-Hardness interpretation
> > **Q3**: NP-hard proof unclear; appears to generalize Theorem 3.3 with large $d$; why not directly reduce to known NP-hard polynomial system (e.g., [E])?
>
> We refer the reviewer to our response to W1 for a full walkthrough of the reparameterization-to-hardness chain.
> - **Reduction justification.** The NP-hardness result in [E] (as cited by the reviewer) is established for general polynomial systems. Since the bilinear system in Theorem 3.5  is a subclass of the general system, this result cannot be directly inherited. Therefore, we construct a dedicated reduction specific to our setting.
>
> > **Q4**: Is NP-hardness due to how optimality is defined? Many problems are NP-hard due to non-convexity and gradient-based methods help find approximate solutions in practice. Can standard training provide a reasonable solution even if the problem is NP-hard in the worst-case?
> - **Theoretical perspective:**
>     - The NP-hardness does not arise from the way optimality is defined, but from the non-convexity of the 1-layer LSA. The difficulty of solving the equation $\phi(b, A) = X$ stems from $\phi$ is bilinear, i.e., it couples the parameters b and A multiplicatively. This bilinear structure arises directly from the 1-layer LSA architecture, where the forward computation depends linearly on the product terms $b_i A_{j,k}$, rather than separately on $b_i$ and $A_{j,k}$.
> - **Empirical perspective:**
>     - **For 1-layer LSA:** As shown in Fig. 5 (Section B.1), as the sequence length d increases, the performance gap between the 1-layer LSA trained with gradient descent with 1000 epochs and the reparameterized optimum X* becomes larger. This aligns with our NP-hardness result, which suggests that recovering the global optimum becomes intractable as $d$ increases.
>     - **For deeper LSA:** Fig. 3 in the main text shows that adding layers improves accuracy to 80~90%, suggesting that increased expressivity helps overcome the limitations we identify for shallow architectures.
>     - **Modern LLMs.** Empirical works [9] have observed the emergence of ICL behavior in deeper transformers trained on language datasets, up to 40 layers and 13B parameters, while such capabilities are largely absent in shallow models with fewer than 8 layers. Our work provides a theoretical lens for this type of observation by framing expressivity as an optimization problem, explaining why shallow models struggle to represent certain function classes in context.
>
> ## 4.  Multi-Layer behavior
> > **W3**: Lack of intuitive explanation for the multi-layer forward pass and the emergence of multiple objectives in Theorem 4.
> > **Q5**: What do the  objectives in Theorem 4.1 represent intuitively, and how does the observed depth-dependent optimization behavior arise?
> - **Emergence of multiple objectives.** The parameter assumption in Eq. 11 requires certain entries in the LSA weights $P, Q$ to be nonzero, with their locations matching the global minimum structure in Theorems 3.3&3.4, though their values remain unrestricted. This construction allows us to interpret the forward computation as iterative steps of preconditioned gradient descent, where the emergence of multiple objectives arises from the structured sparsity in $P$ and $Q$.
> - **Interpretation.** At a high level, $R_1$ captures standard regression behavior, encouraging the linear model to align with the training data through squared error and weighted contributions from each input coordinate. $R_2$ introduces a cubic-order correction that places additional emphasis on the last input dimension.
> - **Explanation of depth-dependent behavior (Fig.2).** To verify the multi-layer interpretation, we fixed the LSA parameter obtained by gradient descent training. The objectives are evaluated for this fixed LSA at each layer.  We observed that initially, the model makes progress on both $R_1$ and $R_2$, but beyond a certain depth, optimization deteriorates, all losses begin increasing. The non-monotonic decreasing is likely due to deeper LSA preconditioners  negatively affecting the performance.
>
> **Minor typos.**
> Thank you for pointing out these typos. We will correct the misplaced underbrace and add the missing bracket in the final version.
>
> ---
> **References**
>
> [1] Johannes von Oswald et al., Transformers Learn In-Context by Gradient Descent. ICML 2023
>
> [2] Ruiqi Zhang, et al., Trained Transformers Learn Linear Models In-Context. J. Mach. Learn. Res. 25: 49:1-49:55 2024
>
> [3] Arvind V. Mahankali et al., One Step of Gradient Descent is Provably the Optimal In-Context Learner with One Layer of Linear Self-Attention. ICLR 2024
>
> [4] Kwangjun Ahn et al., Linear attention is (maybe) all you need (to understand Transformer optimization). ICLR 2024
>
> [5] Kwangjun Ahn et al., Transformers learn to implement preconditioned gradient descent for in-context learning. NeurIPS 2023
>
> [6] Yu Bai et al., Transformers as Statisticians: Provable In-Context Learning with In-Context Algorithm Selection. NeurIPS 2023
>
> [7] Hongkang Li et al., How Do Nonlinear Transformers Learn and Generalize in In-Context Learning? ICML 2024
>
> [8] Yedi Zhang, et al., Training Dynamics of In-Context Learning in Linear Attention, ICML 2025.
>
> [9] Catherine Olsson et al. In-context Learning and Induction Heads. 2022

---

> > ### Comment · Reviewer_MTmD · 2025-08-05
> >
> > I thank the authors for their responses and clarifications. I have also gone over the answers to other reviewers. I have some follow-up questions and remarks, listed in the same order as the authors' responses.
> >
> >
> > ---
> >
> > **1. Setup clarification:** I thank the reviewers for the clarification. While I understand the differences, I still believe that their setup is likely more natural, given that it aligns with how most models are typically trained. That said, I acknowledge that the problem addressed here is interesting and worth exploring.
> >
> > **2. Global minima and reparameterization:**. Upon examining Figure 5, there appears to be a significant deviation between the reparameterized model and LSA as the dimensionality increases. This raises the question: is the reparameterized model a sufficiently accurate proxy for understanding what LSA is learning? While I understand that the two models coincide when $d = 1$, this is a highly constrained and specific case. Could the authors clarify whether the reparameterized model still provides meaningful insight into LSA's behavior in higher dimensions? Additionally, since the reparameterized objective is strictly convex, convergence to the global optimum is more straightforward compared to LSA. That said, is it possible to directly train LSA in the $d = 1$ case and observe its learned solution? Specifically, does it align with the optimum of the reparameterized model? If such an analysis has already been conducted, I would appreciate it if the authors could point me to the relevant section (apologies if I may have overlooked it).
> >
> >
> > **3. NP-Hardness interpretation:** To clarify, my question was more along the lines of whether the NP-hardness arises due to the authors' specific reparameterization. I thank the authors for clarifying the necessity of the NP-hard reduction. However, my follow-up question is: if I understand correctly, isn’t the NP-hardness result specific to the reparameterized network? In that context, I don’t fully follow the statement: *As shown in Fig. 5 (Section B.1), as the sequence length $d$ increases, the performance gap between the 1-layer LSA trained with gradient descent with 1000 epochs and the reparameterized optimum becomes larger. This aligns with our NP-hardness result, which suggests that recovering the global optimum becomes intractable as $d$ increases.* Could the authors clarify how this gap is connected to the NP-hardness result? Specifically, does the NP-hardness pertain to LSA itself or only to the reparameterized formulation?
> >
> >
> > **4. Multi-Layer behavior:** I have two additional clarification questions. First, could the authors provide some intuition for why *$R_2$ induces a cubic-order correction that places additional emphasis on the last input dimension*? A brief explanation of the mechanism or reasoning behind this behavior would be helpful. Second, I found the point raised by the authors quite interesting: *but beyond a certain depth, optimization deteriorates, all losses begin increasing. The non-monotonic decreasing is likely due to deeper LSA preconditioners negatively affecting the performance* Is this deterioration specifically due to LSA, or is it indicative of a more general phenomenon? Typically, increasing depth tends to improve performance in many architectures. Could the authors comment on why or how this behavior deviates from the conventional trend?

---

> > > ### Author Response · Authors · 2025-08-07
> > >
> > > We thank the reviewer for the detailed feedback and careful revisiting of our work. We are encouraged by your appreciation of the usefulness and applicability of the model studied in the manuscript.
> > >
> > > We would like to respectfully address the remaining concerns below:
> > >
> > > **Regarding reparameterization and NP-hardness results**:
> > > - **Q2.1 - Role of reparameterized model for large $d$.**
> > >     > Does the reparameterized model provide meaningful insights for LSA under higher dimensions given the performance gap?
> > >     - **Reparameterized model establishes LSA's expressivity ceiling.** The reparameterized model provides a theoretical upper bound on LSA performance for any problem dimension. The best loss LSA can achieve is lower-bounded by the optimal reparameterized model loss, as formally established in Remark C.15 (Section C.4). The increasing gap in Fig. 5 illustrates this phenomenon that as dimension increases, LSA requires more compute to reach its potential.
> > >     - **Consistency with broader literature.** This expressivity limitation of single-layer LSA aligns with both existing theoretical [1] and empirical [2] studies demonstrating fundamental constraints of shallow transformer architectures.
> > > - **Q2.2 - LSA optimization behavior for $d=1$.**
> > >     > Is it possible to train LSA in the $d = 1$ case and observe its learned solution? Does the optimal LSA align with the optimum of the reparameterized model?
> > >     - **Empirical verification confirms theoretical predictions.** Yes, the global minimum of LSA for $d=1$ aligns with the reparameterized model. We conducted dedicated experiments training 1-layer LSA for the $d=1$ case to verify the alignment.
> > >         - **Training setup details.** We minimize the empirical loss over B = 10000 prompts, each containing n=100 in-context samples. We train using gradient descent (step size 0.07, 25000 epochs), repeated over 50 random seeds. Each prompt is constructed by sampling initial states i.i.d. from $Bernoulli(0.3)$, and transition probabilities $p_{01}, p_{10} \sim U(0,1)$.
> > >         - **Convergence result.** Table 1 shows the training loss evolution of 1-layer LSA across epochs, with the reparameterized model optimum as reference. The results confirm that for $d=1$, gradient descent successfully achieves the reparameterized optimum, validating our theoretical result in the tractable case.
> > > **Table 1: LSA Training Convergence for $d=1$**
> > > | Epoch | LSA Loss (mean $\pm$ std)   | Optimal Reparameterized Loss  |
> > > | --- | --- | --- |
> > > | 1 | 0.443 $\pm$ 0.026 | 0.403 |
> > > | 100 | 0.423 $\pm$ 0.017 | 0.403 |
> > > | 1000 | 0.411 $\pm$ 0.005 | 0.403 |
> > > | 10000 | 0.404 $\pm$ 0.005| 0.403 |
> > > | 20000 | 0.403 $\pm$ 0.005| 0.403 |
> > > | 25000 | 0.403 $\pm$ 0.005  | 0.403 |

---

> > > > ### Author Response · Authors · 2025-08-07
> > > >
> > > > - **Q3.1 - Source of NP-hardness.**
> > > >     > Is the NP-hardness result specific to the reparameterized network? Does the NP-hardness pertain to LSA itself or only to the reparameterized formulation?
> > > >     - **NP-hardness originates from the interplay of data structure and LSA parameter coupling instead of reparameterization choice.** We demonstrate this through analysis comparing linear vs. Markovian data under two different reparameterization frameworks. We consider the case $d=1$ below and the argument generalizes to general $d$. For $d=1$, the LSA parameters contain $P=[[0,0],[b_1,b_2]],Q=[[a_1,0],[a_2,0]]$.
> > > >         - **Our approach**: Uses extended parameter space $X := [a_1b_1,a_1b_2+a_2b_1,a_2b_2] \in \mathbb{R}^3$
> > > >             - **Linear data** with iid isotropic distribution (described in [3]) yields sparse optimal solution $X^* = [0, n, 0]$ because independence assumptions ($w \perp x_i$, $x_{n+1} \perp x_i$) eliminate cross-correlations ($\mathbb{E}[x_{n+1} y_{n+1} \sum x_i^2] = 0$, $\mathbb{E}[x_{n+1} y_{n+1} \sum y_i^2] = 0$) via zero-mean property of task vectors $w$ ($\mathbb{E}[w] = \mathbb{E}[w^3] = 0$). The optimal LSA parameter settings require $b_1 = a_2=0,a_1\cdot b_2=n$.
> > > >             - **Markovian data** yields dense optimal solution $X^\*$ (Lemma C.1) due to temporal dependencies ($\mathbb{E}[y_{n+1} | x_{n+1}] = x_{n+1} p_{11} + (1-x_{n+1}) p_{01}$). This leads to a dense parameter recovery with $b_1,a_2\neq 0.$
> > > >
> > > >         - **Ahn et al. [3] approach**: Uses matrix parameterization $X_{ij} =b_i a_j$
> > > >             - **Linear data** yields sparse optimal solution $X_{ij}^\*$ with single nonzero entry ($X_{21}^\* = -\frac{n}{n+2}$). Then any optimal LSA parameter satisfies $a_1b_1=a_2b_1=a_2b_2=0$ and $a_1b_2=X_{21}^\*$.
> > > >             - **Markovian data**, when applied to squared loss, yields dense solutions because temporal correlations break independence assumptions. Therefore solving $b_ia_j=X_{ij}^{\*}$ might not have any solutions if $X^{\*}$ doesn't have special structures that satisfy the equality constraints, e.g., $\frac{X_{12}^{\*}}{X_{11}^{\*}}=\frac{X_{22}^{\*}}{X_{21}^{\*}}$.
> > > >
> > > >     - **Consistent pattern across frameworks**: For general $d$, because of the optimal X* structure, we have the following distinct conclusions for the two data cases.
> > > >         - Linear data → Sparse solutions → Tractable parameter recovery
> > > >         - Markovian data → Dense solutions → NP-hard parameter recovery
> > > >
> > > >         The NP-hardness emerges from the intrinsic temporal structure of Markovian data instead of the choice of reparameterization technique, which serves as an analytical tool to understand expressivity boundaries.
> > > >
> > > > - **Q3.2 - Connecting empirical gaps to theoretical hardness.**
> > > >     > How is the performance gap between LSA and reparameterized model connected to the NP-hardness result?
> > > >     - **Theoretical impossibility drives empirical gaps.** As d increases, the NP-hardness result indicates efficiently recovering optimal parameters from X* is a highly complex problem. This theoretical difficulty constrains what gradient descent can achieve in practice.

---

> > > > > ### Author Response · Authors · 2025-08-07
> > > > >
> > > > > **Regarding multi-layer interpretation**:
> > > > > - **Q4.1 - Origin of cubic-order correction in $R_2$**
> > > > >     > Why does $R_2$ induce a cubic-order correction that places additional emphasis on the last input dimension?
> > > > >
> > > > >     - **Temporal dependencies in Markovian data force cubic-order terms.** For Markovian data, temporal correlations ($\mathbb{E}[y_{n+1} | x_{n+1}] = x_{n+1} p_{11} + (1-x_{n+1}) p_{01}$) require activating the last row of Q ($a  \neq 0$) to capture label-context dependencies. This creates coupling terms $a  \sum_{i=1}^d b_i (\frac{1}{n}\sum_{k=1}^n z_{k,i} z_{k,d+1})$ involving second-order cross-statistics between context inputs $z_{k,i}$ and labels $z_{k,d+1}$. In our multi-layer interpretation, $R_2$ is constructed as the objective whose gradient produces these coupling terms.
> > > > >     - **Architectural contrast with linear tasks.** For linear data, keeping $a = 0$ avoids these label-dependent coupling terms, allowing the simpler quadratic objective in $R_1$ to suffice. Markovian data requires the cubic objective $R_2$ whose gradient can produce the necessary second-order cross-statistics between inputs and labels.
> > > > >    - **Moving towards fuller attention utilization.** The progression from quadratic (linear tasks) to cubic terms (Markovian tasks) reflects utilizing more attention parameters that were previously restricted to zero in prior work [3,5], revealing more of the attention mechanism's expressivity.
> > > > > - **Q4.2 - Non-monotonic optimization behavior**
> > > > >     > Why does the non-monotonic behavior of objectives $R_1$ and $R_2$ deviate from conventional trends? Is this deterioration LSA-specific or more general?
> > > > >
> > > > >     - **Two-phase optimization dynamics reflect multi-objective nature.** The non-monotonic behavior arises from LSA simultaneously optimizing multiple competing objectives, unlike single-objective optimization in the iid ICL setting [3].
> > > > >         - **Phase 1 - Cooperative descent:** Initially, all objectives move along similar direction as the model is far from optimality frontier, allowing coordinated progress across $R_1$ and $R_2$.
> > > > >         - **Phase 2 - Competitive trade-offs:** As optimization approaches the Pareto frontier, objectives begin competing. We observe conflicting behavior where some components in $R_2$ improve while others in $R_1$ deteriorate, reflecting the fundamental trade-offs inherent in multi-objective optimization. [4]
> > > > >     - **Novel multi-objective phenomenon.** This behavior emerges from utilizing more attention parameters for structured tasks. Unlike linear i.i.d. tasks that reduce to single-objective optimization, Markovian temporal structure necessitates balancing multiple competing objectives. To our knowledge, this is the first demonstration of such multi-objective dynamics emerging from transformer architecture for ICL of structured functions.
> > > > >
> > > > > **References**
> > > > >
> > > > > [1] Clayton Sanford et al., Representational Strengths and Limitations of Transformers. NeurIPS 2023
> > > > >
> > > > > [2] Catherine Olsson et al., In-context Learning and Induction Heads. 2022
> > > > >
> > > > > [3] Kwangjun Ahn, et al., Transformers learn to implement preconditioned gradient descent for in-context learning. NeurIPS 2023.
> > > > >
> > > > > [4] Kaisa Miettinen, Nonlinear Multiobjective Optimization. Springer 1999.
> > > > >
> > > > > [5] Ruiqi Zhang, et al., Trained Transformers Learn Linear Models In-Context. J. Mach. Learn. Res. 25: 49:1-49:55 2024

---

> > > > > > ### Comment · Reviewer_MTmD · 2025-08-07
> > > > > >
> > > > > > I thank the authors for all of their detailed comments and clarifications. I have no further questions. I will update my score accordingly based on the new results and the clarifications provided. Thank you once again for the responses.

---

> > > > > > > ### Author Response · Authors · 2025-08-08
> > > > > > >
> > > > > > > We sincerely thank you for your constructive engagement with our work. We are glad our responses have helped address the concerns, and we appreciate your comments, which have helped improve the clarity of our paper.

---

### Note · Authors · 2025-08-13

We appreciate the constructive engagement from all reviewers throughout the discussion phase. We are encouraged by the positive feedback recognizing our novel ICL setting for Markovian dynamics, rigorous theoretical analysis, and insights into transformer expressivity limitations in structured ICL tasks. Below we summarize how the main concerns raised across reviewers were addressed:

- **NP-hardness's source & implication**: Demonstrated computational hardness emerges from bilinear transformer parameter coupling combined with Markovian data dependencies, establishing that global minimum recovery becomes intractable as problem complexity increases, thereby revealing expressivity ceiling for shallow architectures, and offering theoretical insight into depth benefits observed empirically.
- **Simplified setting & generality**: Concerns about the binary Markov setting and shallow LSA architecture were addressed by noting that such simplifications are standard in theoretical ICL studies for analytical tractability, while still preserving essential attention mechanisms. The core insights on how global minimum recovery difficulty reveals expressivity limitations are broadly applicable.
- **Multi-layer interpretation**: Clarified nonzero attention weights required by Markovian dependencies enable novel multi-objective optimization interpretation, revealing how structured ICL tasks necessitate preconditioned gradient descent across multiple objectives over in-context examples, a discovery distinct from conventional single-objective interpretation for the i.i.d. case.
- **Experimental scope**: Validated LSA convergence in tractable cases; expanded experiments with deeper models (including Bayes error baselines), linking our findings to empirical trends in the literature on layer depth and ICL capability.

We sincerely thank all reviewers for their valuable feedback, the time spent engaging in discussions, and their recognition of our work.

---

### Decision · Program_Chairs · 2025-09-17

**Decision:**

Accept (poster)

**Comment:**

This paper considers an in-context learning problem to learn Markovian Dynamics instead of usual regression problems. The authors show the explicit form of the optimal solution of a linear self-attention network for the binary Markov chain, and prove NP-hardness of computing the optimal solution in general. In addition to that, the authors show that the multi-layer model corresponds to a pre-conditioned gradient descent. These theoretical findings are properly supported by numerical experiments.

Although the model of the analysis is quite limited (LSA), this paper investigates an important problem which generalizes the conventional i.i.d. regression problem to next token prediction. Unlike some related works considering non-i.i.d. dynamics estimation, this paper analyzes a sequence dependent ICL problem. This is a novel contribution. Although it is still at a basic level, it will lead to more general analyses.

The numerical experiments also well support the theoretical findings. The paper is well written and properly survey the existing literatures.

For these reasons, I think this paper is a good paper. I recommend acceptance.